# Spa2 remodels ADP-actin via molecular condensation under glucose starvation

Qianqian Ma[1], Wahyu Surya [1], Danxia He[1], Hanmeng Yang[1], Xiao Han[1], Mui Hoon Nai [2], Chwee Teck Lim [2,3], Jaume Torres [1] & Yansong Miao [1,4] ✉

Actin nucleotide-dependent actin remodeling is essential to orchestrate signal transduction and cell adaptation. Rapid energy starvation requires accurate and timely reorganization of the actin network. Despite distinct treadmilling mechanisms of ADP- and ATP-actin filaments, their filament structures are nearly identical. How other actin-binding proteins regulate ADP-actin filament assembly is unclear. Here, we show that Spa2 which is the polarisome scaffold protein specifically remodels ADP-actin upon energy starvation in budding yeast. Spa2 triggers ADP-actin monomer nucleation rapidly through a dimeric core of Spa2 (aa 281-535). Concurrently, the intrinsically disordered region (IDR, aa 1-281) guides Spa2 undergoing phase separation and wetting on the surface of ADP-G-actin-derived F-actin and bundles the filaments. Both ADP-actin-specific nucleation and bundling activities of Spa2 are actin D-loop dependent. The IDR and nucleation core of Spa2 are evolutionarily conserved by coexistence in the fungus kingdom, suggesting a universal adaptation mechanism in the fungal kingdom in response to glucose starvation, regulating ADP-G-actin and ADP-F-actin with high nucleotide homogeneity.

Undesired environments often lead to energy starvation (ES) in eukaryotic cells, which limits growth by interrupting diverse active-running ATP-dependent processes. To adapt to harsh conditions under ES, cells enter a dormant state as an adaptation strategy before returning to a normal cell state to survive harsh conditions. Actin filament treadmilling maintains diverse cellular activities by undergoing balanced polymerization and depolymerization. A major physiological ES condition is glucose starvation (GS), which was found to remodel actin filaments in budding yeast acutely within a few minutes by converting actin monomer to actin filament and inducing a rapid stabilization of actin cables that are resistant to cofilin-mediated severing[1–3]. Actin cables are thin bundles of actin filaments that serve as tracks to direct biomolecule transport for polarized fungal growth[4]. Actin cable treadmilling is orchestrated spatiotemporally by a series of actin-binding proteins (ABPs) to drive stepwise polymerization and depolymerization steps, coordinating with the nucleotide exchange of actin. GS-induced ATP-depleting conditions interrupted the turnover rate of ATP hydrolysis and inorganic phosphate ($P_i$) release and decreased ATP level, thereby leading to faster aging of the actin filament from ATP to ADP subunits. The primary distinction lies in the ATP-actin and of ADP-actin with changing the conformation of D-loop which is a nucleotide sensing region, inorganic phosphate release traps the closed conformation of D-loop[5–7]. Strikingly, instead of being depolymerized by ADP-actin-based depolymerization factors, the GS triggers actin cable crosslinking to stabilize actin filament and retain the mass of actin networks. An earlier genetic screening found that GS-triggered actin cable bundling is independent of two cable-binding proteins, type V myosin and tropomyosin[1]. However, the underlying mechanism remains unknown, and the ABPs responsible for stabilizing ADP-actin filament have not been identified.

In this study, we demonstrate that Spa2 is the primary regulator that drives the actin cable remodeling during energy starvation. Spa2

[1]School of Biological Sciences, Nanyang Technological University, 637551 Singapore, Singapore. [2]Department of Biomedical Engineering, National University of Singapore, 117583 Singapore, Singapore. [3]Institute for Health Innovation and Technology (iHealthtech), National University of Singapore, 119276 Singapore, Singapore. [4]Institute for Digital Molecular Analytics and Science, Nanyang Technological University, 636921 Singapore, Singapore. ✉e-mail: yansongm@ntu.edu.sg

governs ADP-actin nucleation and bundling through a multilayered spatial and temporal control mechanism that depends on the D-loop conformation. Specifically, a nucleation core within Spa2 (281-535) directly facilitates the polymerization of ADP-bound G-actin. The intrinsically disordered region (IDR, 1-281) of Spa2 allows for subtle multivalent interactions, enabling the N-terminal Spa2 (1-535) to crosslink ADP-F-actin gradually. This cross-linking is facilitated by wetting on the filament surface during phase separation. This reminisces previous findings where actin remodeling is initiated by membrane-associated actin-binding proteins, such as during T-cell immune signaling and plant immune responses upon phytobacterial infection through molecular condensation[8–11]. Here, our research illuminates the energy starvation-induced actin remodeling in budding yeast, specifically mediated through the molecular condensation of Spa2.

## Results

### The N-terminal of Spa2 is responsible for actin cable stabilization upon energy starvation

Upon glucose starvation (GS), the green fluorescent protein fused to actin-binding protein Abp140 (Abp140-3×GFP) marked actin cables in budding yeast showed time-dependent oscillation with rapid bundling at 5 min and recovery at 30 min (Fig. 1a and b, Supplementary Movie 1). The spatially regulated actin cable remodeling was consistent with the initial ~38% drop in the ATP concentration, measured using ATP-biosensor QUEEN[12], after 5 min and with its restoration after 30 min due to respiration provided by bulk autophagy and β-oxidation[5] (Fig. 1c, d, Supplementary Movie 2). In the major glucose kinase mutant hxk2Δ, the GS-triggered rapid actin remodeling at 5 min was abolished, and the change in the ATP level was no longer sensitive to rapid GS (Fig. 1a–d). Similarly, the anaerobic respiration mutant cbp2Δ in which splicing of cytochrome B oxidase preRNA is disabled, was unable to reverse GS-induced actin bundling at 30 min, when ATP could not be replenished[1,13] (Fig. 1a–d). In addition, we found that GS enhanced overall actin cable production by ~2-fold (Supplementary Fig. 1a). This increase was greater than the increase in bundling level (Fig. 1b), suggesting a concurrent increase in actin cable polymerization. Additional ATP decrease conditions were also examined by using the glucose analog 2-deoxyglucose (2-DG). 2-DG diminishes the glycolytic supply of ATP and aerobic respiration-regenerated ATP in all WT, hxk2Δ, and cbp2Δ strains, in which actin cables all showed bundled patterns after 5 min treatment (Supplementary Fig. 1b–d). In addition, in the ATP-production-uncoupled mutants of the G-protein-coupled glucose receptor gpr1Δ and glucose sensors snf3Δrgt2Δ, the changes in the ATP and actin cable bundling were similar to those in the WT (Supplementary Fig. 1e–h). Additionally, we observed that in the absence of GS treatment, there is a significant difference in bundle levels between the hxk2Δ and cbp2Δ strains compared to the WT. This difference might be attributed to changes in metabolite levels.

Next, we sought to identify the potential regulatory actin-binding protein (ABP) by screening ABP mutants under energy starvation (ES) conditions, using both GS and 2-DG. Assuming Abp140-3×GFP signal changes on actin filaments tightly correlate with the changes in actin cable production, we characterized Abp140-3×GFP in mutants that are defective in either actin cable nucleation (bni1Δ, bnr1Δ, bud6Δ, and aip5Δ), capping (cap1Δ and cap2Δ), actin cable crosslinking (tpm1Δ, sac6Δ and myo2-66) or depolymerization (cof1-4). However, none of the above-listed mutants abolished the ES-induced enhancement in actin cable production and bundling (Supplementary Fig. 2a, b), except for a mutant knocking out SPA2, which encodes a polarisome complex scaffolding protein[14,15]. In spa2Δ or spa2Δcbp2Δ, the actin cables were insensitive to GS (Fig. 1e, f, Supplementary Fig. 2c, Supplementary Movie 3), although a similar decrease in the ATP remained as in the WT (Fig. 1g).

Next, we asked if Spa2 localizes to the actin cable during ES. After 5 min of ES, bud tip-localized Spa2-GFP were depolarized from the cortex and formed filamentous structures or puncta along the filaments, which largely overlapped with Abp140-Tomato-labeled actin cable bundles (Fig. 1h, i, Supplementary Movie 4, 5). Such relocation of Spa2-GFP upon GS was not observed in hxk2Δ. The relocated filamentous Spa2-GFP was unable to be restored to the bud tip in cbp2Δ after 30 min (Fig. 1j, k, Supplementary Movie 6). Except for spa2Δ, all the examined ABP mutants and glucose-related mutants (gpr1Δ and snf3Δrgt2Δ) exhibited a similar response as the WT, forming Spa2-GFP filaments upon ES, similar to actin cable bundling (Supplementary Fig. 2d, e).

We then dissected the core regions of Spa2 that are responsible for ES-triggered actin cable remodeling. Based on the prediction of the structured and intrinsically disordered region (IDR)[16], we created GFP-tagged Spa2 variants with truncations. In vivo localization experiments showed that Spa2(1-535) was the shortest examined version that retained ES-triggered filament formation (Fig. 2a–c, Supplementary Fig. 3a, Supplementary Movie 7, 8). Spa2(1-535) displayed a weak filamentous pattern on its own under normal growth conditions (Fig. 2b). In contrast, Spa2(281-535)-GFP and Spa2(1-281)-GFP showed cytoplasmic puncta and diffusive patterns, respectively, and neither of them formed cables (Fig. 2b–d). spa2(1-535) cells maintained the ability to crosslink and stabilize actin cables by ES, but spa2(281-535) and spa2(1-281) cells did not (Fig. 2e, f, Supplementary Movie 9, 10). However, spa2(281-535), but not spa2(1-281), could still increase overall actin cable production upon ES (Fig. 2g, Supplementary Fig. 3b). In addition, Spa2(1-535)-GFP filaments also showed ATP-dependent bundling upon GS in the WT and cbp2Δ, similar to the thickening of Spa2-GFP and Abp140-3×GFP cables, but showed little response in hxk2Δ (Supplementary Fig. 3c–e). Moreover, we measured the concentration of Spa2 at the bud tip and the concentrations of Spa2 variants in vivo. Similarly, the concentrations of Spa2(1-535) in the filament and Spa2 (1-281) in the cytosol are also approximately 1 μM, mirroring the Spa2 concentration at the bud tip. However, the concentration of Spa2(281-535) in puncta was lower, approximately 0.3 μM. This reduction may be attributable to a lower assembly state without IDR-mediated multivalent interactions (Supplementary Fig. 3f).

### Spa2 is an ADP-specific actin nucleator that functions through Spa2(281-535)

We could not find any homologous region of Spa2(1-535) in known actin nucleators, including the Arp2/3 complex, formins, or WH2-domain family proteins. To understand the Spa2(1-535)-mediated increase in actin cable signal in vivo, we first examined the potential interactions between Spa2(1-535) and actin monomer in both ATP- and ADP-bound states. Recombinant Spa2(1-535), Spa2(281-535), and Spa2(1-281) proteins were expressed and purified from bacteria (Supplementary Fig. 4a) and mixed with nonpolymerizable yeast ADP-actin monomer or ATP-actin monomer. The introduced mutations (D286A/V287A/D288A) of actin are strategically located at the interfaces between actin monomers in a longitudinal contact, ensuring they do not interfere with the D-loop. The binding affinities were subsequently monitored using a fluorescence anisotropy assay. Spa2(1-535) and Spa2(281-535) showed similar affinities toward ADP- actin monomer, with Kd values of $0.52 \pm 0.25$ μM and $0.21 \pm 0.05$ μM, respectively, whereas no ATP-actin monomer binding was detected (Fig. 3a, Supplementary Fig. 4b). Spa2(1-281) bound neither ADP-actin nor ATP-actin monomer (Fig. 3a, Supplementary Fig. 4b). We next examined actin nucleation and elongation activities via a total internal reflection fluorescence microscopy (TIRFM)-actin polymerization assay. To mimic the in vivo actin remodeling of preexisting actin filament upon ES, we applied 50 nM Cof1 and Spa2 variants to preformed actin filament derived from ATP-actin monomer. When actin filaments were quickly severed by Cof1, we observed that after ~80 s from the samples

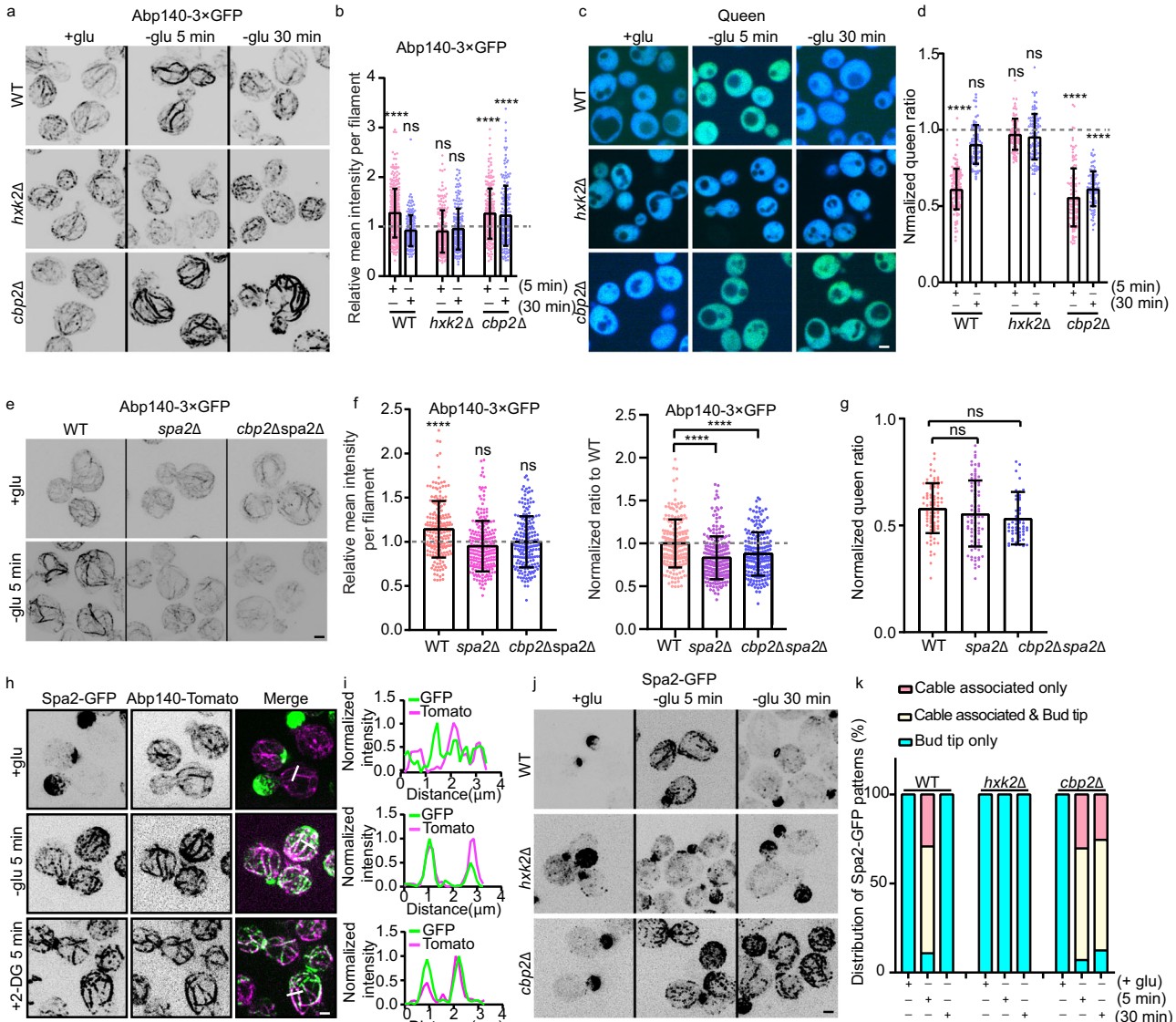

**Fig. 1 | Spa2 regulates actin cable remodeling upon energy starvation.** Representative maximum Z-projection images (**a**) and quantification of ABP140-3×GFP signals (**b**) in WT, *hxk2Δ*, and *cbp2Δ* cells upon glucose starvation for the indicated time points before imaging. *n* = 312, 186 206,194, 209, 182 cables from ~20 cells per condition. The dashed line indicates 1. (**c, d**) Representative images of the ATP-sensor QUEEN in the indicated yeast by excitation at 410 nm or 480 nm and ratiometric analysis (from left to right: *n* = 115, 73, 96, 94, 128, 99 cells). The dashed line indicates 1. **e, f** Representative images of Abp140-3·GFP in WT, *spa2Δ*, and *spa2Δcbp2Δ* grown upon rapid GS for 5 min and fluorescence intensity quantification. *n* = 195, 229 and 196 cables from >20 cells per condition. The dashed line indicates 1. **g** The normalized QUEEN ratio (410 nm ex/480 nm ex) in WT, *spa2Δ*, and *spa2Δcbp2Δ* cells was calculated from the signal intensity of each pixel to

generate the QUEEN ratio image of cells. *n* = 82, 84, and 56 cells from left to right. **h** Dual-color images of Spa2-GFP and Abp140-Tomato upon ES under the indicated conditions. **i** Colocalization analysis of Spa2-GFP and Abp140-Tomato by 20-pixel-long line cross filaments. **j, k** Representative maximum Z-projection images of Spa2-GFP localization in WT, *hxk2Δ* and *cbp2Δ* after 5 min and 30 min of GS and quantification of the distribution pattern of Spa2 under the indicated conditions. *n* = 150, 175, 150, 150, 150, 150, 150, 142, 153 cells from left to right. Scale bars, 2 μm. Means, SDs, and *P*-values were calculated using GraphPad Prism 6. The significance between two sets of data values was determined by the one-way analysis of variance (****p < 0.0001, and ns = not significant). Source data is provided as a Source Data file.

containing either the additional Spa2(1-535) or Spa2(281-535), but not Spa2(1-281), in which a large number of short actin filaments were produced (Fig. 3b, c, Supplementary Movie 11). However, when we introduced Spa2 variants to actin filaments prepared by ATP-actin monomer, we didn't observe any noticeable differences compared to the actin control, as fragment filaments were not evident, this means that Spa2 is less likely to have severing ability on its own (see Supplementary Fig. 4c). We next quantified the cumulative severing events from 20 to 80 seconds frames. We observed a subtle distinction between Cof1 alone and Cof1 in conjunction with Spa2 (1-535), particularly in comparison to other Spa2 variants, when evaluating the severing rate during the early depolymerization phase spanning from

20 to 60 seconds (Supplementary Fig. 4d, e). This observation implies Spa2(1-535)-aided slight severing through Cof1 requires its IDR. Nonetheless, given the substantial increase in short filaments observed not only in Spa2(1-535) but also in Spa2(281-535), which did not enhance Cof1-mediated severing, our findings suggest that the primary contributing factors may be derived from de novo ADP-actin polymerization in cope with Cof1-mediated F-actin severing.

We then asked whether Spa2 can nucleate ADP-bound actin monomers. We found that both Spa2(281-535) and Spa2(1-535), but not Spa2(1-281), were highly potent in nucleating ADP-actin monomer by rapidly increasing seeds even at a low concentration of 10 nM, in the TIRFM-actin assay (Fig. 3d, Supplementary Movie 12). The ADP-actin

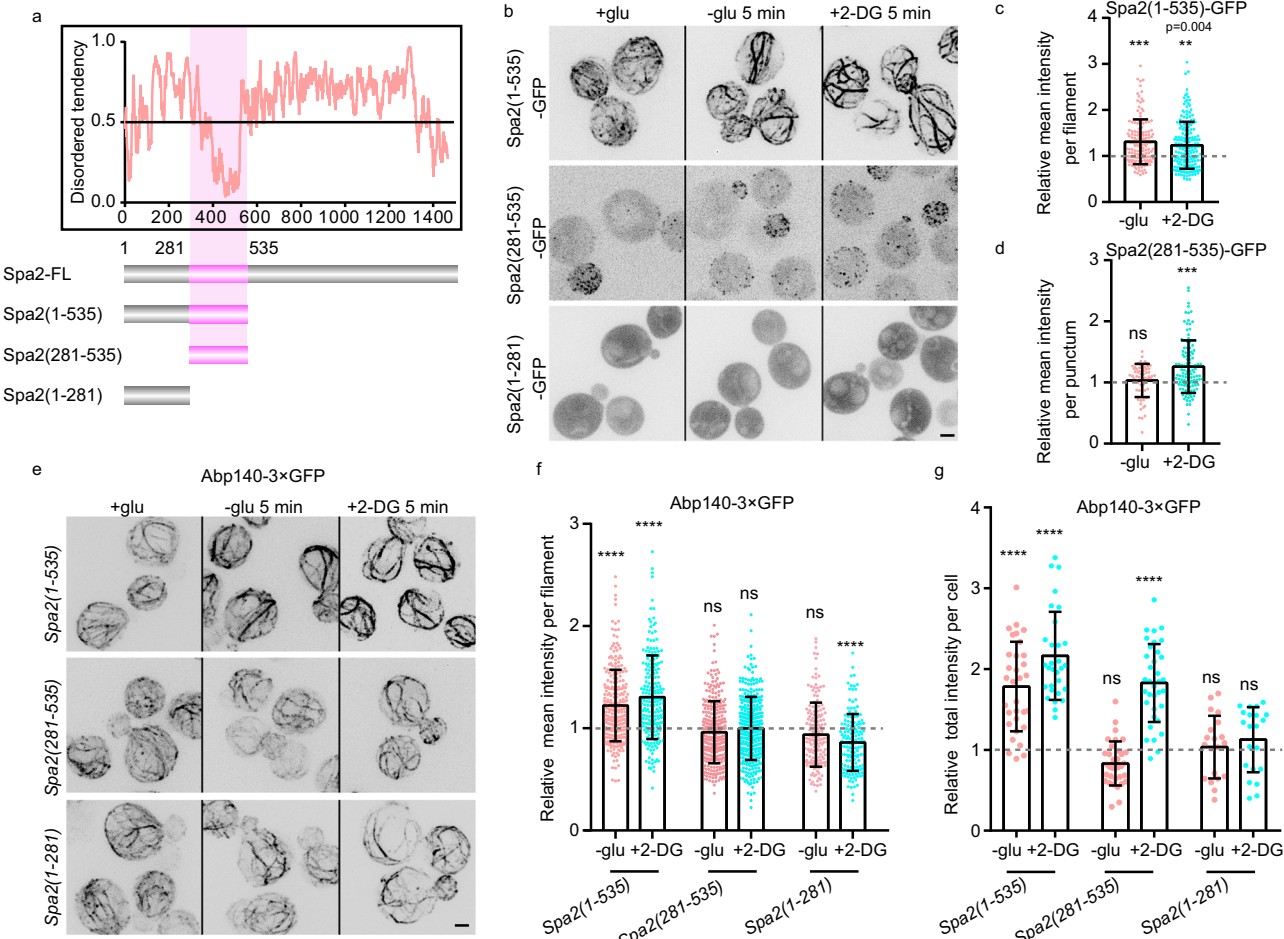

**Fig. 2 | Spa2(1-535) regulates actin cable remodeling upon ES. a** Intrinsically disordered region analysis. **b** Representative fluorescent images of GFP-tagged Spa2 truncation variants upon ES under the indicated conditions. Scale bars, 2 μm. **c, d** Quantification of the mean signal intensity of Spa2(1-535)-GFP filaments and puncta of Spa2(281-535)-GFP upon ES ($n = 14, 19, 20$ and 20 cells for each condition). The dashed line indicates 1. **e** Localization of Abp140-3×GFP in Spa2 truncating mutants, with or without ES at 5 min. Scale bars, 2 μm. **f, g** Quantification of the mean intensity of Abp140-3×GFP per filament and total signal intensity per indicated truncated Spa2 mutant cells in the presence or absence of ES (from left to right: $n = 266, 207, 299, 327, 166, 150$ actin filaments for **f**; $n = 32, 30, 34, 35, 20, 21$ cells for **g**). The dashed line indicates 1. Means, SDs, and *P*-values were calculated using GraphPad Prism 6. Significance between two sets of data were determined by the one-way analysis of variance (***$p < 0.001$, ****$p < 0.0001$, **$p < 0.01$ and specified and ns = not significant). Source data is provided as Source Data file.

nucleation rate was increased by ~5.4-fold with Spa2(281-535) and ~2.9-fold with Spa2(1-535) (Fig. 3e). In addition, the elongation rate was also boosted by ~2.8-fold by either Spa2(281-535) or Spa2(1-535) (Fig. 3f). Additionally, we evaluated the ratio of total actin intensity at 10 minutes to the actin elongation speed. In comparison to the ADP-actin control, Spa2(1-535) increased this ratio approximately 3.3-fold, while Spa2(281-535) resulted in a roughly 4.4-fold increase. These results further suggest that Spa2(1-535) acts as an ADP-actin nucleator (Supplementary Fig. 4f). To rule out the potential defect of high ADP-actin monomer concentration, we conducted a TIRF actin assay using 0.8 μM ADP-actin monomer. Our results showed that both Spa2(1-535) and Spa2(281-535), but not Spa2(1-281), increased the actin nucleation rate and elongation speed (Supplementary Fig. 4g–i), which aligns with the findings using 3 μM ADP-actin. Additionally, our bulk actin pyrene assay demonstrated that both Spa2(1-535) and Spa2(281-535) promote actin polymerization, while Spa2(1-281) did not (Supplementary Fig. 4j). Typically, actin-binding proteins involved in actin filament assembly may affect the critical concentration for nucleation. To investigate whether Spa2 regulates ADP-actin critical concentration, we conducted an experiment which indicated that Spa2 did not influence the ADP-actin critical concentration (Supplementary Fig. 4k). In contrast, at the same concentration, neither Spa2(281-535) nor Spa2(1-535) could promote the polymerization of ATP-actin monomer

(Supplementary Fig. 4l–n, Supplementary Movie 13) or ADP-$P_i$-G-actin (Supplementary Fig. 4o–q, Supplementary Movie 14). Our results indicate that Spa2 nucleates and elongates monomeric actin in an ADP-specific manner by its N-terminus, where the region from 281 aa to 535 aa of Spa2 is critical.

## Molecular condensation of Spa2 depends on the valency derived from its N-terminal IDR

The promotion of actin dimer and trimer formation can effectively overcome the rate-limiting step of actin polymerization[17]. Multivalent binding of actin monomer via protein oligomerization has also been known to promote nucleation[18] or increase the elongation rate by incorporating multiple actin monomers at the growing end[19,20]. Understanding how the multivalent motif and catalytic domain work together to create functional assembly states is crucial[9,21]. Then, we tested our hypothesis that the N-terminal domains of Spa2 are in a low oligomeric state that enables ADP-actin nucleation and elongation. First, we determined the oligomeric states of Spa2 fragments using an analytical ultracentrifugation sedimentation velocity (AUC-SV) experiment. The estimated size of the oligomer formed by the three Spa2 fragments indicated a dimeric state for both Spa2(1-535) ($MW_{app} = 111$ kDa) and Spa2(281-535) ($MW_{app} = 62.4$ kDa), but a monomeric state for Spa2(1-281) ($MW_{app} = 37$ kDa) (Fig. 4a–c). We next

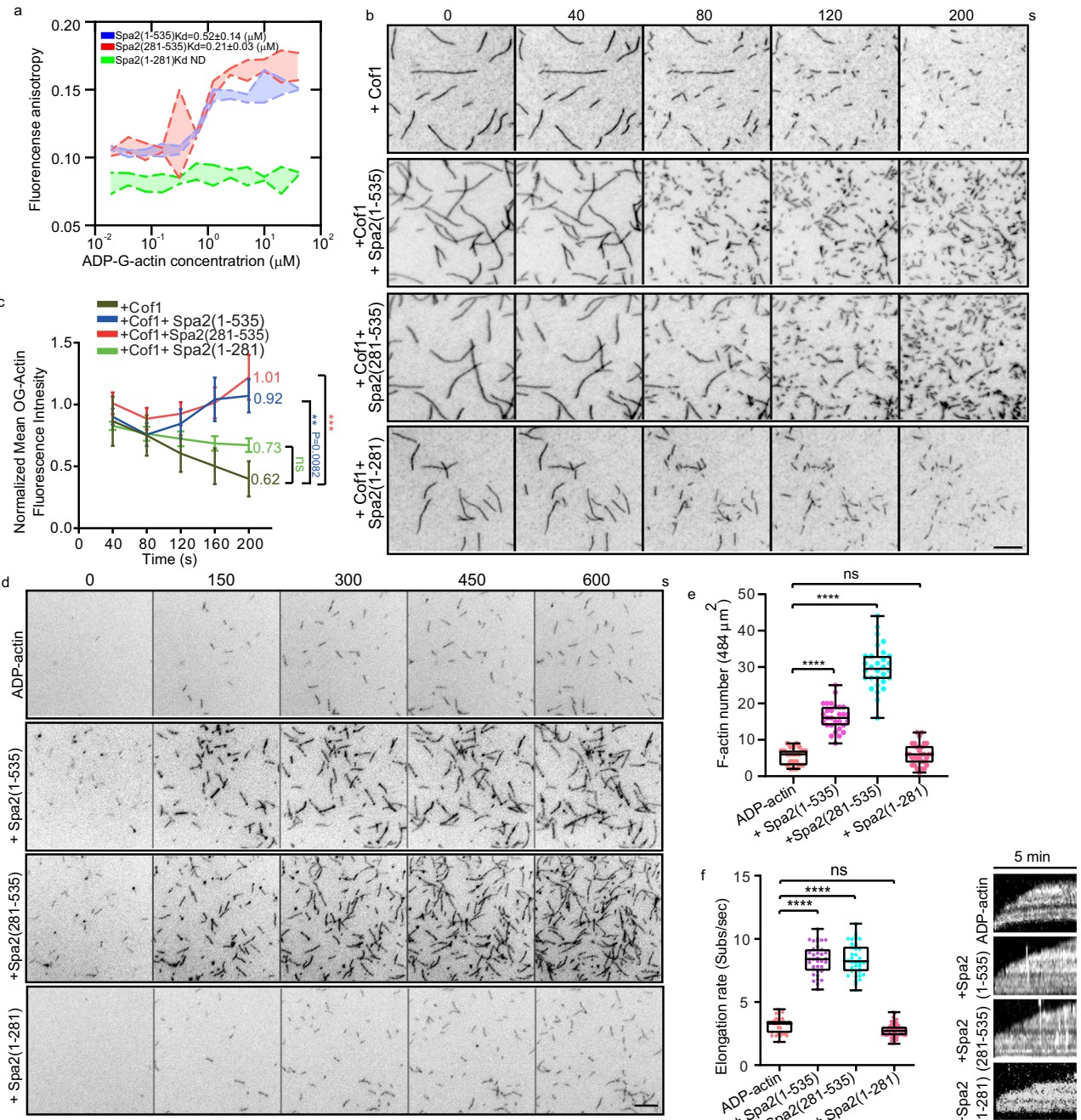

**Fig. 3 | Spa2(1-535) specifically nucleates and elongates ADP-actin. a** The Fluorescence anisotropy profile of the indicated recombinant Spa2-N-terminal truncation variants (60 nM, Alexa 488-labeled) were titrated with increasing concentrations of ADP-actin monomer. Measurements from three biological replicates were plotted and fit with the Hill equation. **b** Representative time-lapse TIRFM images of actin filament severing by incubating 500 nM Spa2-truncation variants and 50 nM Cof1 with 0.1 µM F-actin derived from 0.5 µM ATP-G-actin with 10% Oregon Green 488 (OG 488) and 0.5% biotin labeling. **c** Quantification of the average actin filament signal intensity from multiple time points during 40 - 200 s (from top to bottom n = 34, 36, 35, 35 ROIs of 22 × 22 µm²). **d** Representative time-

lapse TIRFM images of actin polymerization at the indicated time point using 3 µM ADP-G-actin (10% OG 488- and 0.5% biotin-labeled), with or without the indicated 10 nM Spa2 variants. **e** Comparison of nucleation efficiency at 5 min by measuring the seed number (from left to right: n = 22, 28, 29, and 29 ROIs at 64×64 µm²). **f** Quantification of the actin elongation rates and kymograph in **d** (left to right, n = 34, 36, 39, and 52 actin filaments). Whiskers represent min to max in **e** and **f**. Bars for b and d, 5 µm; Bar for f, 2 µm. Error bars, mean ± S.D. P-values were calculated using GraphPad Prism 6. Significance between two sets of data were determined by the one-way analysis of variance (****p < 0.0001, ***p <0.001, **p < 0.01 and specified, ns = not significant). Source data is provided as Source Data file.

investigated the interaction modes in the self-association of Spa2 variants underlying their biochemical activities. We investigated their intermolecular interactions via surface plasmon resonance (SPR). Spa2(1-535) displayed weak dimer-dimer interactions, which were derived from the self-interactions between the IDR (Spa2(1-281)),

whereas Spa2(281-535) dimers did not show obvious association between dimers (Fig. 4d–f).

Multivalent interactions of IDR-containing proteins (IDP) have been found to promote cytoskeleton nucleation by undergoing phase separation[8,11,22]. Given the intermolecular interactions shown above for

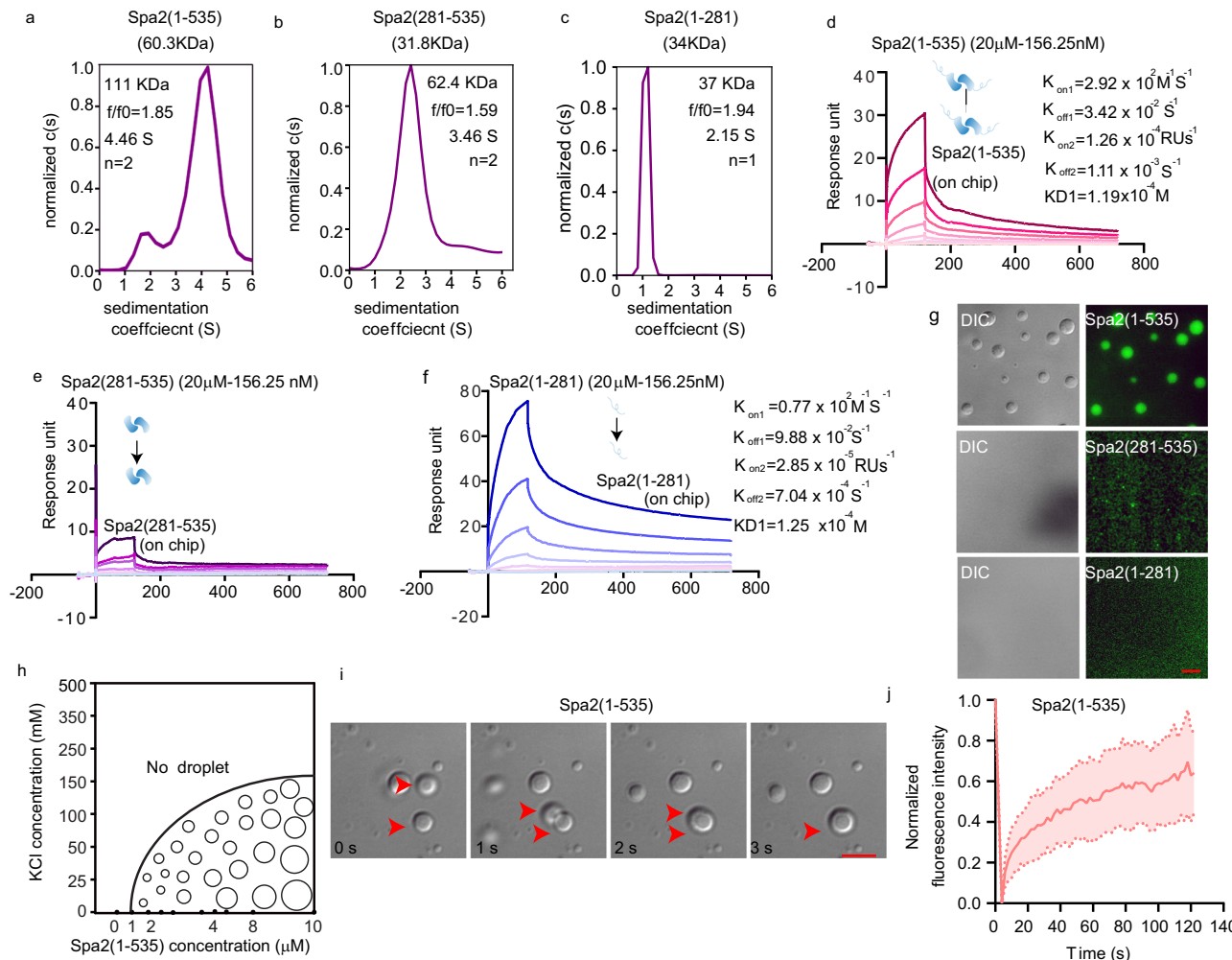

**Fig. 4 | Spa2 undergoes multivalent self-assembly through its N-terminal IDR and dimerization domain of Spa2(281-535). a−c** Sedimentation coefficient distribution of Spa2(281-535) (18 μM), Spa2(1-535) (13 μM), and Spa2(1-281) (17 μM), with calculated apparent molecular weights (*MW*_{app}) indicated. The sedimentation coefficient and frictional ratio (*f/f0*) of the major species of each Spa2 variant are shown. The predicted oligomerization status (n) was indicated based on differential sedimentation coefficient distribution analysis. **d−f** SPR sensorgrams of self-interacting Spa2(1-535), Spa2(281-535), and Spa2(1-281). The corresponding curves were fitted with the bivalent model. Molecules in the mobile and immobilized phases were indicated by the arrow. **g** Representative phase separation images of 5 μM Spa2(1-535), Spa2 (281-535) and Spa2(1-281) (10% Alexa 488-labeled) at 50 mM KCl from two independent experiments. **h** Spa2(1-535) phase diagram showing the protein concentration and KCl concentration dependence. **i** Transmission light time-lapse images of representative coalescence of 5 μM Spa2(1-535) at 50 mM KCl. **j** Fluorescence recovery analysis of Spa2(1-535) proteins after photobleaching. Error bands are ± SD. Scale bar, 5 μm. Source data is provided as a Source Data file.

Spa2(1-281), we then asked whether IDR (Spa2(1-281)) aided in the multivalent assembly of Spa2(1-535) dimers by undergoing liquid–liquid phase separation (LLPS)[23]. Via microscopic imaging, we found that Spa2(1-535) proteins are immiscible with aqueous solvents and exhibit protein concentration- and ionic strength-dependent macromolecular condensation by forming spherical droplets (Fig. 4g, h). The homotypic assembly of Spa2(1-535) displayed LLPS properties with coalescence (Fig. 4i; Supplementary Movie 15) and high fluidity with rapid recovery after fluorescence photobleaching (Fig. 4j, Supplementary Fig. 5a).

The wetting of the microtubule by the phase-separated targeting protein for Xklp2 (TPX2) results in TPX2-tubulin cocondensation[24,25]. The dynamic multivalent Spa2(1-535) interactions motivated us to ask whether the Spa2(1-535) droplets would coat actin filament in an ADP-specific manner. We incubated actin filaments prepared from monomeric ADP-actin and ATP-actin, respectively, with 5 μM Spa2(1-281), Spa2(281-535), or Spa2(1-535). None of the Spa2 variants were associated with or changed actin filaments, prepared through 30-minute polymerization with ATP-actin monomers (Supplementary Fig. 5b). In contrast, Spa2(281-535) proteins were decorated along filamentous

actin filaments prepared from ADP-actin monomer as puncta, whereas Spa2(1-535) proteins condensed on actin filaments prepared from ADP-actin monomer on which Spa2(1-535) associated and crosslinked actin filaments (Fig. 5a). Spa2(1-535) mediated actin crosslinking in a concentration-dependent manner with a high correlation to Spa2 homotypic phase separation behavior (Supplementary Fig. 5c), suggesting Spa2 valency-dependent ADP-actin filament bundling. We intended to mitigate the issue of nucleotide heterogeneity due to rapid ATP hydrolysis in F-actin, where most protomers transition to ADP-P_i or ADP states, we compared actin filaments prepared from 0.5 μM ATP-actin monomer over 30 minutes with those from 3 μM ATP-actin monomer over 2 minutes, shortest incubation we can manage. Notably, even in 2-minute polymerized F-actin from ATP-actin monomers, there could be around 30% ADP-actin and 50% ADP-P_i-actin[26]. Despite these conditions, our results indicate that Spa2(1-535) exhibited no crosslinking effect in either scenario. Furthermore, we prepared F-actin from 3 μM AMPPNP[27–33], which is a slowly hydrolyzed ATP analog and stable for at least 16h[33], before incubating with different concentrations of Spa2(1-535), the results were consistent: Spa2(1-535) did not exhibit any crosslinking effect, corroborating our observations

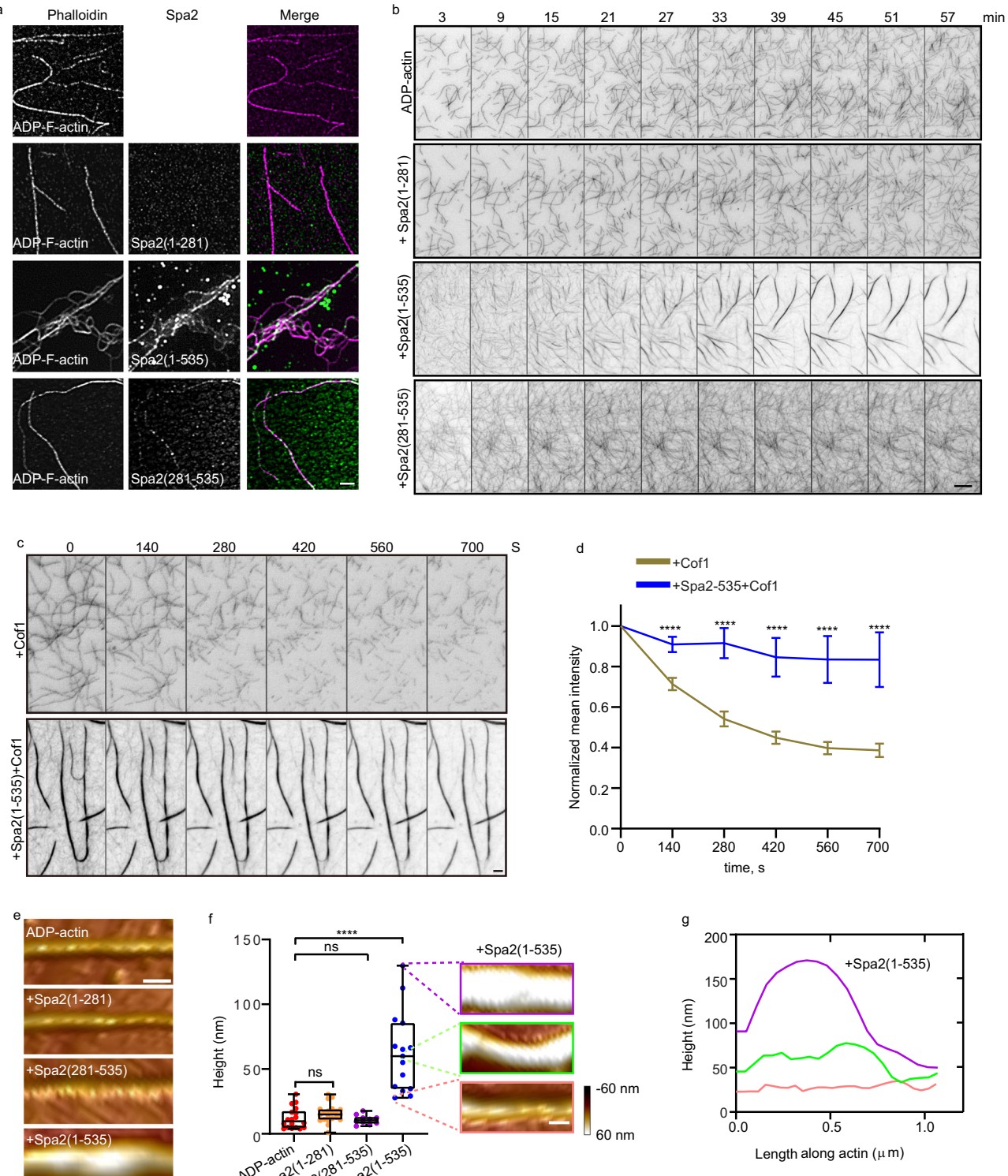

**Fig. 5 | Spa2 proteins crosslink ADP-actin filaments. a** Fluorescence micrographs 3 μM ADP-actin filament incubated with 5 μM Alexa 488-labeled Spa2(1-281), Spa2(1-535), and Spa2(281-535) and labeled with 565-phalloidin before being imaged by spinning disk confocal microscope-coupled superresolution imaging.
**b** Representative time-lapse TIRFM images of ADP-F-actin formed at the indicated timepoint from 3 μM ADP-G-actin (10% OG 488- and 0.5% biotin-labeled) in the presence of 5 μM Spa2 variants. **c** Representative time-lapse TIRFM images of Cof1-mediated depolymerization. ADP-F-actin were formed for 30 min from 3 μM ADP-G-actin (10% OG 488- and 0.5% biotin-labeled) in the absence or presence of 5 μM Spa2(1-535), before adding 100 nM Cof1. **d** Quantification of the average actin

filament signal intensity over time in (c) (both $n = 17$ ROIs of 25×25 μm²). Error bars, mean ± S.D. **e, f** Representative AFM image and height quantification of ADP-actin filament without or with Spa2 truncating variants (from left to right: $n = 18, 21, 10, 15$ filaments). Error bars, mean ± S.D. Whiskers represent min to max.
**f, g** Representative topography AFM images and height measurement of three populations of low height, average height, and high height of the actin filament without or with Spa2(1-535). P-values were calculated using GraphPad Prism 6. The significance between two sets of data were determined by the one-way analysis of variance (****$p < 0.0001$, ns = not significant). Scale bars are 2 μm in a and c; 5 μm in b; 200 nm in e and f. Source data is provided as a Source Data file.

with actin filament prepared from ATP-actin monomer (Supplementary Fig. 5c). Additionally, the application of 5% PEG, which introduces the crowding and enhances condensation, reduced the concentration of Spa2(1-535) to 1 μM, which is still potent enough for the formation of actin bundles from actin filament prepared from ADP-actin monomer, but not actin filament from ATP-actin or AMPPNP-actin monomer (see Supplementary Fig. 5d, e), suggesting a molecular condensation dependent F-actin crosslinking. The previously noted specific affinity of Spa2 for ADP-F-actin suggests a strong cooperativity between Spa2 and highly homogeneous ADP-F-actin.

We next sought to understand the interplay between Spa2 activities in nucleating and crosslinking ADP-actin upon ES. Using TIRFM, we monitored real-time actin polymerization from ADP-G-actin in the presence of 5 μM Spa2(1-281), Spa2(281-535), and Spa2(1-535). Spa2(281-535) and Spa2(1-535) rapidly generated a large amount of ADP-actin filaments with initial nucleation and elongation within 3 min. After ~15 min in the presence of Spa2(1-535), but not Spa2(1-281) or Spa2(281-535), actin filaments started to exhibit striking rearrangement, converging, and bundling with a progressive increase in filament intensity over time (Fig. 5b, Supplementary Movie 16). Interestingly, Spa2(1-535)-induced actin filament bundles did not develop into pearling and spindle-shaped anisotropic actin filament droplets, which can be generated by actin crosslinkers, such as mammalian filamin[34] and phase-separated bacterial effector XopR[8]. Given the material characteristics of Spa2 condensates, Spa2(1-535) proteins stimulate cohesion among actin filaments in sync with their elongation. This concurrent process leads to the relaxation of F-actin bundles into an elongated form, avoiding the generation of uniform interfacial tension that could otherwise prompt bundle contraction or shrinkage. The Spa2(1-535)-crosslinked actin filaments prepared from ADP-actin monomer showed high resistance to cofilin-mediated severing (Fig. 5c, d, Supplementary Movie 17), likely due to the mechanical stabilization effect of actin bundling since Spa2 does not compete with Cof1-mediated severing on single filament (Fig. 3b). In contrast, none of the three shorter truncating Spa2 variants colocalized with or rearranged actin filament prepared from ATP-actin or ADP-P$_i$ actin monomer (Supplementary Fig. 5b, f and g, Supplementary Movie 18, 19). To understand the flexible interactions between Spa2 condensates on the F-actin surface, we next investigated the Spa2 wetting on actin filament by performing topography imaging of the Spa2-actin filament complex via a FastScan atomic force microscopy (AFM) system. The heights of filamentous structures with or without Spa2 were measured, allowing us to differentiate the coalescence of Spa2(1-535) condensation from the colloidal assembly of Spa2(281-535) oligomers. While the bare actin filament is 11.97 ± 8.1 nm in height (Fig. 5e, f), the height of Spa2(1-535) decorated actin filament was increased to 62.4 ± 30.8 nm (Fig. 5e–g). However, the actin filament height had no significant change after adding Spa2(281-535) and Spa2(1-281) (Fig. 5e–g). In addition, Spa2(1-535) is highly conserved in fungal species, with folded regions corresponding to Spa2(281-535) and the presence of an N-terminal flexible region (Supplementary Fig. 6). This suggests a conserved coexistence of both folded and unfolded components, which could potentially be a critical characteristic in the mechanism of ADP-actin remodeling.

To determine the importance of Spa2 valency for actin nucleation and crosslinking, we next created a multivalent Spa2(281-535) via protein engineering using a trimerization coiled-coil (CC) motif[35] (Spa2(281-535)-trimer, Supplementary Fig. 7a). The recombinant Spa2(281-535)-trimer exhibited higher-order assemblies than the CC-defined trimeric state (Supplementary Fig. 7b, c), likely due to mismatched intermolecular interactions in the combination of the trimeric CC and dimeric Spa2(281-535), which, nevertheless, mimics a IDR(aa 1-281)-mediated multivalent effect of Spa2(281-535). In the results, Spa2(281-535)-trimer showed potent ADP-actin monomer-specific nucleation activity, which was even higher than that of the

original Spa2(281-535) (Supplementary Fig. 7d–h, Supplementary Movie 20), suggesting an increased biochemical activity with enhanced oligomeric states. In addition, Spa2(281-535)-trimers are also capable of colocalizing and crosslinking actin filament performed by ADP-actin monomer with a slightly faster actin filament convergence than Spa2(1-535) overtime at the same protein concentration (Supplementary Fig. 7i, j, Supplementary Movie 21), suggesting that the ability to crosslink actin filament performed by ADP-actin monomer by Spa2 is dependent on the oligomeric states of the 281-535 aa region. In contrast, the Spa2(281-535)-trimer showed no ATP-actin-dependent activities, such as nucleation (Supplementary Fig. 7d, g, and h, Supplementary Movie 22), binding (Supplementary Fig. 7i), or bundling (Supplementary Fig. 7j, Supplementary Movie 23).

## Spa2 interacts preferentially with the ADP-nucleotide bound state depending on the actin D-loop conformation

We next sought to understand how Spa2 differentiates the D-loop conformation of actin in different nucleotide states. Phosphate release from ADP-P$_i$-actin to ADP-actin turns the DNase I binding loop (D-loop) into a closed conformation, an important difference between ATP- and ADP-actin filaments. Whereas phalloidin stabilizes the D-loop in a closed conformation, by inhibiting phosphate release, in contrast jasplakinolide (JASP) inhibits phosphate release, prevents dynamic conformational changes of the intrastrand interface, and locks actin filament in the open-D-loop state regardless of the nature of bound nucleotides[7,36–38] (Supplementary Fig. 7k). We found that JASP-stabilized actin abolished Spa2(1-535) function in crosslinking actin filament performed by ADP-actin monomer, whereas phalloidin did not (Fig. 6a). We additionally examined ES-triggered actin cable remodeling in different temperature-sensitive mutant alleles of ACT1, act1-125 (K50A, D51A), which has point mutations on D-loop (Supplementary Fig. 7k). Interestingly, act1-125 displayed clear attenuation in actin cable stabilization upon ES triggering, whereas act1-101 (D363A, E364A), with a mutation in actin subdomain I, was comparable to that in WT cells (Fig. 6b, c). Strikingly, while Spa2(1-535) in act1-101 showed a comparable pattern as WT yeast, with or without GS, Spa2(1-535) has completely lost filamentous structures in the act1-125, resulting in diffusive signals in the cytoplasm (Fig. 6d, e), indicating abolished ability in ADP-actin binding.

## Discussion

The dynamic and timely remodeling of the actin cytoskeleton during diverse signaling events is essential to coordinate physiological processes or adapt to pathological conditions. Changes in treadmilling and network reorganization of the actin cytoskeleton are orchestrated by many different ABPs in an actin nucleotide-dependent fashion. The nucleotide state is one of the major factors that can control actin conformation via allosteric communication between nucleotide binding sites and their spatially distinct regions[39]. Whereas the nucleotide state can directly modulate actin nucleation by creating a favorable conformation for the nucleator, nucleotide-specific binding ABPs also play critical roles in evoking the changes in actin conformation.

In the past, nucleation, elongation, and crosslinking were mostly studied only in the presence of ATP but remained unclear for ADP-actin, which is a dominant state during certain cell signaling events, such as energy starvation. Under normal growth conditions, actin filament treadmilling requires balanced polymerization and depolymerization, in which actin-depolymerizing factor/Cof1, cyclase-associated protein/Srv2, and twinfilin/Twf1 synergize during the depolymerization of aged actin filament to maintain the actin turnover rate because they have a higher affinity for ADP-actin than ATP-actin[40–43]. When the ATP drops upon glucose starvation in yeast, we found that both actin cable polymerization and bundling[1] are rapidly enhanced. Here, we show that Spa2 controls ES-triggered actin remodeling and is a multifunctional regulator that initiates actin monomer

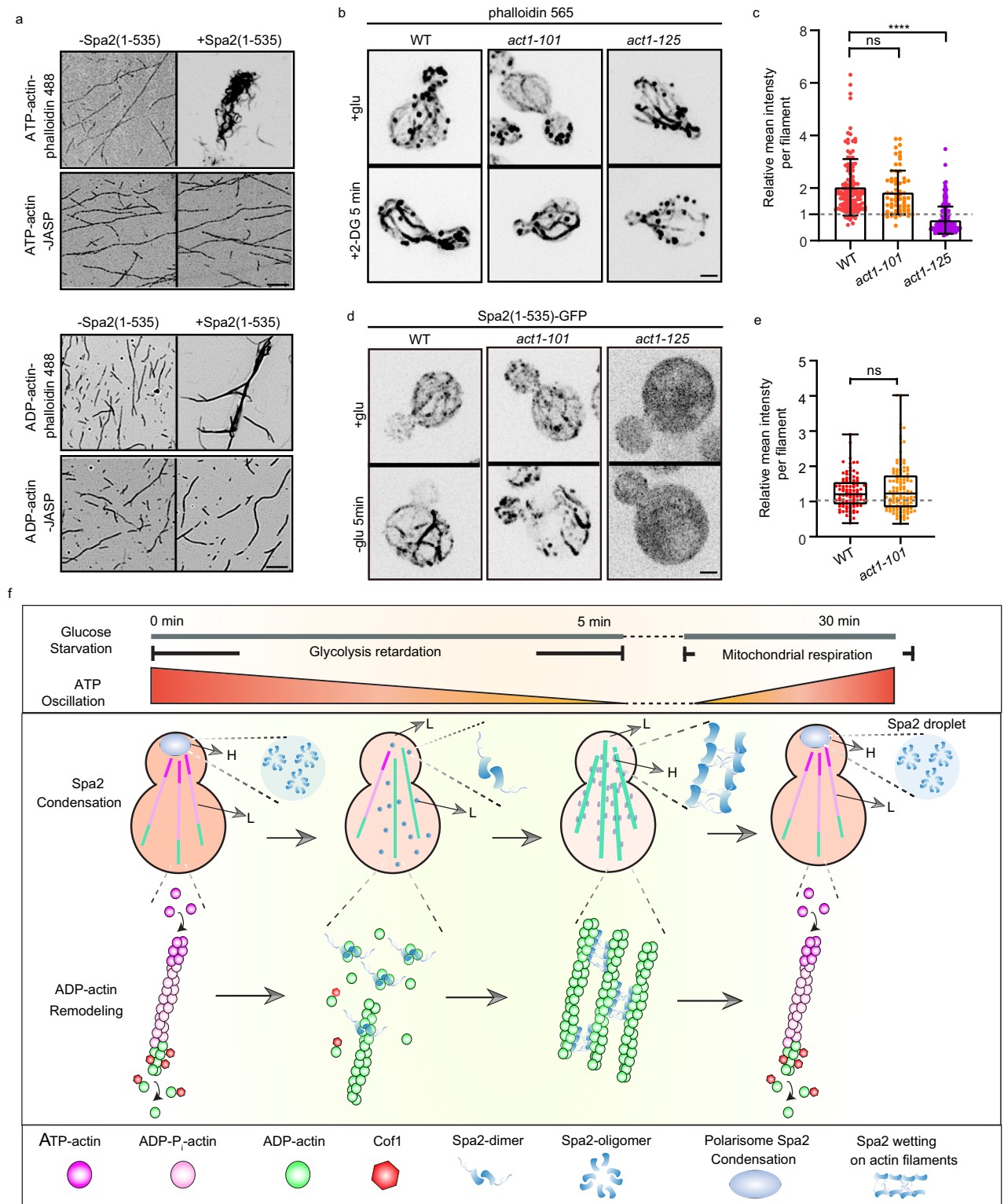

nucleation and triggers actin filament crosslinking in an ADP-actin-specific manner (Fig. 6f). While we currently lack direct in vitro evidence of Spa2's association with the barbed-end of ADP-actin, it is important for future studies to definitively demonstrate whether and how Spa2 acts as an ADP-actin nucleator with a tight barbed-end association. In contrast to ATP-actin nucleators that attract multiple G-actins to kick-start polymerization, Spa2 exhibits an affinity towards G-actin that is several orders of magnitude lower. For example, the affinities of Formin, Spire, and Ena/VASP for G-actin are 0.001 nM, 0.1 nM, and 3.2 nM, respectively[44,45]. This observed phenomenon of Spa2 might serve to establish a high threshold for its activation of ADP-actin polymerization, occurring only when a significant concentration recruitment occurs during ES. Spa2 is known as a macromolecular polarisome complex scaffolder located at the fungal bud tip for polarity establishment by engaging clients, including formin and formin nucleation-promoting factors, via cophase separation[14,15]. To exert efficient protection of actin cables to adapt to the abrupt change in energy, Spa2 exhibited striking multistep functions with high

**Fig. 6 | Multifaceted remodeling of ADP-actin by Spa2 in a D-loop conformation-dependent manner. a** Fluorescence micrographs of actin filament prepared from ATP -actin and ADP-actin monomer that were preincubated with phalloidin or jasplakinolide for 30 min at a 1:1 molar ratio before adding 10 µM Spa2(1-535). Fluorescence micrographs (**b**) and fluorescent signal quantification (**c**) of 565-phalloidin-stained actin cables in WT, *act1-125*, and *act1-101* after 5 min of 2-DG treatment. *n* = 127, 183, and 222 cables per treatment. Error bars, mean ± S.D. Fluorescence micrographs (**d**) and fluorescent signal quantification (**e**) of Spa2(1-535)-GFP in WT, *act1-101* and *act1-125* after 5 min of glucose starvation treatment. n = 102 and 121 cables per treatment for **e**. Error bars, mean ± S.D. Whiskers represent min to max. The relative mean intensity per filament in **c** and **e** were normalized by using the absence of glucose against the presence of glucose. The dashed line indicates 1. *P*-values were calculated using GraphPad Prism 6. Significance between two sets of data were determined by the one-way analysis of variance (****$p < 0.0001$, and ns = not significant). Scale bars are 2 µm in **b**, **d**. **f** Working model of ADP-actin remodeling by Spa2 during glucose starvation. Schematic illustration of the molecular mechanisms by which Spa2 progressively remodels ADP-actin filament polymerization and network formation during GS. Upon rapid GS, the ATP concentration drops significantly within 5 min via inhibiting glycolysis. Spa2 proteins gradually relocate from macromolecular polarisome condensates into the cytoplasm. Spa2 first nucleates ADP-actin monomer, next associates with and wets on ADP-actin filaments, then crosslinks ADP actin filaments. Within 30 min of GS, once respiration replenishes ATP, Spa2 proteins are disassociated from ADP-actin filaments and recruited back to the polarisome, restoring ADP-actin bundles to ATP-actin filaments. L means low Spa2 protein condensation. H means high Spa2 protein condensation. fSource data is provided as Source Data file.

ADP-actin specificity that ensured rapid polymerization and crosslinking of ADP-actin filament after the ATP decrease. Spa2 proteins are rapidly released from the cell cortex upon ES and start to nucleate and elongate ADP-actin monomer, which is driven by the activities of the homodimeric portion of Spa2 (residues 281-535). The specific and highly potent polymerization of ADP-actin monomer by Spa2, starting from a nanomolar concentration, allows rapid generation of ADP-actin filament by utilizing cytosolic ADP-actin monomer, including those from Cof1-mediated depolymerization (Fig. 3), without disrupting filamentous cable structures. Simultaneously, Spa2 from the polarized bud tip is recruited to ADP-actin filament over time and forms actin cable-associated Spa2 condensates in a concentration-dependent manner, depending on the multivalency of IDR (aa 1-281) and specific ADP-actin binding by aa 281-535. Under such physiological stress, the Spa2 translocation from the cell cortex to ADP-actin filament reestablishes Spa2's equilibrium between the polarisome complex at the bud tip and the cohesion- and cocondensation-formed assemblies on ADP-actin filament. The multivalent assembly of Spa2 facilitates its tight adhesion and wetting onto the surface of ADP-actin filaments. This process is driven by coalescence associated with the filament surface and the capillary effect, which progressively crosslinks neighboring filaments (Figs. 4, 5). Together, our work suggest that the actin cable bundles triggered by glucose starvation (GS) result from a combination of rapidly de novo polymerized ADP-F-actin, which requires the cooperative effort of nanomolar concentrations of Spa2 and Cof1-depolymerization, as well as the crosslinking of pre-existing aging F-actin via on-site accumulation and condensation of Spa2. These processes eventually lead to resistance to cofilin through bundling in a feedback regulation manner. Notably, the C-terminus of Spa2(1-535) (amino acids 536-1466) is likely to regulate the material properties of Spa2 condensates on the actin filament surface but does not diminish Spa2's ability to associate with and crosslink ADP-actin filament. Spa2-GFP was observed as puncta along actin cables compared to relatively smoother wetting by Spa2(1-535)-GFP in vivo upon ES.

Spa2 exhibited the ability to specifically bind actin filament prepared from ADP- actin monomer and sense the phosphate moiety, which is reminiscent of the actin depolymerization factor cofilin[46]. However, the multilayer control of ADP-actin polymerization and crosslinking by Spa2 is largely different from the known depolymerization mechanism of ADP-actin filament. While the D-loop in G-actin is flexible, regardless of the nucleotide state, the ADP-actin filament D-loop exhibited an interface in the closed state[7,32,47]. The sensing of the nucleotide state by Spa2 with specificity for ADP, rather than ATP and ADP-P$_i$, suggests the importance of C-terminal and D-loop conformations. Actin is a plastic molecule that adopts different conformations in a nucleotide-dependent manner[39]. Conformational changes are enabled by allosteric communication between the nucleotide-binding site and subdomain 2, which includes the sensing loop[48] and D-loop[7]. ADP-actin might permit a wedging of Spa2 between

the C-terminus and D-loop, whereas the intrastrand interface between the C-terminus and opened D-loop[7,49] would inhibit Spa2 binding. Instead of competing with Cof1 on ADP-actin, Spa2 cooperates with Cof1 for ADP-actin remodeling. Our findings indicate that this coordination likely involves a few biochemical activities of Spa2, including ADP-G actin nucleation using depolymerized actin and a Spa2-assisted depolymerization through Cof1, where IDR of Spa2 is required. Investigating the impact of this conformationally flexible Spa2 IDR on Cof1-mediated severing during their association with F-actin is a topic that warrants future research and necessitates high-resolution structural approaches[50] for further insights. Compared to Cof1's high affinity toward ADP-actin, Spa2 has a lower affinity ($K$d = ~0.21 µM for Spa2(281-535), $K$d = ~0.52 µM for Spa2(1-535), and $K$d = 0.02–0.1 µM for Cof1[51–53]). The TIRF assay did not reveal direct severing of F-actin by Spa2. To address challenges in controlling precise starting times in seconds for comparing different samples and the differential removal of short filaments with different lengths from the TIRF field due to liquid diffusion, the assessment of Spa2-mediated severing utilized pre-assembled long F-actin instead of comparing the length of initial formed short filaments during the early polymerization from G-actin. Moreover, Cof1 is still able to effectively sever ADP-actin at a nanomolar concentration in the presence of Spa2. Instead, Spa2 efficiently utilizes the cofilin-released ADP-actin monomer for de novo polymerization into ADP-actin filament and also turns them into bundles. Phalloidin, which competes with cofilin on the same binding pocket[54], does not inhibit Spa2 function. However, if the D-loop is locked into an open conformation, such as by JASP, Spa2-mediated actin filament crosslinking is diminished (Fig. 6). Nevertheless, we are unable to exclude two potentially important properties of actin filament, flexibility and the conformational change of filaments, which might also contribute to Spa2's specificity for ADP-actin. ADP-actin filament has greater flexibility than ATP- or ADP-P$_i$-actin filament[55,56], whereas the conformational change of actin filament also impacts the binding of multiple ABPs, such as cofilin[50], coronin[49], and calponin homology domain[57], to actin filament in different nucleotide-bound states. In addition, mechanical bending forces modify the structural landscape of actin filament[58], which might cooperate with nucleotide-specific binding by ABPs.

It is important to note that while Spa2's differing responses to AMPNP- and ADP-actin filaments highlight its nucleotide specificity in vitro, the use of F-actin prepared from ATP-actin monomers raises an unresolved question about the necessity for high-purity ADP-actin in cross-linking. Currently, we also lack the approach to specifically identify the nucleotide species on F-actin in living cells upon rapid glucose starvation, such as within 2 minutes. It remains a possibility that ATP hydrolysis and Pi release on F-actin occur more efficiently in vivo compared to in vitro conditions. Alternatively, other proteins might aid Spa2 in achieving a strong association with F-actin in vivo, thereby reducing the necessity for extremely high purity of ADP-actin species as observed in vitro. Together, the multiple functions of Spa2

offer distinct features to sense environmental stress and provide rapid actin remodeling. ADP-actin filament engagement with Spa2 does not require prior depolymerization or de novo ADP-actin filament polymerization. Upon energy starvation, the multivalency and condensation of Spa2 permit a rapid cellular adaptive response to utilize ADP-actin monomer in situ to crosslink actin filament with a high content of ADP-species. Such Spa2-dependent mediated actin cytoskeleton remodeling under stress conditions might facilitate the transition from ES to recovery when energy is in supply by having and rapidly regenerate dynamic intact F-actin for cellular processes.

## Methods

### Yeast culture and plasmids

Yeast strains were grown in standard-rich media or complete synthetic media as previously described[14,59]. Genomic C-terminal fluorescent fusion tagging at the endogenous locus was performed using the lithium acetate-based method by transforming PCR-fragments derived from pFA6a-GFP(S65T)-His3MX6[14,59,60]. Yeast strains expressing Spa2-truncating variants were generated from the integrated vector pRS306 containing Spa2 genomic +500 bp promoter and DNA fragment 843-1605 bp. Transformants were selected on the YPD agar plates containing 200 μg/ml G418 or complete synthetic media supplemented with appropriate amino acids. The yeast strains and oligonucleotide primers used are indicated in Supplementary Tables 1 and 2.

### Live-cell fluorescence imaging

Yeast strains were cultured overnight at 30 °C or 25 °C in the synthetic complete medium (SM) without tryptophan and reinoculated into fresh medium the following day to a final $OD_{600} = 0.15$ for an additional 3 h of culture before imaging. Cells were immobilized onto concanavalin A (ConA, 1 mg/ml)-coated coverslips and imaged by spinning disc confocal (SDC) microscopy coupled with structured illumination microscopy (SDC-SIM). The imaging was performed by a Nikon Ti2 inverted microscope equipped with a Yokogawa CSU-W1 confocal spinning head, a Plan-Apo objective (100 × 1.45-NA), a back-illuminated sCMOS camera (Prime95B; Teledyne Photometrics), and a super-resolution module (Live-SR; GATACA Systems). All image acquisition and processing were controlled by MetaMorph (Molecular Device) software. The images were acquired continuously at a 0.25-μm interval for a total range of 7.5 μm in the z-direction, using 200 ms exposure for actin cable and 100 ms for Spa2.

### Energy starvation treatment

To examine in vivo Spa2 and actin cable remodeling upon ES, early log-phase cells were treated by either replacing SMD with SM without glucose or adding 20 mM 2-DG (inhibition of glycolysis) and 10 μM antimycin A (inhibition of mitochondrial ATP production).

### Queen-expressing strains, imaging, and analysis

The queen plasmid used in this study was purchased from NBRP Yeast Genetic Resource Center (NBRP/YGRC)[12]. Queen-expressing budding yeast strains were generated by linearizing the constructs with PstI and then inserting them into the $his3\Delta1$ locus of the YMY2012 and glucose-relevant mutant strains $hxk2\Delta$, $gpr1\Delta$, $snf3\Delta rgt2\Delta$. Budding yeast cells were immobilized on a glass cover slip-coated with concanavalin A at 1 mg/ml before imaging. The immobilized cells were imaged by a Nikon Ti2 SDC microscope equipped with a Yokogawa CSU-W1 confocal spinning head, a Plan-Apo objective (100×1.45-NA), and a back-illuminated sCMOS camera (Prime95B; Teledyne Photometrics). Imaging lasers were provided by 405 nm/100 mW (Vortran), 488 nm/150 mW (Vortran), combined in a laser launch (iLaunch, GATACA Systems). The QUEEN ratio was calculated using Fiji software[61] and was calculated as follows. First, both the QUEEN images were converted to signed 16-bit floating-pointed grayscale. The QUEEN signals were subjected to background subtraction using the mean intensity per pixel from the signals outside the cells. Then, the intensity of ex405 images was divided by ex480 images to calculate the QUEEN ratio at each pixel, as the intensity ratio to represent the cellular ATP concentration. The normalized queen ratio represents the value in the presence of ES divided to in the absence of ES.

### Total internal reflection fluorescence microscopy

For TIRFM experiments, 25 × 50-mm coverslips (Marienfeld Superior) were cleaned with 20% sulfuric acid overnight and rinsed thoroughly with sterile water. The coverslips were then coated with 2 μg/ml biotin-PEG-silane (Laysan Bio Inc.) in 80% ethanol (pH 2.0, adjusted by HCl) and 2 mg/ml methoxy-PEG-silane at 70 °C overnight. The next day, the coverslips were rinsed thoroughly with sterile water, dried in a nitrogen stream, and kept at −80 °C before use. The functionalized coverslip was attached to a plastic flow cell chamber (Ibidi, sticky-Slide VI 0.4), followed by a 30 s incubation with HEKBSA buffer (20 mM HEPES, pH 7.5, 1 mM EDTA, 50 mM KCl, and 1% bovine serum albumin) and then 60 s incubation with 0.1 mg/ml streptavidin in HEKG10 (20 mM HEPES, pH 7.5, 1 mM EDTA, 50 mM KCl, 10% [vol/vol] glycerol). Then, the flow cell chamber was washed with TIRF buffer (10 mM imidazole, 50 mM DTT, 15 mM glucose, 50 mM KCl, 1 mM MgCl2, 1 mM EGTA, 100 μg/ml glucose oxidase, 40 μg/ml catalase and 0.5% methylcellulose [4000 cP], 0.3 mM ADP or ATP, pH 7.4). Recombinant proteins prepared in TIRF buffer for actin nucleation studies were mixed with 0.8 μM and 3 μM ADP-actin monomer, 0.5 μM ATP-actin, and 3 μM ADP-P$_i$ monomer (10% Oregon Green 488 labeled, 0.5% biotin-labeled) before flowing into the chamber. Time-lapse images were acquired at room temperature at 5-s intervals for 10 min when measuring nucleation efficiency (Fig. 3d; Supplementary Fig 4g, l, o; Supplementary Fig 7d) and 90 s intervals for 60 min (Fig. 5b; Supplementary Fig. 5f, g; Supplementary Fig 7j) when evaluating the crosslinking effect with the TIRF module on the SDC mentioned above (iLasV2 Ring TIRF, GATACA Systems). The specific F-actin preparation conditions for crosslinking assays are described in the following section. For actin elongation rate quantification, the fast elongation end of each individual filament was traced by hand for a period of 2 min each. We used the conversion factor of 370 subunits per micrometer of actin filament to calculate the elongation rate. For Fig. 5c Cof1-mediated ADP-actin filament depolymerization assay, images were deconvolved by Huygens Essential deconvolution software (Scientific Volume Imaging, Netherlands) to diminish background signal. To determine the severing rates of Cof1 with various Spa2 variants for Supplementary Figs. 4d and e, we initially recorded the filament lengths before protein addition. Next, we manually recorded the total number of severing events for each filament during the first 80 seconds post-protein introduction. For each protein combination, we computed the cumulative number of severing events from 20 to 60 seconds, then divided this by the filament's initial length to derive the values for the Y-axis in Supplementary Fig. 4d. Finally, we calculated the slope of the Y-axis against the X-axis to plot the individual points in Supplementary Fig. 4e.

### Nonpolymerizable yeast actin expression and purification

Nonpolymerizable (NP)-actin protein was purified from insect cells as previously reported, in which three point-mutations at the barbed-end of *Drosophila melanogaster* 5 C actin (D287A, V288A, D289A) were introduced[62]. Using a similar strategy, we generated nonpolymerizable *S. cerevisiae* Act1 actin via mutagenesis of the homologous residues (D286A, V287A, D288A). Ligation-independent cloning was used to insert the PCR-amplified gene containing *S. cerevisiae* NP-actin into the pFB-LIC-Bse cloning vector that includes an L21 enhancer sequence to drive gene expression. The cloned plasmid was then transformed into MAX Efficiency™ DH10Bac *E. coli* cells (ThermoFisher), and successful transformants were selected using Blue-White screening. Bacmid extracted from DH10Bac was transfected into Sf9 cells using Cellfectin® II Reagent from ThermoFisher. Baculovirus generated from the

transfection was further amplified for two passages, and the resulting P2 virus was used to infect Sf9 cells at 27 °C for 72 h. The Sf9 cells were resuspended in a lysis buffer containing 20 mM Tris (pH 8.0), 50 mM KCl, 0.1 mM $CaCl_2$, 0.2 mM ATP, and 2 mM β-mercaptoethanol and disrupted using SONICS Vibra-Cell™ ultrasonic liquid processor. The lysate was clarified via centrifugation and filtration. Purification of NP-actin was performed at 4 °C by passing the crude lysate through a nickel-nitrilotriacetic acid affinity column (GE Healthcare) and eluted with a gradient increase of imidazole from 20 to 500 mM. Additional purification was performed by gel filtration (HiLoad Superdex 200 16/60, GE Healthcare) using the same lysis buffer. The purified NP-actin was concentrated and aliquoted to 5–10 μl each tube and frozen in liquid $N_2$.

### Actin anisotropy assay
For measuring the binding affinity towards actin monomer, fluorescent proteins were diluted to 60 nM (fluorescein Alexa 488-labeled Spa2(1-281), (281-535), (1-535)) in 20 mM HEPES 500 mM NaCl buffer. The stock of nonpolymerizable monomeric actin that can not form actin filament was titrated down starting from 40 μM, from which dilutions were further made. Equal volumes of Alexa 488-Spa2 and actin monomer were incubated for at least 2 h in the dark at 25 °C. Alexa 488 fluorescence (485/510 nm) was measured by the plate reader Cytation 5 (BioTek, USA). The final curve was derived from the subtraction of Alexa 488-Spa2. The total volume was 25 μl containing 60 nM Spa2 protein.

### In vivo actin cable imaging and image analysis
To compare actin cable intensity, the immobilized cells were imaged by the SDC-SIM (Live-SR; GATACA Systems) based on structured illumination with optical reassignment and image processing improvement[63]. The method, known as multifocal structured illumination microscopy[64], allows for combining the doubling of the resolution with the optical sectioning capability of confocal microscopy. The maximum resolution is 128 nm with a pixel size in SR mode of 64 nm. Excitation light was provided by 488 nm/150 mW (Vortran) (for GFP), 561 nm/100 mW (Coherent) (for mCherry/mRFP/tagRFP) and all image acquisition and processing were controlled by MetaMorph (Molecular Device) software. The mean fluorescence signal intensity per filament and total fluorescence actin cable intensity per cell was measured as previously described[65]. Yeast cells were cropped with background subtraction from average projections in Fiji. Individual observed actin cables in average projections of mother cells were traced in Fiji using a line that encompassed the entire width of the cable. The total intensity per cell was measured by multiplying the mean intensity per filament with the actin filament length. The relative mean intensity per filament and relative total intensity per cell after ES were normalized to the glucose control.

### In vivo concentration measurements for Spa2 variant proteins
To quantify the in vivo protein concentration of Spa2 variant proteins, three reference strains expressing Syp1-GFP, Crn1-GFP, and Sec3-GFP were used. The cytosolic protein concentration of Syp1, Crn1, and Sec3 were previously determined by FCCS[66,67]. Sum intensity Z-projection images of Spa2 variant protein tagged with the same GFP tag as reference proteins were used for intensity analysis. The total cytosol signal intensity of reference and Spa2 variant proteins were measured from the $10 \times 10$ pixel box in the cytoplasm (Ic). The background signal intensity and auto-fluorescence signal were measured of $10 \times 10$ pixels areas from areas outside of cell (Ib). A signal intensity standard curve and equation were derived from the reported cytosolic concentration of reference proteins, Syp1, Crn1, and Sec3, and the measured value of Ic-Ib (Supplementary Fig 3f). The cytosol concentrations of Spa2 variant proteins were calculated based on the above equation. The protein concentration at the tip region, at the filament and puncta was

derived from the ratio of signal intensity (R). R was measured by the equation $R = (T−B)/(C−B)$, $R = (F − B)/(C−B)$ or $R = (P−B)/(C− B)$ using the signal intensity in the background (B), bud tip (T), filament (F), puncta (P) and cytosol (C) region. More than 20 medium-size budded cells were measured by ImageJ using a $3 \times 35$ pixel as ROI for bud tip intensity quantification, a $3 \times 10$ pixel as ROI for filament intensity quantification, and a $6 \times 35$ pixel as ROI for puncta intensity quantification.

### Surface plasmon resonance (SPR)
The SPR experiment was performed at room temperature in a buffer containing 20 mM HEPES and 50 mM NaCl pH 7.4 using a Biacore T200 instrument (GE Healthcare). Recombinant Spa2 variants were immobilized on the CM5 chip (GE Healthcare) by amine coupling. The carboxyl group on the dextran surface of the chip was converted to an amine-reactive ester by reacting with 0.2 M 1-ethyl-3-(3-dimethylpropyl)-carbodiimide and 0.1 M N-hydroxysuccinimide. The molecules flowed over the sensing surface by injection at a flow rate of 10 ml/min at pH 4.5, while the reference cell was left blank without the injected protein. To test the binding of analyte Spa2 with the immobilized ligand Spa2 protein variants, serially diluted (1:1) Spa2 variant was flowed over the surface of the control and ligand for 60 s and dissociated with buffer (20 mM HEPES, 50 mM NaCl, pH 7.4) for 150 s at a rate of 30 μl/min. Spa2 was injected at gradient concentrations of 20, 10, 5, 2.5, 1.25, 0.625, 0.3125, and 0.15625 μM. The chip surface with left-over protein captured was regenerated by treatment with 50 mM NaOH for 3 s at 100 μl/min after each cycle. The sensorgram for the binding experiment was normalized with the reference cell and fitted to the bivalent analyte model using Biacore T200 Evaluation software (GE Healthcare). The $K_{on}$ represents the association constant, while the $K_{off}$ stands for the dissociation constant, the equilibrium constant KD1 was derived by $K_{off1}/K_{on1}$.

### Recombinant protein expression and purification
Spa2(1-281), Spa2(281-535), Spa2(1-535), and Spa2(281-535)-trimer recombinant proteins were expressed and purified from *Escherichia coli* (BL21(DE3) Rosetta T1R). Cells were cultured in 2 L TB medium (24 g yeast extract per liter, 20 g tryptone per liter, 4 ml glycerol per liter, phosphate buffer pH 7.4) containing 50 μg/ml of kanamycin at 37 °C to an $OD_{600}$ of 0.6 before induction by 0.5 mM IPTG at 16 °C overnight. The cells were harvested by centrifugation at 4 °C and $5000 \times g$ (rotor JA10) for 15 min. The pellet was resuspended in 20 mM HEPES, pH 7.4 and 500 mM NaCl, 20 mM imidazole, 1 mM PMSF, protease inhibitor cocktail Set III, EDTA free from Thermo Fisher, and lysed by an LM20 microfluidizer (20000 psi). The lysate was clarified by centrifugation at $25,000 \times g$ for 1 h using rotor JA25.5 (Beckman Coulter). The supernatant was purified by an FPLC AKTAxpress system (GE Healthcare) using Ni-NTA affinity chromatography. The elution fractions containing targeted proteins from gradient elution with increasing imidazole concentrations were pooled and further purified by size-exclusion chromatography using a HiLoad 16/600 Superdex75 column (GE Healthcare) in 20 mM HEPES and 500 mM NaCl. Proteins were flash-frozen in liquid N2 prior to storage at −80 °C in small aliquots.

To obtain monomeric $Ca^{2+}$-ATP-actin with or without labeling with Oregon Green™ 488 iodoacetamide (ThermoFisher) or NHS-dPEG®4-biotin (Sigma), 5 g of rabbit skeletal muscle acetone powder (Pel-Freez Biologicals) was dissolved in 60 ml of G-buffer (5 mM Tris, pH 8.0, 0.2 mM ATP, 0.1 mM $CaCl_2$, 0.5 mM dithiothreitol) for 30 min at 4 °C. The mixture was filtered through cheesecloth three times to collect the actin-rich extracts in the supernatant. The filtration was repeated three times, and the actin-rich extracts were combined and subjected to centrifugation at $18,000 \times g$ by rotor JA-25.5 (Beckman). Afterward, the clear actin-rich supernatant was supplemented with 50 mM KCl and 2 mM $MgCl_2$ to allow actin polymerization with slow stirring at 4 °C for

1 h. To remove actin filament binding proteins, KCl powder was added slowly until a final concentration of 0.8 M and was subsequently stirred slowly for an additional 30 min. The solution was subjected to 95,800× g centrifugation for 3 h with rotor Ti45 (Beckman) at 4 °C to collect the polymerized actin filament. The actin filament pellet was rinsed with 1 ml G-buffer, all the actin filament pellets were transferred to the 10 ml homogenizer with 5 ml G-buffer, and the pellet was homogenized by moving up and down. To further depolymerize the actin filament, we applied four cycles of sonication with 3 s-on and 10 s-off. Actin filament was then dialyzed against 1 liter of G-buffer without DTT overnight. The next day, we changed to 1 liter of new G-buffer without DTT and kept on dialysis. In the meanwhile, we freshly prepared Oregon Green™ 488 iodoacetamide or NHS-dPEG®4-biotin with high-quality dimethylformamide into a final concentration of 10 mM. The concentration of dialyzed actin was determined by $OD_{290}$ using NanoDrop. Before proceeding to actin labeling, we diluted the actin with cold 2× labeling buffer (50 mM imidazole pH 7.5, 200 mM KCl, 0.3 mM ATP, 4 mM $MgCl_2$) to an actin concentration of 23 µM. A 12–15-fold molar excess of Oregon Green™ 488 iodoacetamide/NHS-dPEG®4-biotin stock was added dropwise with very gentle vortex. To allow sufficient labeling, we kept the solution in the dark with aluminum foil and rotated at 4 °C overnight. The next morning, we pelleted the labeled actin with Type 50.2 rotor (Beckman) at 111,000× g for 3 h. We then collected all the pellets and transferred them to the homogenizer with 5 ml G-buffer. Afterward, the actin was homogenized for 20 times and sonicated with 4 cycles of 3 s-on and 10 s-off. To further depolymerize the actin, it was dialyzed against 1 liter of G-buffer at least 2 times (24–36 h). We then applied the dialyzed actin to centrifugation at 167,000× g for 2.5 h with rotor SW55 Ti (Beckman) at 4 °C. To further purify actin, the top 2/3 of the supernatant was collected and injected into a GE AKTA FPLC and separated through column HiPrep™ 16/60 Sephacryl™ S-300 HR column. The collected labeled actin was dialyzed overnight against G-buffer with 50% glycerol to reduce the total volume. Small aliquots of actin were prepared and flash-frozen in liquid N2 for long-term storage.

## In vitro protein condensation assay and imaging

Recombinant Spa2 variants (10% Alexa 488-labeled) were incubated for 5 min at room temperature before applying 5 µl on the coverslip and being imaged by SDC-SIM. After protein purification, a serial dilution of protein concentration and ionic strength was performed, starting from the high concentration of 500 mM NaCl.

## Analytical ultracentrifugation (AUC)

Analytical ultracentrifugation sedimentation velocity (AUC-SV) data were collected using a Beckman ProteomeLab XL-I analytical ultracentrifuge using an 8-hole An-50 Ti analytical rotor. Protein samples were dialyzed overnight in 20 mM HEPES (pH 7.4) and 50 mM NaCl and loaded into 2-sector AUC cells fitted with 1.2 cm epon centerpieces and quartz windows. The samples were centrifuged at 163,004 × g at 20 °C and the radial absorbance profile at 280 nm was recorded every 5-10 minutes for 15 h. The data were analyzed using SEDFIT using c(s) and c(s, ff0) size distribution models[68] and plotted with GUSSI[69]. Sedimentation coefficients were standardized to $s_{20,w}$ using the partial specific volume of the proteins (calculated using SEDFIT) and the density and viscosity of the buffer (calculated using SEDNTERP[70]).

## ADP-actin monomer preparation

Every 100 µl ATP actin monomer was added 1 µl 100×ME (50 µM $MgCl_2$ and 250 µM EGTA) and incubated for 30 min on ice. Then 1 µl glucose (1 mM) and 5.4 µl hexokinase (0.02 units/µl) were added and mixed for 3 h at a cold room. The samples were centrifuged at 80 K× g for 55 min at 4 °C (TLA, Ultra Centrifuge Rotor from Beckman), and the 80% supernatant was obtained and measured the concentration ($A_{290}$ = 0.0266 µM/cm).

## AMPPNP-actin monomer preparation

To prepare AMPPNP-G-actin, we started by polymerizing ATP-G-actin to F-actin within ADP-F-buffer, which consisted of 0.2 M KCl, 20 mM imidazole (pH 7.0), 2 mM $MgCl_2$, 0.4 mM EGTA, 0.2 mM ADP, and 2 mM DTT. The resulting F-actin was then collected into a pellet and reconstituted in the AMPPNP-G-buffer. This buffer was made up of 2 mM Tris-HCl (pH 8.0), 0.2 mM $CaCl_2$, 0.2 mM AMPPNP, and 1 mM DTT. After re-suspension, we dialyzed the mixture against the same buffer at 4 °C for overnight (around 12 h), and this dialysis repeated two times. Post dialysis, the sample underwent centrifugation to clarify it, and the resulting supernatant was used as the AMPPNP-G-actin.

## ADP-$P_i$-actin filament preparation

In our total internal reflection microscopy experiments, ADP-Pi-actin filaments were created using Spa2 and ADP-actin monomers. These experiments utilized a TIRF buffer with an additional 60 mM potassium phosphate. For the crosslinking assays, we prepared pre-assembled ADP-Pi-actin filaments by combining ADP-actin monomers with our ADP-Pi-F buffer. This buffer consists of 2 mM Tris-HCl (pH 8.0), 0.2 mM $CaCl_2$, 0.2 mM ADP, 1 mM DTT, 50 mM KCl, 1 mM $MgCl_2$, 1 mM EGTA, and 60 mM potassium phosphate.

## In vitro actin polymerization and imaging for F-actin crosslinking assay

First, a 10 µM actin monomer $Ca^{2+}$-G-actin prepared in a G-buffer was converted to $Mg^{2+}$-G-actin on ice for 5 min. In Fig. 5a, Supplementary Fig. 5b, c, d and Supplementary Fig. 7i, ATP-, ADP- and AMPPNP-F-actin were prepared with the following conditions. For ADP-F-actin preparation, 5 µM ADP-F-actin was polymerized by adding 10× KME (500 mM KCl, 10 mM $MgCl_2$, and 10 mM EGTA) into the prepared actin monomer for 30 min. 3 µM ADP-F-actin was used to incubate with Spa2 before being diluted to 0.05 µM for TIRF imaging. For actin filament prepared from ATP-actin monomer, 2 µM and 5 µM ATP-actin monomer were respectively used to generate 0.5 µM F-actin (30 min incubation) and 3 µM F-actin (2 min incubation) by adding 10× KME before Spa2 incubation. For AMPPNP-F-actin preparation, 5 µM AMPPNP-actin monomer was polymerized by adding 10× KME (500 mM KCl, 10 mM $MgCl_2$, and 10 mM EGTA) into the prepared actin monomer for 5 min at room temperature. 3 µM AMPPNP-F-actin was used to incubate with Spa2 before being diluted to 0.05 µM for TIRF imaging. All above F-actin were incubated with Spa2 for 30 min, which were subsequently incubated with ActinStain ATTO 565 (Hypermol) for 5 min, diluted to 0.05 µM with F-Buffer (G-buffer plus 1× KME), and applied to polylysine (0.01%)-coated coverglass for microscopic imaging using a 100× oil objective lens. For Fig.6a, actin filament prepared from 0.5 µM ATP-actin monomer and 3 µM ADP-actin monomer were prepared by overnight room temperature polymerization before a 30 min incubation with acti-stain 488-phalloidin (Cytoskeleton, Inc.) or polymerized directly for overnight in the presence of 10% Oregon green 488-actin, prepared F-actin was incubated with JASP at 1:1 molar ratio for 30 min before incubating with 10 µM Spa2(1-535) for 30 min, which was subsequently diluted to 0.05 µM with F-Buffer and applied to polylysine (0.01%)-coated coverslip for microscopic imaging using a 100× oil objective lens.

## Pyrene-actin assembly

Pyrene-labeled actin was purchased from Cytoskeleton Inc. A 20 µM G-actin (15% pyrene actin) was first converted to $Mg^{2+}$-ADP-actin for 5 min on ice before use and then mixed rapidly with various proteins in the G buffer. The spontaneous actin polymerization was initiated by 10 × KME buffer mix (10 mM $MgCl_2$, 10 mM EGTA, and 500 mM KCl), at a total reaction volume of 120 µl. The pyrene-actin fluorescence signal was monitored in a plate reader Cytation 5 (BioTek, USA) at excitation and emission wavelengths of 365 and 407 nm, respectively. The values

shown in the graph were the average data from three times of independent experiments.

## Actin critical concentration

Actin, labeled with pyrene (10%), was polymerized either in the presence of 5uM Spa2 variants or without them. This was done by adding an F-buffer, diluting the solution to various concentrations, and allowing it to incubate overnight at room temperature. Fluorescence measurements were then obtained from triplicate samples using a Cytation 5 fluorescence plate reader (BioTek, USA) at excitation and emission wavelengths of 365 and 407 nm, respectively. The results were interpreted using a two-line equation: $y = y_{cross} + (x - x_{cross}) \times k$. Where $x_{cross}$ and $y_{cross}$ are the shared x and y coordinates for the two lines at their intersection and k represents the slope.

## Atomic force microscopy (AFM)

4 μM ADP-Actin was polymerized first in F buffer (1 X KME in 1X G buffer) for 1 hour. Actin filaments were diluted 5 times in 30 μL F buffer with 100 nM proteins and incubated for 30 minutes.

For topography imaging of actin filaments, 0.8 μM of actin filaments in 30 μL F buffer containing indicated protein was deposited on 0.01% poly-L-lysine coated glass coverslips and incubated for extra 30 minutes before image. The coverslips were rinsed three times in F buffer to remove unattached actin filaments and 30 μl of F buffer was added to the sample immediately to prevent drying. A silicon nitride probe (Scanasyst Fluid+, Bruker) with nominal tip radius of 2 nm was used. All samples were imaged with a Bruker Dimension FastScan AFM system and Nanoscope V controller operating under ScanAsyst fluid mode. AFM images were acquired at a maximum scan rate of 2.5 Hz and 256 × 256 data points per image. Raw AFM data were analyzed using Bruker Nanoscope Analysis 1.90 software.

## Protein sequence prediction and conservation analysis

To identify the Spa2(1-535) homologs and perform conservation analysis, the query Spa2(1-535) sequence was submitted to FungiDB (https://fungidb.org/fungidb/) with default parameters and the NCBI. The fungal species containing Spa2 homolog with E-value < e-20 were chosen for further analysis. Their corresponding sequence alignment was performed with the online server Clustal Omega (https://www.ebi.ac.uk/Tools/msa/clustalo/), and the phylogenetic tree was generated by the interactive tree of life (http://itol.embl.de/). ANCHOR and the PHYRE2 protein fold recognition server (http://www.sbg.bio.ic.ac.uk/~phyre/) were used, respectively, in identifying the unstructured and structured domains. Structures of the ScSpa2(1-535) homologs in other fungi species were predicted by Alphafold2[71].

## Statistical analysis

All statistical analyses were performed in GraphPad Prism 6. p values were determined by two-tailed Student's t-test assuming equal variances and the one-way analysis of variance (*$p < 0.05$, **$p < 0.01$, ***$p < 0.001$, ****$p < 0.0001$, and ns = not significant). Error bars indicate the standard deviation (S.D.).

## Reporting summary

Further information on research design is available in the Nature Portfolio Reporting Summary linked to this article.

## Data availability

The datasets generated and analysed during the current study are available in the figshare repository (https://figshare.com/s/f1d73eb1ff79bdd92615). Source data are provided with this paper.

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

## Acknowledgements

We are grateful to Dr. Steven Chou (University of Connecticut, USA) for his critical reading of the manuscript. We thank the NTU Protein Production Platform (www.proteins.sg) for several initial protein expression and purification trials. We also thank Dr. Yonggui Gao (NTU, Singapore) for NP-G-actin purification. This study was supported by MOE Tier 2 (MOE-T2EP30121-0015), National Research Foundation Singapore under its Open Fund - Individual Research Grant (MOH-000955) and administered by the Singapore Ministry of Health's National Medical Research Council, National Research Foundation Singapore NRF-NRFI08-2022-0012, and MOE Tier 3 (MOE2019-T3-1-012) to Y.M. in Singapore.

## Author contributions

Q.M. and Y.M. conceived and designed this study. Q.M. performed most of the wet lab experiments. H.Y. contributed to yeast genetics. W.S. and Q.M. performed the AUC experiments with guidance from J.T. M.N. and D.H. performed AFM experiments with the guidance from C.L. X.H. contributed to protein biochemistry. The manuscript was drafted by Q.M. and Y.M. and received feedback from all authors. Y.M. supervised these studies.

## Competing interests

The authors declare no competing interests.
