## [Peer Review File · Nature Communications]

Reviewers' Comments:

Reviewer #1:

Remarks to the Author:

This is a very interesting and well-supported manuscript Qianqian Ma, Yansong Miao and colleagues investigating the function of the fungal Spa2 protein in actin filament regulation. Spa2 is a component of the polarisome, which associates with formin at the bud tip of budding yeast cells. Through energy starvation experiments *in vivo*, they first show that Spa2 is necessary for the observed formation of thicker actin cables during glucose depletion, and that this depends on its N-terminal half, which associates with the actin bundles. Through *in vitro* experiments, they then dissect the function of Spa2 N-terminus, showing that it binds ADP-actin-binding, promotes ADP-actin filament nucleation and elongation, and filament bundling. Bundling depends on multivalent interactions conferred in part through a disordered region, and the Spa2 N-terminus can form LLPS droplets *in vitro*. Finally, they show that Spa2 bundling activity *in vitro* and acting cable stabilization *in vivo* requires the actin D-loop to be in the closed conformation and is abolished by point mutations in this region.

The work is well presented, very complete and of high quality. I have only relatively minor comments.

One of the major points that is unconvincing is that what criteria are used to differentiate a role in actin cables crosslinking vs production *in vivo*. From what I understand, the authors use two different measurements of Abp140-3GFP intensity: the fluorescence normalized per cable, which indeed gives an indication of cable thickness (and thus cable bundling); and the total cell fluorescence. What this latter measurement indicates is unclear. A change in the total cellular fluorescence (if total cell volume is captured) will indicate a change in protein levels, not a change in localization. If the quantification is performed only on one z-plane, then it may indicate some localization change, but I don't think it can be specifically interpreted as a change in actin cable production. Certainly, this value will change also upon changes in bundling patterns. Thus, I think that the data *in vivo* cannot distinguish roles in bundling and nucleation. It is a limitation of what can be done *in vivo*, and the *in vitro* data largely compensates for that, but the authors should rephrase some of the statements made in the initial part of the manuscript. For instance, I don't think the statement that spa2-281-535 cannot stabilize actin cables but can promote their production (end of page 5) can be made at this stage of the results. Similarly (at the beginning of page 6), the text should be re-written to avoid postulating a specific role of Spa2-535 in actin polymerization. The data shows that cables are thicker, but, at this point of the results, what specific aspect of actin cable regulation is altered is not known.

A second aspect I find unconvincing is the idea of Spa2 wetting the actin cables. While the idea is appealing, the data does not directly support it. From the images shown in Fig 5a, I am not convinced that Spa2(1-535) and Spa2(281-535) display different localizations on the actin filaments. The second does not seem substantially more punctate than the former, though it is clear that the former promotes actin filament bundling.

Minor comments:

Fig 1b: The quantification lacks values for non-starved cells (which would control whether the mutants affect actin cables in the non-starved conditions). The difference in Abp140-3GFP intensity between pre- and post-glucose starvation appears very clear in the chosen example images, but the quantification shows that there is a strong overlap of intensities in the two distributions. Is it that a few cables become bundled, but others remain unchanged, or are intensities homogeneous per cell, or is it a technical problem?

Fig 1h, what is ED?

Fig 4h, typo in droplet

Could I suggest relabelling the fragments into Spa2(281-535) or Spa2(1-535) (and similar for

other fragments) to clarify which aa are included?

The note and fig S6 on conservation of Spa2 is a bit tentative. There is no need to do work in this direction for this manuscript, but the observation that similar regions are folded may not be sufficient to postulate conservation of function. I agree it is likely, but the organisation of the alpha helices shown in Fig S6 are sometimes quite different from each other.

Reviewer #2:

Remarks to the Author:

This manuscript by Ma et al. characterizes Spa2 protein in the budding yeast. It consists of in vivo and in vitro studies, and in the in vitro studies, the authors show that fragments of Spa2 protein promotes polymerization of ADP-G-actin but not ATP-G-actin, and that they bundle ADP-actin filaments. They further conclude that the filament formation is due to the actin nucleation activity of Spa2. In the in vivo studies, they show that Spa2 is necessary for the formation of actin bundles in ATP-depleted (ES) cells. They suggest that those actin bundles are derived from de novo polymerization of ADP-G-actin generated in the ES cells. ADP-G-actin-specific nucleator has not been reported, and the actin remodeling in ES yeast cells is also a novel phenomenon. Moreover, the authors utilized various cutting edge technologies to make their point, and I enjoyed reading this manuscript. However, as detailed below, both in vitro and in vivo parts have weaknesses, and I am hesitant to recommend publication of this manuscript in a top-ranking journal unless all major issues in either in vivo or in vitro sections are properly addressed.

Major comments for the in vitro studies:

1. The authors conclude that Spa2 "nucleates" polymerization of ADP-G-actin. While I also feel that Spa2 is a nucleator based on the evidence provided in this current ms, I must say rigorous evidence is lacking to demonstrate that Spa2 is indeed a new type of actin nucleator. In the experiment shown in Figure 3b, short filaments generated by the severing activity of cofilin might have elongated in the presence of Spa2. There are no preformed filament seeds in the experiment shown in Figure 3d, but if Spa2 accelerates polymerization, longer filaments may be generated more rapidly in the presence of Spa2 than in the absence, and those longer filaments would appear on the observation surface more efficiently due to the size exclusion effect of methylcellulose. To demonstrate Spa2 is truly nucleating de novo polymerization, the authors should show that a nascent filament emerges from a dimer (or oligomer) of Spa2 molecules.

2. The authors' model on how Spa2 promotes actin polymerization predicts that the critical concentration for polymerization of ADP-actin should not be affected by Spa2. It would be nice if this point is examined experimentally.

Major comments for the in vivo studies:

3. The authors first characterized several mutant cells lacking known actin binding proteins in terms of assembling actin bundles induced by ES, but only cells lacking Spa2 exhibited defective actin remodeling, leading them to conclude that Spa2 is responsible for this cellular response. I wonder why they did not include fimbrin and alpha-actinin, well-established actin crosslinking proteins present in yeast, in the screening. The present data unequivocally show that Spa2 is necessary for the actin remodeling, but it is still possible that fimbrin and/or α -actinin may also be required for the bundle formation in vivo. This is a real concern since bundle formation in vitro was observed in the presence of a high concentration (5 μ M) of Spa2-525, which is probably higher than the cellular concentration of Spa2. Moreover, the bundling process took \sim 15 min in vitro, whereas cellular response was complete within 5 min. Possible involvement of fimbrin and α -actinin should be addressed experimentally.

4. The authors assume that the actin bundles formed in ES cells were assembled de novo using ADP-G-actin in a manner dependent on the nucleation and bundling activity of Spa2. However, evidence is lacking to exclude the possibility that Spa2 simply bundled the existing actin filaments. This latter possibility is consistent with the results that actin filaments bundled by Spa2 were resistant to severing by cofilin, and cofilin mutation did not perturb the bundle formation in ES cells.

5. Very low concentration (10 nM) of Spa2-535 was sufficient to promote polymerization of ADP-actin in vitro, while much higher concentrations (5 μ M) were needed to form bundles in vitro and also for phase separation (>3 μ M in physiological KCl conc). Please discuss these concentrations with the estimated cellular concentration of Spa2.

6. Discussion starts with a very general statement "The dynamic and timely remodeling of the actin cytoskeleton during diverse signaling events is essential to coordinate physiological processes or adapt to pathological conditions.". This is true in many cases, but whether it also applies to the ES-induced actin remodeling in yeast is unclear. The authors suggest "Such (Spa2-dependent) actin cytoskeleton remodeling under stress conditions might avoid the energy-intensive regeneration of actin filament and its treadmilling cycles, allowing better cell recovery while living conditions improve" in the concluding sentence of Discussion. I am not convinced by this argument, since, in my understanding, the cells under the ES condition are unable to charge monomeric actin with ATP even if they wished to do so and are not polymerizing ADP actin in order to save ATP. Moreover, physiological significance of Spa2-dependent actin bundles is somewhat questionable because those actin filaments are thickly surrounded by amorphous Spa2 clouds (AFM images in Fig 5e and f), and they are unlikely to interact normally with other actin binding proteins including myosin V.

Minor points:

What is "wetting"? Apparently this word is used frequently in this ms to describe "binding", but it sounds strange to me because actin is a hydrophilic molecule and is solubilized in water during experiments. If it has some special implication, please explain. Otherwise, please change to "binding".

"The IDR and nucleation core of Spa2 are evolutionarily conserved by coexistence in the fungus kingdom, suggesting a universal adaptation mechanism by remodeling ADP-actin under glucose starvation." (the last sentence of Abstract) If this response is restricted to the fungal kingdom, it is misleading to say it is a "universal" adaptation mechanism.

Fluctuations of ATP/ADP ratio in response to glucose deprivation and to the addition of 2-DG is difficult to follow for outsiders of the energy metabolism research field. For instance, it was difficult for me to understand why ATP level is restored after 30 min of glucose removal, and why aerobic respiration is restored in the absence of glycolytic pathway when 2-DG was added (page 3-4). Although those issues are not central to the topic of this ms, it would be nice if a few words are added to avoid the frustration of cytoskeleton researchers.

Nomenclature is somewhat confusing. Spa2-535, Spa2-281-535 and Spa2-281-535-GFP represent a truncated Spa2 from aa 1 to 535, from aa 281-to 535, and a truncated Spa2 of aa 281 to 535 fused with GFP, respectively. This is confusing because "-" means three different things in those names.

In the experiment shown in Figure 3a, a modified, non-polymerizable actin was used to examine binding of G-actin with Spa2. The modification is nicking of D-loop, which has profound functional consequence of inhibiting polymerization. Meanwhile the authors found that the structure of D-loop is critically important for Spa2 binding. Some brief discussion is needed in relation to this issue.

ATP-actin filaments are used in some experiments (e.g., middle of page 9 and middle of page 10). How were those ATP-actin filaments prepared and maintained in the ATP-state during the experiment?

"Whereas phalloidin stabilizes the D-loop in a closed conformation, jasplakinolide (JASP) inhibits phosphate release, prevents dynamic conformational changes of the intrastrand interface, and locks actin filament in the open-D-loop state regardless of the nature of bound nucleotides 20". (top of page 10). More recently, a different result was reported by Pospich et al. (Structure 28:437-449, 2020).

Some figures are difficult to understand and need improvements.

In some graphs (such as Figure 2f and g), the horizontal axis is labeled as "+ - (-glu)" and "- + (+2-DG)". This is VERY confusing. Please simply write -glu and +2-DG below the axis, as in Figure 2c and d.

The numbers associated with each trace in Figure 3c do not seem to correspond to the values shown along the vertical axis. Please check.

What does "normalized c(s)" mean along the vertical axis of Figure 4a-c?

In Figure 4d and f, a number of kinetic parameters are written, but it is not shown what those parameters refer to. Also, it is not clear to me what the two cartoons connected by an arrow in those figures mean.

The vertical axis of Figure 5d has no label.

The labels for the vertical axis of Figure 6c and d say "Mean intensity per filament (+/-2-DG)" and "Mean intensity per filament (-/+glu)". My guess is that they mean "Mean intensity in the presence of 2-DG divided by mean intensity in the absence of 2-DG" and "Mean intensity in the absence of glucose divided by mean intensity in the presence of glucose", respectively. Simply labeling as "Relative mean intensity per filament" along the axis and noting in the legend that the values were normalized against the value in the absence of 2-DG (or presence of glucose) would be probably much easier to grasp. Same comments apply to other similar figures.

What are "L" and "H" in Figure 6f?

Although this ms is mostly understandable, there are some difficult sentences, in addition to awkward expressions and grammatical errors. The following need to be improved or corrected for clarity.

"The formation of the actin nucleus with more than two actin subunits can surpass the rate-limiting step of actin polymerization" (bottom of page 7)

"biological replicates" (in the legend to Figure 3, page 38)

"Spa2-535 proteins induce actin filament cohesion and relaxation to the elongated shape without creating isotropic interfacial tension to contract the bundles" (line 4 of page 10).

"a nucleation-potent conformation" (line 2 of page 13).

"For measurements of Spa2 with actin monomer" (middle of page 20)

"Yeast cells were cropped with background subtraction background was subtracted from average projections in Fiji." (middle of page 21)

"conservative analysis" (middle of page 27)

"Screening of ES-triggered actin remodeling in yeast mutants." (Title of Extended Fig 2, page 49).

"actin cables in WT actin-binding protein mutants" (legend to Extended Fig 2, page 49).

Reviewer #3:

Remarks to the Author:

This paper identifies a molecular mechanism underlying actin remodeling induced by glucose depletion in budding yeast. The authors examined several mutant yeasts and identified Spa2 as a factor required for this process. A portion of Spa2 directly interacts with actin and modulates actin biochemistry in vitro. Although the initial findings are interesting and may potentially provide an insight into our understanding of the role of actin in energy-depleted cells, interpretation of the data is not rigorous enough to support the authors' conclusion (see below). I don't recommend publication of this paper in the current form.

Major Points:

1) Most of all, I am concerned with the way the authors interpret the TIRF data. Figure 3b and 3d show an increase in the actin filaments from ATP-F-actin and ADP-G-actin, respectively. A plausible explanation for these findings is the enhanced fragmentation of F-actin induced by Spa2. This activity is quite apparent in the second and third rows of Figure 3b and in some images of Extended Fig. 4c. I would point out that in Figure 3b, the amount of G-actin at early time points is too low to support filament nucleation by any mechanisms. As long known for cofilin and ADF, the severing activity effectively increase the filament number, which is not readily distinguished from the nucleation activity. Spa2 perhaps interacts with F-actin formed from both ATP- and ADP-G-actin and enhances its fragmentation. This severing activity of Spa2 is probably ineffective against Pi-bound F-actin as in Extended Fig. 4f.

I would also emphasize that "ATP-F-actin" has the same properties as ADP-F-actin after several minutes of polymerization and the release of Pi. Difference with "ATP-F-actin" and "ADP-F-actin" in this paper can be attributed to the different concentrations (0.5 μ M versus 3 μ M) of F-actin but not the initial nucleotide states. There is no 'memory' for initial nucleotide states in the filament. Indeed, Figure 6a shows that phalloidin, which can differentiate the structures of ADP-F-actin and ADP-Pi-F-actin (ref. 20 and its cited *Curr Biol* 2015, 25:2057-2062), allows the bundling of F-actin by Spa2. This shows that phalloidin stabilizes the ADP-F-actin structure in the filaments made from "ATP-F-actin". Thus, Spa2 binds and bundles F-actin regardless of the initial nucleotides. A plausible reason for the negative data throughout this study would be the low concentration of "ATP-F-actin". The lack of consideration of F-actin fragmentation by Spa2 and its effects on filaments in the bundling assay hampers the publication of this paper in the current form.

2) A highly related study which first described the rapid glucose deprivation-induced loss in the yeast actin polarity (*MBoC* 2004, 15:1544) is not cited.

3) A finer time course of ATP-biosensor output and actin morphological changes than in Figure 1 must be shown. Which comes first, ATP depletion, actin cable remodeling or Spa2 translocation? Which is the most persistent?

4) It is also very important to show the exact ATP concentration. The original paper (*JCS* 132: jcs230649) did some estimation but the true relationship between the "Queen ratio" and the ATP concentration was not directly determined. Calibration of this relationship using the current microscope or measurement of the ATP/ADP ratio in soluble G-actin (isolatable using DNase I resin, ex. *Plant Physiol.* 100:1716) will signify this paper.

5) Include supplementary timelapse movies for the images shown in Figure 1a, 1c, 1e, 1h, 1j, 2b and 2e.

6) Statistical analysis methods are unclear in places. The method section does not provide sufficient information. In Figures 1b, 1d, 1f and 2, for example, the data are not presented with appropriate statistical information.

7) The y-axis label is often confusing. For example, in Figure 1b, clearer wording such as 'the fold increase in the mean intensity on the filament' is recommended. It will be difficult for many readers to realize that the 'ratio' is shown, which is indicated only in the parenthesis. Besides, why do the authors use "Mean intensity per filament"? This value may change by whether a focal plane contains strongly label actin cables or not. Such errors should be avoided. Another example is Figure 1d. "Queen ratio" is confusing. It is not immediately clear whether the "ratio" indicates the FRET ratio (as used in the original paper) or its ratio between glu(-) and glu(+). Clarify these problems on the graph as well as in the figure legends.

8) In Figure 2a, the data must be presented by absolute anisotropy values.

9) In Extended Fig. 4b, Spa2 281-535 indeed changes anisotropy around 10 μ M ATP-actin, contrary to the authors' statement in p6.

10) Show evidence that no filaments were formed or contaminated during the anisotropy measurement (Figure 2a and Extended Fig. 4b).

11) Figure 2a data cannot be interpreted as showing the actin interacting portion because some Spa2 probes may just not have labeled residues that change anisotropy upon actin binding. This issue can be addressed by examining competition between Spa2 fragments in anisotropy.

12) Although timelapse imaging of actin assembly and disassembly using TIRF is useful to visualize elongation and severing reactions, it does not accurately report the amount of actin filaments and nuclei formed. In the TIRF assay, it is not readily apparent how much of the filament in the solution is trapped on the glass surface. Reanalysis of several key biochemical properties of Spa2 with the pyrene actin assay is strongly recommended. In addition, describe the labeling method for fluorescein Alexa 488-labeled Spa2 in the method section.

13) What is "nonpolymerizable monomeric actin (Hypermol, Germany)" in the method?

14) Remove Extended Fig. 6 which does not show any evidence that this portion is strongly conserved in fungi Spa2-related proteins. In addition, AlphaFold is not reliable in predicting the structure of unfolded peptides.

15) Significance of Fig. 4 is unclear for this study as there is no Spa2 condensates along the yeast actin cable. Remove this to the supplement or examine the effects of a wide range of Spa2 concentrations on actin biochemistry.

Minor Points:

- 16) Although Abp140 was used for quantification of F-actin in live yeast cells, full-length Abp140 has an F-actin bundling activity in vitro (Oncogene 16:121) and hence, may bind F-actin cooperatively and in a non-linear fashion. Confirmation of the key findings in Figure 1 using fluorescent phalloidin is recommended.
- 17) Include a summary and a graphical explanation for the known domain functions and interacting molecules of Spa2 (in Figure 2a).
- 18) Describe in detail how actin solutions were diluted with TIRF buffer before imaging and what is the final constituents during imaging.
- 19) Did TIRF buffer contain "50 mM" DTT? Doesn't it affect actin polymerization and disassembly?
- 20) Clearly indicate in the method section which data was obtained by using "Total internal reflection fluorescence microscopy" and "In vitro actin polymerization and imaging".
- 21) "Whereas phalloidin stabilizes the D-loop in a closed conformation, jasplakinolide (JASP) inhibits phosphate release, ..." in p12: This sentence needs correction. Phalloidin also inhibits phosphate release (Biochimica et Biophysica Acta 1990, 1035:197-200).

Reviewer #4:

Remarks to the Author:

Review of Nature Communications 414160

In the manuscript "Spa2 remodels ADP-actin via molecular condensation under glucose starvation", Ma and colleagues identify the protein, Spa2, responsible for actin bundle formation that is observed shortly after glucose and energy starvation of yeast. Through the detailed experiments, the authors discovered that Spa2 can nucleate ADP-G-actin filaments and protect ADP-actin filaments from severing by cofilin. Spa2 modulation of ADP-actin monomers and filaments depends on the actin D-loop. Furthermore, the authors implicate condensation of Spa2 as a key mechanism by which it can modulate both filament nucleation, stabilization, and bundling. These results are exciting and provide insight into a new mechanism by which actin dynamics can be regulated in cells.

Importantly, this study is well suited for the wide readership of Nature Communications. The manuscript will be of interest for researchers in stress response, actin regulation, and condensate biology, as well as those interested in fungi biology, as Spa2 is well conserved. Below are comments for the authors to consider while revising their manuscript. If these can be addressed, this reviewer will support publication of this study in Nature Communications.

Comments:

- 1) In the abstracts, the authors should highlight the new role for Spa2 that they have discovered. Perhaps they can contract this new role with a few words describing what was previously understood about Spa2.
- 2) In the third line on page 3 of the current version of the manuscript, the authors write ADP:ATP ratio. For consistency with the rest of the manuscript, they should use ATP:ADP.
- 3) In Figure 1 and associated methods, the authors quantify various aspects of actin bundling. It would be helpful in either the results or methods to comment on the thresholds used for identifying quantifiable filaments. Was there an intensity or length of filament that served as a cut off, below which structures were not analyzed?
- 4) In Figure 1E, why does the WT image appear to have similar bundling as GS after 5 minutes?
- 5) For experiments shown in Figure 2, were the Spa2 variants expressed in a Spa2 delta background, or is wild-type Spa2 also present in these cells?
- 6) In Figure 2, are Spa2 variants expressed equally? One may imagine that some of the observed effects may be due to expression differences. Because they are fluorescently tagged in initial experiments, fluorescence intensity per cell can be used as an expression calibration. However, in experiments using unlabeled Spa2 variants and ABP-GFP, the authors should include data from western blots showing what the expression is across the population of cells expressing each variant. If the expression levels are different, the authors should include a comment about what impact this may have on cellular response to GS and ES, especially since phase separation is invoked later in the manuscript.
- 7) In Figure 4, since the N-terminal IDR contributes to condensation in the following assays, why is

no assembly detected in these assays? It would be interesting to see if AUC at higher concentrations induced self-assembly of the 2-281 fragment. This data and the microscopy data in Figure 4 suggest that it is both N-terminal and dimerization of 281-535 that drive condensate formation. The title of this section makes it seem that the N-terminal alone drive phase separation, but there seems to be contribution from the entire protein. The title of this section should be corrected.

8) The final sentence in the top paragraph of page 8 of the current version of the manuscript is a bit confusing. If there is no obvious self-association, how is there dimerization?

9) In the first sentence of the first full paragraph on page 9, the authors refer to microtubule wetting. This has also been observed for LAT condensate proteins on actin filaments, PMID 31268421.

10) The authors refer to mechanical stabilization of actin filaments preventing cofilin severing. The concept of mechanical stabilization should be clarified a bit for the reader. Is the intended description that the filament is super stable and thus cannot be severed or that Spa2 coats the filament and prevents cofilin binding and severing. In the former, cofilin can still access the filament. In the latter, cofilin cannot access the filament. It's probably a combination of both, so a more detailed description is warranted.

11) In the 'In vivo actin cable imaging and image analysis, a sentence states that Yeast cells were cropped with background subtraction background was subtracted...' Remove one of the background subtractions.

12) If possible, please use magenta and green color schemes in their figures for microscopy images to improve accessibility for readers with red/green color blindness.

13) In Figure 4H, drpolets should be droplets.

Reviewer #1 (Remarks to the Author):

This is a very interesting and well-supported manuscript Qianqian Ma, Yansong Miao and colleagues investigating the function of the fungal Spa2 protein in actin filament regulation. Spa2 is a component of the polarisome, which associates with formin at the bud tip of budding yeast cells. Through energy starvation experiments *in vivo*, they first show that Spa2 is necessary for the observed formation of thicker actin cables during glucose depletion, and that this depends on its N-terminal half, which associates with the actin bundles. Through *in vitro* experiments, they then dissect the function of Spa2 N-terminus, showing that it binds ADP-actin-binding, promotes ADP-actin filament nucleation and elongation, and filament bundling. Bundling depends on multivalent interactions conferred in part through a disordered region, and the Spa2 N-terminus can form LLPS droplets *in vitro*. Finally, they show that Spa2 bundling activity *in vitro* and acting cable stabilization *in vivo* requires the actin D-loop to be in the closed conformation and is abolished by point mutations in this region.

The work is well presented, very complete and of high quality. I have only relatively minor comments.

We are grateful for Reviewer 1's highly constructive feedback and valuable recommendations, which help us greatly to enhance the quality of our manuscript.

1. One of the major points that is unconvincing is that what criteria are used to differentiate a role in actin cables crosslinking vs production *in vivo*. From what I understand, the authors use two different measurements of Abp140-3GFP intensity: the fluorescence normalized per cable, which indeed gives an indication of cable thickness (and thus cable bundling); and the total cell fluorescence. **What this latter measurement indicates is unclear.** A change in the total cellular fluorescence (if total cell volume is captured) will indicate a change in protein levels, not a change in localization. If the quantification is performed only on one z-plane, then it may indicate some localization change, but I don't think it can be specifically interpreted as a change in actin cable

production. Certainly, this value will change also upon changes in bundling patterns. Thus, I think that the data in vivo cannot distinguish roles in bundling and nucleation. It is a limitation of what can be done in vivo, and the in vitro data largely compensates for that, but the authors should rephrase some of the statements made in the initial part of the manuscript. For instance, I don't think the statement that spa2-281-535 cannot stabilize actin cables but can promote their production (end of page 5) can be made at this stage of the results. Similarly (at the beginning of page 6), the text should be re-written to avoid postulating a specific role of Spa2-535 in actin polymerization. The data shows that cables are thicker, but, at this point of the results, what specific aspect of actin cable regulation is altered is not known.

Response: We appreciate the insightful suggestion from Reviewer 1. We agree with the points raised by the Reviewer and appreciate the points that allow us to clarify our methods and provide modifications we have undertaken to prevent potential overstatements.

Firstly, we did measure the protein level of Abp140-3GFP on the actin cables without analyzing cytoplasmic signals of Abp140-3GFP, a method we employed to indicate the total quantity of actin filaments by assuming a linear range of Abp140-3GFP binding to F-actin. We acknowledge that this assumption may not be entirely accurate as we lack direct evidence supporting it. As such, we have now clarified the potential semi-quantitative limitations associated with using the Abp140 marker to describe in vivo actin cables. Despite this, it remains one of the few feasible methods for cell biologists to investigate these processes in yeast, as acknowledged by the Reviewer.

Secondly, we have given careful consideration to the quantification of Z-stack imaging to ensure a fair comparison between groups. We have ensured the acquisition of images across the same volume and at comparable cellular positions.

Finally, to avoid presuming a specific role for Spa2-535, as suggested by the Reviewer, we have made modifications to the text. We have deleted the sentence on page 5, which read, "suggesting a role of Spa2-281-535 in cable polymerization but not crosslinking". We have replaced "To understand the Spa2-535-mediated increase in actin

polymerization" with "To understand the Spa2-535-mediated increase in actin cable signal in vivo".

2. A second aspect I find unconvincing is the idea of Spa2 wetting the actin cables. While the idea is appealing, the data does not directly support it. From the images shown in Fig 5a, I am not convinced that Spa2(1-535) and Spa2(281-535) display different localizations on the actin filaments. The second does not seem substantially more punctate than the former, though it is clear that the former promotes actin filament bundling. Response: We are thankful to Reviewer 1 for bringing up the need for a more thorough explanation of these polymer physics terms and their connection to biological macromolecules.

Biomolecular wetting describes the property of protein condensates that exhibit low surface tension, which facilitates their two-dimensional fusion and expansion. We utilized atomic force microscopy (AFM) in this study to discern the nanometer-scale lateral properties and the increase in height due to nano-condensation. It should be noted that the XY resolution of AFM is typically at least an order of magnitude lower, contingent on the type of tip used. Our Z-directional measurements in Fig. 5 reveal a significant increase in the coalescence of the nanoscale colloidal assembly formed by Spa2(1-535).

Wetting involves intermolecular interactions when two substances are in contact, with the degree of wetting (or wettability) determined by the balance between adhesive and cohesive forces. Conversely, colloidal assemblies, such as homotypic protein oligomers, are more concerned with the structured organization of proteins. Incorporating these concepts into protein assembly aids in understanding the behavior of protein materials at the micro and nano scales.

To clarify these connections between physical terms and biological phenomena, we have now included a brief elaboration in the relevant section of our manuscript at line 256 on page 12.

Minor comments:

3. Fig 1b: The quantification lacks values for non-starved cells (which would control whether the mutants affect actin cables in the non-starved conditions). The difference in Abp140-3GFP intensity between pre- and post-glucose starvation appears very clear in the chosen example images, but the quantification shows that there is a strong overlap of intensities in the two distributions. Is it that a few cables become bundled, but others remain unchanged, or are intensities homogeneous per cell, or is it a technical problem?

Figure R1. Quantification of actin cable mean intensity and ATP level upon GS. The fluorescence intensity of the actin cable and ATP sensor Queen in WT, *hxx2Δ*, and *cbp2Δ* strains was measured and compared in glucose starvation and normal conditions.

Response: First, the measurement of actin filament in the absence of GS serves as a control, as the Reviewer pointed out, but it also reveals baseline differences in the mutants. Our analysis indicates that, without GS treatment, the *cbp2* null mutant exhibits reduced actin bundling compared to the wild type (WT), which is consistent with its lower cellular ATP concentration. Conversely, in the absence of GS treatment, the *hxx2* null mutant shows an increase in actin bundling compared to the WT, even though their ATP concentration are comparable. We are currently unable to ascertain if the *hxx2* null mutant displays a change in the absolute levels of ADP compared to the WT, as practical methods to measure these quantities quantitatively are lacking. However, we believe these results represent the cellular conditions and might be due to complex metabolic shifts in the mutant, which extends beyond the scope of this study. Hence, we have decided to retain these findings, but we have included a brief discussion on this in the revised results section at line 78 on page 4.

Second, we appreciate Reviewer's keen observation. In GS-treated cells, there is an increase in crosslinking of thin filaments, but some thin filaments are retained, maybe especially those that are distantly positioned from one another. Our method employed an unbiased selection of all observed actin cables, hence the observed overlap in crosslinking levels between treated and untreated cells.

The analysis was conducted using raw data in TIFF format. All figures displayed have been collectively assembled across all individual panels in the same figure to avoid a biased comparison of signals. While the visual perception of the images might suggest a more significant difference than what the computer-calculated values indicate, it is important to note that all figures and calculations truthfully represent their original data. The differential response to the images and the software-based analysis likely stems from the variable sensitivity in interpreting the images.

4. Fig 1h, what is ED?

Response: we have now corrected to +2DG

5. Fig 4h, typo in droplet

Response: typo corrected, thanks.

6. Could I suggest relabelling the fragments into Spa2(281-535) or Spa2(1-535) (and similar for other fragments) to clarify which aa are included?

Response: We thank reviewer suggestion and we have now changed Spa2-535 to Spa2(1-535), Spa2-281 to Spa2(1-281), Spa2-816 to Spa2(1-816), Spa2-1087 to Spa2(1-1087), Spa2-1306 to Spa2(1-1306).

7. The note and fig S6 on conservation of Spa2 is a bit tentative. There is no need to do work in this direction for this manuscript, but the observation that similar regions are folded may not be sufficient to postulate conservation of function. I agree it is likely, but

the organisation of the alpha helices shown in Fig S6 are sometimes quite different from each other.

Response: We concur with the Reviewer that the shape of the folded region alone cannot definitively support the conservation of functionality. This has also been underscored by Reviewer #3. Consequently, we have rephrased our analysis to propose that the coexistence of intrinsically disordered regions (IDR) and neighbouring folded regions is conserved, and such combination is essential for protein condensation and function, but we also acknowledge that the conformation of both parts requires empirical validation.

Reviewer #2 (Remarks to the Author):

This manuscript by Ma et al. characterizes Spa2 protein in the budding yeast. It consists of *in vivo* and *in vitro* studies, and in the *in vitro* studies, the authors show that fragments of Spa2 protein promotes polymerization of ADP-G-actin but not ATP-G-actin, and that they bundle ADP-actin filaments. They further conclude that the filament formation is due to the actin nucleation activity of Spa2. In the *in vivo* studies, they show that Spa2 is necessary for the formation of actin bundles in ATP-depleted (ES) cells. They suggest that those actin bundles are derived from *de novo* polymerization of ADP-G-actin generated in the ES cells. ADP-G-actin-specific nucleator has not been reported, and the actin remodeling in ES yeast cells is also a novel phenomenon. Moreover, the authors utilized various cutting edge technologies to make their point, and I enjoyed reading this manuscript. However, as detailed below, both *in vitro* and *in vivo* parts have weaknesses, and I am hesitant to recommend publication of this manuscript in a top-ranking journal unless all major issues in either *in vivo* or *in vitro* sections are properly addressed.

Major comments for the *in vitro* studies:

1. The authors conclude that Spa2 “nucleates” polymerization of ADP-G-actin. While I also feel that Spa2 is a nucleator based on the evidence provided in this current ms, I must say rigorous evidence is lacking to demonstrate that Spa2 is indeed a new type of actin nucleator. In the experiment shown in Figure 3b, short filaments generated by the severing activity of cofilin might have elongated in the presence of Spa2. There are no preformed filament seeds in the experiment shown in Figure 3d, but if Spa2 accelerates polymerization, longer filaments may be generated more rapidly in the presence of Spa2 than in the absence, and those longer filaments would appear on the observation surface more efficiently due to the size exclusion effect of methylcellulose. To demonstrate Spa2 is truly nucleating *de novo* polymerization, the authors should show that a nascent filament emerges from a dimer (or oligomer) of Spa2 molecules.

Response: We appreciate the insightful comments which delve into the core aspects of nucleator studies but also highlight the challenges when the association between nucleator and filament ends involves weak and transient interactions. To address these points, we have provided additional clarification and performed further experiments, which we hope will now offer more substantial support to our findings.

First, we quantified the ratio of total actin intensity to actin elongation speed. Our results showed that Spa2(1-535) increases this ratio by approximately 3.3-fold, which further increases to around 4.4-fold with Spa2(281-535). This further suggests that Spa2 is an ADP-actin polymerizer. We have now incorporated these findings into Extended Fig. 4d.

Second, we conducted a new TIRF assay using 0.8 μ M ADP-actin, a concentration significantly lower than the other 3 μ M condition. Despite this, we still observed a robust enhancement of seed generation and elongation by Spa2(281-535) and Spa2(1-535). These results are now included as Extended Fig. 4e-g.

Third, we attempted a single-molecule Total Internal Reflection Fluorescence (TIRF) assay to track the association between the barbed end of Spa2(281-535)-tri-Alex647 and Oregon ADP-actin, as well as the growth of its filament. However, due to the low affinity between Spa2 and ADP-actin and the inherent challenges of this single-molecule technique, we were unable to obtain robust results as requested by the Reviewer. The affinity of Spa2 for actin was in the micromolar range, while other nucleators capable of directly aligning several ATP-G-actin together for nucleation exhibit much higher affinities, orders of magnitude greater. These include Formin (0.001 nM) and Spire (0.1 nM)¹, and Ena/VASP(3.2 nM)². We have included this information in the discussions of the revised manuscript at line 327 on page 16.

2. The authors' model on how Spa2 promotes actin polymerization predicts that the critical concentration for polymerization of ADP-actin should not be affected by Spa2. It would be nice if this point is examined experimentally.

We are grateful for the Reviewer's suggestion. In accordance with this, we conducted an actin critical concentration assay. Our results showed that the Spa2 variants did not affect the critical concentration of ADP-actin. In our study, the critical concentration of ADP-actin was around 0.7 μM , which is close to the reported value of 0.8 μM ³. These findings have now been included in Extended Fig. 4i.

Major comments for the in vivo studies:

3. The authors first characterized several mutant cells lacking known actin binding proteins in terms of assembling actin bundles induced by ES, but only cells lacking Spa2 exhibited defective actin remodeling, leading them to conclude that Spa2 is responsible for this cellular response. I wonder why they did not include fimbrin and alpha-actinin, well-established actin crosslinking proteins present in yeast, in the screening. The present data unequivocally show that Spa2 is necessary for the actin remodeling, but it is still possible that fimbrin and/or α -actinin may also be required for the bundle formation in vivo. This is a real concern since bundle formation in vitro was observed in the presence of a high concentration (5 μM) of Spa2-525, which is probably higher than the cellular concentration of Spa2. Moreover, the bundling process took ~15 min in vitro, whereas cellular response was complete within 5 min. Possible involvement of fimbrin and α -actinin should be addressed experimentally.

Response: We thank the Reviewer for the excellent points.

First, in budding yeast, there is no actinin protein but does have fimbrin Sac6, we had this *in vivo* data in Extended Fig 1. Sac6 deletion mutant still showed bundling response to GS, similar to WT. Now, we added particular emphasis on this in the result following the corresponding data.

Second, we performed additional experiments of Sac6-localization, which would not be relocated onto actin cables upon ES and remains on the actin patches, indicating their different roles during ES for F-actin crosslinking.

Figure R2. Localization of Sac6 during energy starvation. The GFP tag was added to the C-terminus of Sac6 at the endogenous locus. Fluorescence imaging was performed under the conditions specified. The fluorescence intensity of the actin patch signal of Sac6 was measured and compared in states of energy starvation and normal conditions.

Thirdly, in our prior submission, we indeed evaluated the concentration-dependent impact of Spa2 on ADP-F-actin crosslinking, utilizing a range of Spa2(1-535) concentrations (currently depicted in Extended Fig. 5c, d). A noticeable crosslinking of ADP-F-actin begins to occur at 2.5 μM Spa2(1-535). An important consideration when studying multivalent associative macromolecules is the variance in crowding between *in vivo* and *in vitro* conditions, which can significantly influence the critical saturation concentration required for phase separation⁴⁻⁶. We now conducted additional experiments that illustrate how the incorporation of a crowding-mimicking agent (5% PEG, mw3350) can shift the critical concentration of Spa2(1-535) to 1 μM necessary for phase separation, which aligns with the concentration of Spa2 observed *in vivo* at the bud tip and on the actin cables (see the reply of Question 5 below). Furthermore, under crowded conditions, Spa2 displays a heightened ability to crosslink F-actin. We have now included this new data in Extended Fig. 5d, e.

4. The authors assume that the actin bundles formed in ES cells were assembled de novo using ADP-G-actin in a manner dependent on the nucleation and bundling activity of Spa2. However, evidence is lacking to exclude the possibility that Spa2 simply bundled the existing actin filaments. This latter possibility is consistent with the results that actin filaments bundled by Spa2 were resistant to severing by cofilin, and cofilin mutation did not perturb the bundle formation in ES cells.

Response: We fully agree with the Reviewer's perspective. We apologize if our previous statements caused any confusion. We, too, believe that both scenarios occur concurrently. *De novo* polymerization of ADP-actin could start early and require nanomolar concentrations of Spa2, while the crosslinking of de novo polymerized and pre-existing filaments may occur slightly later, awaiting a significantly higher local concentration of Spa2 accumulated through condensation on the F-actin. We have now clarified this point in the discussion on page 17.

5. Very low concentration (10 nM) of Spa2-535 was sufficient to promote polymerization of ADP-actin in vitro, while much higher concentrations (5 μ M) were needed to form bundles in vitro and also for phase separation (>3 μ M in physiological KCl conc). Please discuss these concentrations with the estimated cellular concentration of Spa2.

Response: We appreciate the insightful comment.

Firstly, we have provided a detailed explanation of the concentration-dependent remodeling of F-actin at various stages, including polymerization, crosslinking, and depolymerization, in the discussion on pages 16-17.

Secondly, the combined effect of concentration and crowding on Spa2 condensation and F-actin crosslinking has been addressed in response to Question 4 above.

Lastly, in an effort to illustrate the concentration-dependent function of Spa2 both in vivo and in vitro, as suggested by the Reviewer, we have quantified the in vivo concentration of Spa2(1-535) on ADP-F-actin. We've used a method that estimates the in vivo concentration of all the proteins in the polarisome⁷. Our findings show that the local

concentration of Spa2(1-535) on F-actin can reach up to approximately 1 μ M. We have included this new data in the Extended Fig. 3f.

6. Discussion starts with a very general statement "The dynamic and timely remodeling of the actin cytoskeleton during diverse signaling events is essential to coordinate physiological processes or adapt to pathological conditions.". This is true in many cases, but whether it also applies to the ES-induced actin remodeling in yeast is unclear. The authors suggest "Such (Spa2-dependent) actin cytoskeleton remodeling under stress conditions might avoid the energy-intensive regeneration of actin filament and its treadmilling cycles, allowing better cell recovery while living conditions improve" in the concluding sentence of Discussion. I am not convinced by this argument, since, in my understanding, the cells under the ES condition are unable to charge monomeric actin with ATP even if they wished to do so and are not polymerizing ADP actin in order to save ATP. Moreover, physiological significance of Spa2-dependent actin bundles is somewhat questionable because those actin filaments are thickly surrounded by amorphous Spa2 clouds (AFM images in Fig 5e and f), and they are unlikely to interact normally with other actin binding proteins including myosin V.

Response: We concur with the Reviewer's assertion that a reversion of actin to ATP-actin is less likely under energy starvation (ES) conditions. Consequently, our discussion proposes a physiological rationale for this phenomenon during recovery when ATP stress subsides. We apologize for any confusion caused by our previous lack of clarity, and we have now elaborated on this aspect in the revised manuscript. This point is also demonstrated in the right half of our model in Fig. 6, during the recovery stage.

Moreover, while the binding of Spa2 might affect the binding behavior of Myo2 in a time and dose-dependent manner, it would not prevent their association. This assumption takes into account the temporal local recruitment of Spa2 and the wetting process. Previous studies have also shown that Myo2 can be concurrently immobilized with F-actin⁸. The Atomic Force Microscopy (AFM) results revealed a relatively stable

equilibrium stage with a high concentration of Spa2. This concentration was employed to decipher the progression and dynamic equilibrium of Spa2 condensation by enabling observation of the size of Spa2 condensates under the microscope. Even at such a high concentration, images in Fig 5a and the distribution in Fig 5h still show unoccupied spaces on the F-actin.

Minor points:

1. What is “wetting”? Apparently this word is used frequently in this ms to describe “binding”, but it sounds strange to me because actin is a hydrophilic molecule and is solubilized in water during experiments. If it has some special implication, please explain. Otherwise, please change to “binding”.

Response: We apologize for not providing a detailed explanation of wetting, a biophysical term that denotes the unique process of biomolecular condensation on the surface of flat membranes and cytoskeletal filaments⁹⁻¹⁵. This is a crucial aspect in comprehending how the material properties of multivalent condensates influence Spa2 function, which cannot be reduced to a simple binding process.

Biomolecular wetting in relation to phase separation involves sophisticated behaviors such as multivalent interactions, coalescence, and capillary effects. Multivalent interactions denote a molecule's ability to simultaneously bind with numerous targets, thereby amplifying the overall binding affinity and triggering phase transition due to the aggregated impact of weak interactions. As condensation begins, coalescence, facilitated by the wetting effect as seen in the case of Spa2, allows assemblies to merge and associate flexibly along the filament. This process involves the unification of multiple droplets or phases into a single larger entity, which is driven by the reduction of surface tension. Concurrently, the capillary effect contributes to the deformation of a liquid phase interface due to its interactions with a surface or another phase, leading to occurrences such as the spreading of the phase across the surface. These collective phenomena, fundamental to phase separation and other mesoscale phenomena, are separate from the simpler process of biomolecular binding, which pertains to one-on-one interactions or protein

oligomerizations referring to colloidal assembly and is distinctive from wetting. We added more discussion on Page 17 to clarify it better.

2. "The IDR and nucleation core of Spa2 are evolutionarily conserved by coexistence in the fungus kingdom, suggesting a universal adaptation mechanism by remodeling ADP-actin under glucose starvation." (the last sentence of Abstract) If this response is restricted to the fungal kingdom, it is misleading to say it is a "universal" adaptation mechanism.

Reply : We have now updated the description to specify 'within the fungal kingdom' in both the figure and the text.

3. Fluctuations of ATP/ADP ratio in response to glucose deprivation and to the addition of 2-DG is difficult to follow for outsiders of the energy metabolism research field. For instance, it was difficult for me to understand why ATP level is restored after 30 min of glucose removal, and why aerobic respiration is restored in the absence of glycolytic pathway when 2-DG was added (page 3-4). Although those issues are not central to the topic of this ms, it would be nice if a few words are added to avoid the frustration of cytoskeleton researchers.

Response: we have elaborated these points briefly for a broad audience and added necessary references about yeast stress biology now on pages 3 and 4.

4. Nomenclature is somewhat confusing. Spa2-535, Spa2-281-535 and Spa2-281-535-GFP represent a truncated Spa2 from aa 1 to 535, from aa 281-to 535, and a truncated Spa2 of aa 281 to 535 fused with GFP, respectively. This is confusing because "-" means three different things in those names.

Response: We appreciate the Reviewer's suggestion. We have now changed Spa2-535 to Spa2(1-535), Spa2-281 to Spa2(1-281), Spa2-816 to Spa2(1-816), Spa2-1087 to Spa2(1-1087), Spa2-1306 to Spa2(1-1306).

5. In the experiment shown in Figure 3a, a modified, non-polymerizable actin was used to examine binding of G-actin with Spa2. The modification is nicking of D-loop, which has profound functional consequence of inhibiting polymerization. Meanwhile the authors found that the structure of D-loop is critically important for Spa2 binding. Some brief discussion is needed in relation to this issue.

Response: The original nonpolymerizable (NP) actin was first reported and characterized by Rosen's lab¹⁶, where three mutations (D286A/V287A/D288A) were introduced in *Drosophila* actin. In our previous paper published in *JBC*¹⁷, we generated a nonpolymerizable variant of *S. cerevisiae* Act1 (NP-actin) by introducing the same three mutations and subsequently expressed it in insect cells for thorough characterization. These mutations are located in the longitudinal contact interface between actin monomers (along the filament axis), not affecting the D-loop. We have now added a better introduction of it on page 7.

6. ATP-actin filaments are used in some experiments (e.g., middle of page 9 and middle of page 10). How were those ATP-actin filaments prepared and maintained in the ATP-state during the experiment?

Response: We converted Ca-ATP-actin into Mg-ATP actin by incubating it in G-Buffer with 0.2mM ATP, 100 μ M MgCl₂, and 0.2 mM EGTA for 5 minutes on ice prior to use. The spontaneous actin polymerization was initiated by adding a 10 \times KME buffer mix (10 mM MgCl₂, 10 mM EGTA, and 500 mM KCl), and then allowing it to polymerize for 30 minutes at room temperature. The methods are more detailed now in the revised manuscript.

Our protocol follows the same procedures previously reports^{7,18,19}. Our specific preparations for ATP- and ADP-F-actin have highlighted Spa2's differential binding to F-actin in a nucleotide-specific manner.

7. "Whereas phalloidin stabilizes the D-loop in a closed conformation, jasplakinolide

(JASP) inhibits phosphate release, prevents dynamic conformational changes of the intrastrand interface, and locks actin filament in the open-D-loop state regardless of the nature of bound nucleotides 20". (top of page 10). More recently, a different result was reported by Pospich et al. (Structure 28:437-449, 2020).

Response: We appreciate the Reviewer's suggestion and have indeed incorporated the recommended citations. As reported by Felipe Merino et al., 2018 ²⁰, it is demonstrated that JASP locks F-actin in an open state irrespective of the nature of bound nucleotides. In another study by Sabrina Pospich et al., 2020 ²¹, they conclude that jasplakinolide behaves differently from phalloidin, despite their overlapping binding sites and similar interactions with the actin filament. While phalloidin locks F-actin in a closed state, JASP maintains F-actin in an open state. These references are in the revised manuscript.

Some figures are difficult to understand and need improvements.

8. In some graphs (such as Figure 2f and g), the horizontal axis is labeled as "+ - (-glu)" and "- + (+2-DG)". This is VERY confusing. Please simply write -glu and +2-DG below the axis, as in Figure 2c and d.

Response: We apologize for the confusion, we have now revised it

9. The numbers associated with each trace in Figure 3c do not seem to correspond to the values shown along the vertical axis. Please check.

Response: We apologize for the confusion. The values provided represent the average number across multiple time points (40s, 80s, 120s, 160s, 200s), not the final value at 200s. We have now updated the figure legend to clarify this point.

10. What does "normalized c(s)" mean along the vertical axis of Figure 4a-c?

Response: By 'normalized c(s)', we mean that we have scaled the data such that the maximum value of a single intensity peak is represented as 1, and the minimum value as 0. This approach helps to eliminate any confusion that might arise from slight variations

in concentration-driven intensity, which are not directly pertinent to the question of oligomerization species using AUC technique.

11. In Figure 4d and f, a number of kinetic parameters are written, but it is not shown what those parameters refer to. Also, it is not clear to me what the two cartoons connected by an arrow in those figures mean.

Response: The corresponding SPR curves were analyzed using a bivalent model. In this model, the K_{on} represents the association constant, while the K_{off} stands for the dissociation constant. These parameters are detailed further in the methods section. The arrow in the diagram illustrates the interaction between molecules in the mobile phase and those immobilized on the chip. We have now made these points clearer in the figure legend.

12. The vertical axis of Figure 5d has no label.

Response: We have now added the label.

13. The labels for the vertical axis of Figure 6c and d say "Mean intensity per filament (+/-2-DG)" and "Mean intensity per filament (-/+glu)". My guess is that they mean "Mean intensity in the presence of 2-DG divided by mean intensity in the absence of 2-DG" and "Mean intensity in the absence of glucose divided by mean intensity in the presence of glucose", respectively. Simply labeling as "Relative mean intensity per filament" along the axis and noting in the legend that the values were normalized against the value in the absence of 2-DG (or presence of glucose) would be probably much easier to grasp. Same comments apply to other similar figures.

Response: We appreciate the Reviewer's point that helps us to clarify our statement. We have now changed these descriptions in the method.

14. What are "L" and "H" in Figure 6f?

Response: L means low protein condensation. H means high protein condensation. We have now added these details in the figure legend in the revised MS.

Although this ms is mostly understandable, there are some difficult sentences, in addition to awkward expressions and grammatical errors. The following need to be improved or corrected for clarity.

15. "The formation of the actin nucleus with more than two actin subunits can surpass the rate-limiting step of actin polymerization" (bottom of page 7)

Response: The promotion of actin dimer and trimer formation can effectively overcome the rate-limiting step of actin polymerization.

16. "biological replicates" (in the legend to Figure 3, page 38)

Response: added now, thanks!

17. "Spa2-535 proteins induce actin filament cohesion and relaxation to the elongated shape without creating isotropic interfacial tension to contract the bundles" (line 4 of page 10).

Response: We have now rephrased the sentence to describe the biological phenomeme and biophysical properties better. "Given the unique material characteristics of Spa2 condensates, Spa2(1-535) proteins stimulate cohesion among actin filaments in sync with their elongation. This concurrent process leads to the relaxation of F-actin bundles into an elongated form, avoiding the generation of uniform interfacial tension that could otherwise prompt bundle contraction or shrinkage."

18. "a nucleation-potent conformation" (line 2 of page 13).

Response: changed to "a favorable conformation for the nucleator"

19. "For measurements of Spa2 with actin monomer" (middle of page 20)

Response: changed to "For measuring the binding affinity towards actin monomer,..."

20. "Yeast cells were cropped with background subtraction background was subtracted from average projections in Fiji." (middle of page 21)

Response: changed to Yeast cells were cropped with background subtraction from average projections in Fiji.

21. "conservative analysis" (middle of page 27)

Response: changed to conservation analysis.

22. "Screening of ES-triggered actin remodeling in yeast mutants." (Title of Extended Fig 2, page 49).

Response: change to "Characterization of ES-triggered actin cable changes in yeast mutants."

23. "actin cables in WT actin-binding protein mutants" (legend to Extended Fig 2, page 49).

Response: change to "... labeled actin cables in mutants..."

Reviewer #3 (Remarks to the Author):

This paper identifies a molecular mechanism underlying actin remodeling induced by glucose depletion in budding yeast. The authors examined several mutant yeasts and identified Spa2 as a factor required for this process. A portion of Spa2 directly interacts with actin and modulates actin biochemistry in vitro. Although the initial findings are interesting and may potentially provide an insight into our understanding of the role of actin in energy-depleted cells, interpretation of the data is not rigorous enough to support the authors' conclusion (see below). I don't recommend publication of this paper in the current form.

Major Points:

1) Most of all, I am concerned with the way the authors interpret the TIRF data. Figure 3b and 3d show an increase in the actin filaments from ATP-F-actin and ADP-G-actin, respectively. A plausible explanation for these findings is the enhanced fragmentation of F-actin induced by Spa2. This activity is quite apparent in the second and third rows of Figure 3b and in some images of Extended Fig. 4c. I would point out that in Figure 3b, the amount of G-actin at early time points is too low to support filament nucleation by any mechanisms. As long known for cofilin and ADF, the severing activity effectively increase the filament number, which is not readily distinguished from the nucleation activity. Spa2 perhaps interacts with F-actin formed from both ATP- and ADP-G-actin and enhances its fragmentation. This severing activity of Spa2 is probably ineffective against Pi-bound F-actin as in Extended Fig. 4f.

Response: We are grateful to the Reviewer for highlighting these points and providing us with an opportunity to clarify and enhance the presentation and interpretation of our results. We also value the Reviewer's insights into how a theoretically calculated low concentration of actin might enable nucleation. This is immensely helpful for us to better explain the concept of "local concentration" versus "global theoretical concentration". Additionally, it guides us to discuss how conditions of cytoplasmic freezing under energy

starvation could likely result in lower diffusion and a higher local concentration of actin, in contrast to the more fluidic cytoplasm under normal growth conditions.

We have incorporated a new TIRF control of fragmentation exclusively with Spa2, which confirms that Spa2 on its own does not cause F-actin fragmentation. This addition can be found in Extended Fig. 4c. Consequently, we explain in the following why the second and third rows of Figure 3b might give an impression of a higher degree of fragmentation into shorter filaments.

1. Spa2 could be utilizing F-actin severing to generate new actin seeds, especially considering the depolymerization occurring near the coverslip. This would result in a high local concentration of ADP-G-Actin. In the TIRF field of view, which penetrates 100-200nm in the Z direction, the ROIs of second and third rows of Figure 3b measure $22 \times 22 \mu\text{m}^2$ in XY directions, thereby resulting in a volume of $\sim 53.24 \mu\text{m}^3$ in the view of imaging. This volume would contain approximately 122 micrometers of F-actin, comprising roughly 45140 monomers, potentially reaching a maximum local concentration of $1.4 \mu\text{M}$. This concentration theoretically is sufficient to drive local nucleation.

Moreover, we have expanded our discussion to further reconcile the observations from these in vivo TIRF-based actin assays with the in vivo conditions experienced by cells under energy starvation. Previous research has shown that energy deprivation decreases cytoplasmic diffusion in yeast, which significantly contributes to protein phase separation^{4,16}. This reduced diffusion under conditions of energy starvation also aids in maintaining a higher local concentration of actin if depolymerization occurs near the actin cable. This is likely to increase the local actin concentration and facilitate Spa2-mediated nucleation of ADP-G-actin. A weak association of Spa2 with F-actin might facilitate filament cracking mediated by Cof1. However, this scenario would necessitate a detailed structural investigation at a high resolution, which is beyond the scope of this study.

2. Moreover, we quantified the actin severing events as shown in Figure 3b. We found no significant difference in the severing efficiency over time after

introducing Spa2, when compared with the Cof1 control. This suggests that Spa2 does not enhance the fragmentation of actin filaments.

Figure R3. Rate of actin filament severing by Cof1 and Spa2. (a) Quantification of cumulative filament breaks per micron of filament at each time point, as shown in the TIRF assays from Figure 3b. Each data point is an average derived from 15 filaments. For part b, we determined the maximum severing rates for each condition by averaging the slopes of the curves taken from the time interval between 40 and 80 seconds. The error bars represent the standard deviation (\pm S.D.) around the mean.

3. In addition, with respect to the original Extended Fig. 4f, an actin nucleation assay starting from G-actin, neither Spa2(1-535) nor Spa2 (281-535) induces fragmentation.

In summary, our newly added data permit us to rule out direct severing activities by Spa2. The rough calculations provided above reasonably explain how a high local concentration can support the nucleation of ADP-G-actin. In the revised manuscript, we also briefly discuss the potential for Cof1-mediated severing promotion, while avoiding excessive speculation.

I would also emphasize that "ATP-F-actin" has the same properties as ADP-F-actin after several minutes of polymerization and the release of Pi. Difference with "ATP-F-actin" and "ADP-F-actin" in this paper can be attributed to the different concentrations (0.5 μM versus 3 μM) of F-actin but not the initial nucleotide states. There is no 'memory' for initial nucleotide states in the filament. Indeed, Figure 6a shows that phalloidin, which can differentiate the structures of ADP-F-actin and ADP-Pi-F-actin (ref. 20 and its cited Curr Biol 2015, 25:2057-2062), allows the bundling of F-actin by Spa2. This shows that

phalloidin stabilizes the ADP-F-actin structure in the filaments made from "ATP-F-actin". Thus, Spa2 binds and bundles F-actin regardless of the initial nucleotides. A plausible reason for the negative data throughout this study would be the low concentration of "ATP-F-actin". The lack of consideration of F-actin fragmentation by Spa2 and its effects on filaments in the bundling assay hampers the publication of this paper in the current form.

Response: We have corrected the wrongly cited reference. Thanks for pointing this out.

Regarding the fragmentation of F-actin by Spa2 and concerns about a low concentration of "ATP-F-actin", we have clarified the points raised in the earlier question and provide further explanations below.

Firstly, the Reviewer's point that "Spa2 binds and bundles F-actin regardless of the initial nucleotides" only applies to a drug-treated condition wherein the fixed D-loop conformation directly determines binding. Phalloidin transitions the open D-loop conformation of "ATP-F-actin" into a closed D-loop conformation, mimicking the ADP-F-actin conformation. Thus, the D-loop conformation is key to distinguishing binding. The relationship between nucleotide-binding and the D-loop conformation of native non-drug treated actin is well documented²¹. Phalloidin or JASP does not alter the nucleotide. These drugs change the D-loop conformation and confirm how a nucleotide-specific D-loop conformation differentiates Spa2 bundling. To address a concern of actin filament concentration in the crosslinking assay, we performed an experiment by using the same concentration of F-actin (with 0.1 μ M actin), adding phalloidin before and after polymerization, only phalloidin-stained F-actin aged with closed D-loop conformation can be crosslinked by Spa2(1-535). Our conclusions describe the physiologically relevant nucleotide-specific binding event, while drug-treated conditions support the D-loop conformation specificity. During energy starvation in yeast, neither phalloidin nor JASP are involved. To avoid confusion or misleading interpretation, we have clarified the purpose of using these drugs in our results.

Figure R4. Spa2(1-535) crosslinks F-actin in closed D-loop conformation. 0.1 μ M F-actin was performed by adding phalloidin before and after polymerization, incubated with in the presence or absence of 5 μ M Spa2(1-535).

Secondly, we have further clarified the fragmentation of Spa2 in the preceding question with additional experimental data and expanded explanations.

Thirdly, we wish to clarify that the negative result on ATP-G-actin nucleation is not due to a low concentration of G-actin used. We utilized 500 nM ATP-G-actin, an appropriate concentration to examine the barbed-end nucleation activity of the nucleator. This is lower than the critical concentration at the pointed end but higher than at the barbed end. A 500 nM concentration of ATP-G-actin is sufficient to investigate actin nucleation by various nucleators, as published in several references²²⁻²⁴.

Different concentrations used for "ATP-F-actin" and "ADP-F-actin" are based purely on empirical evidence from numerous prior studies^{25,26}. To address concerns about nucleation, we performed an ADP-actin nucleation assay close to its barbed-end critical concentration. We conducted a new TIRF assay using 0.8 μ M ADP-actin, significantly lower than the other 3 μ M conditions. Despite this, we still observed robust seed generation and elongation enhancement by Spa2-(281-535) and Spa2(1-535). These results are now included in Extended Fig. 4e-g. To address a concern in the low concentration for crosslinking ATP-F-actin, we performed an experiment by using 3 μ M ATP-F-actin in the absence and presence of 5 μ M Spa2(1-535). Spa2(1-535) has no crosslinking effect on ATP-F-actin. We have now added this data in the new Extended Fig. 5f All these results suggest that Spa2 specifically binds to actin filaments with a closed conformation of D-loop.

2) A highly related study which first described the rapid glucose deprivation-induced loss in the yeast actin polarity (MBoC 2004, 15:1544) is not cited.

Response: we have now added the missed citation on Page 2, line 41.

3) A finer time course of ATP-biosensor output and actin morphological changes than in Figure 1 must be shown. Which comes first, ATP depletion, actin cable remodeling or Spa2 translocation? Which is the most persistent?

Response: We have endeavored to conduct a time-course experiment to differentiate the sequential order of Spa2 crosslinking, actin cable remodeling by Spa2 and ATP changes by Queen within two minutes, using 30-second intervals. However, discerning their temporal sequence has proven to be incredibly challenging. Within the first 30 seconds, all three events appear to engage simultaneously, which is already at the limit of feasible time management when it comes to adding samples and commencing imaging on the microscope. If the reactions occur simultaneously in a dose-dependent manner within these few seconds, then it would be impossible to resolve. Furthermore, if there is a sequential effect within a few seconds, our current microscopy imaging system would not be capable of accurately capturing such minute temporal variations. Below we provide the results we obtained.

Figure R5. Time-course imaging of ATP fluctuations, actin cable bundling, and the recruitment of Spa2 onto the actin cable during energy starvation.

4) It is also very important to show the exact ATP concentration. The original paper (JCS 132: jcs230649) did some estimation but the true relationship between the “Queen ratio” and the ATP concentration was not directly determined. Calibration of this relationship

using the current microscope or measurement of the ATP/ADP ratio in soluble G-actin (isolatable using DNase I resin, ex. Plant Physiol. 100:1716) will signify this paper.

Response: Indeed, JCS 132: jcs230649 did not systematically examine the relationship between ATP concentration and QUEEN ratio. The quantitative relationship between the "QUEEN ratio" and the ATP concentration was originally determined by measuring the *in vitro* ATP gradient with a recombinant protein²⁷. Upon the suggestion, we performed the measurement now based on the reported *in vitro* calibration, the ATP level is proportional to Queen ratio from 0.6 to 2. Our estimated range of ATP concentration is around 3.7 mM, which is close to the reported ATP concentration *in vivo*²⁸.

For the suggested alternative approach in "...or measurement of the ATP/ADP ratio in soluble G-actin...", we feel it might not be practical to address the underlying point. The nucleotide-specific actin remodeling reaction was meticulously characterized by *in vitro* biochemistry. Our *in vivo* cellular assay aims to underscore the importance of such a mechanism by quantitatively describing the changes in cytosolic ATP levels. The quantitative ATP/ADP measurement on *in vivo* F-actin extends beyond the scope of our study, and is an extremely challenging task that is not practically feasible to experiment. The 1992 paper (extractable using DNase I resin, e.g., Plant Physiol. 100:1716) isolated the ATP-bound actin, which does not permit a quantitative estimation of ATP/ADP ration. In addition, this is also highly challenging for us to process the cell and actin samples from such a rapid shift in a few minutes while locking the actin nucleotide status along the long processing time and still achieve a precise measurement of ATP/ADP ratio.

5) Include supplementary timelapse movies for the images shown in Figure 1a, 1c, 1e, 1h, 1j, 2b and 2e.

Response: Thanks for the suggestion, we have now added the timelapse movies as the supplementary movies S1-S10.

6) Statistical analysis methods are unclear in places. The method section does not

provide sufficient information. In Figures 1b, 1d, 1f and 2, for example, the data are not presented with appropriate statistical information.

Response: We appreciate the Reviewer's suggestion and have now included detailed statistical analysis information in all the figure legends.

7) The y-axis label is often confusing. For example, in Figure 1b, clearer wording such as 'the fold increase in the mean intensity on the filament' is recommended. It will be difficult for many readers to realize that the 'ratio' is shown, which is indicated only in the parenthesis. Besides, why do the authors use "Mean intensity per filament"? This value may change by whether a focal plane contains strongly label actin cables or not. Such errors should be avoided. Another example is Figure 1d. "Queen ratio" is confusing. It is not immediately clear whether the "ratio" indicates the FRET ratio (as used in the original paper) or its ratio between glu(-) and glu(+). Clarify these problems on the graph as well as in the figure legends.

Response: We appreciate the Reviewer's suggestion and have revised the labeling accordingly. Thank you for drawing attention to this critical aspect of imaging and image analysis. We took this into careful consideration during imaging, consistently choosing the middle plane of the cell and capturing a total of 7.5 μm in the Z-direction with a step size of 0.25 μm , taking into account Nyquist sampling to ensure that the step size is at least 2.3 times smaller than the object being resolved. Then we generated Z-projected images from an equal number of slices using average intensity. Our imaging conditions allowed us to encompass the entire cell and actin filament for comparison.

8) In Figure 2a, the data must be presented by absolute anisotropy values.

As suggested, we have replaced Figure 2a with the original anisotropy profile with absolute value.

9) In Extended Fig. 4b, Spa2 281-535 indeed changes anisotropy around 10 μM ATP-actin, contrary to the authors' statement in p6.

Response: Given the low signal in binding, fluctuations may lead to some confusion. To clarify this, we've repeated the experiment and updated in the Extended Fig. 4b, which consistently supports our original conclusion.

10) Show evidence that no filaments were formed or contaminated during the anisotropy measurement (Figure 2a and Extended Fig. 4b).

Response: We used a non-polymerizable variant of *S. cerevisiae* Act1 (NP-actin) by introducing three mutations (D286A/V287A/D288A) and expressed it in insect cells. This ensures that it will not form filaments and become contaminated. We have also clarified this in the method.

11) Figure 2a data cannot be interpreted as showing the actin interacting portion because some Spa2 probes may just not have labeled residues that change anisotropy upon actin binding. This issue can be addressed by examining competition between Spa2 fragments in anisotropy.

Response: We apologize for not detailing this in the methods section. Our results have already factored in the reactions occurring in the absence of actin, considering the potential interference between labelled and unlabelled Spa2. We have now incorporated this clarification in the methods section.

12) Although timelapse imaging of actin assembly and disassembly using TIRF is useful to visualize elongation and severing reactions, it does not accurately report the amount of actin filaments and nuclei formed. In the TIRF assay, it is not readily apparent how much of the filament in the solution is trapped on the glass surface. Reanalysis of several key biochemical properties of Spa2 with the pyrene actin assay is strongly recommended. In addition, describe the labeling method for fluorescein Alexa 488-labeled Spa2 in the method section.

Response: We have now conducted a pyrene assay that have been added to Extended Fig. 4 h.

13) What is “nonpolymerizable monomeric actin (Hypermol, Germany)” in the method?

Response: We apologize for the incorrect citation. The non-polymerizable actin we used is *S. cerevisiae* Act1 (NP-actin), which was engineered by introducing three mutations (D286A/V287A/D288A). This was used in our previous publication¹⁷, and the mutation strategy was originally developed for *Drosophila* actin²⁹. We have now clarified this in the results (page 7) and methods (page 23-24).

14) Remove Extended Fig. 6 which does not show any evidence that this portion is strongly conserved in fungi Spa2-related proteins. In addition, AlphaFold is not reliable in predicting the structure of unfolded peptides.

Response: We agree that AF2 cannot predict precise folding and conformation at a single-residue resolution in the Intrinsically Disordered Region (IDR). We have clarified the purpose of this figure to show the co-existence of the folded 281-535 region and its disordered N-terminus, without intending to claim exact protein structures. We believe that such an evolutionarily conserved partnership is important to understand dynamic macromolecular assembly, which requires both the IDR and the folded catalytic core.

15) Significance of Fig. 4 is unclear for this study as there is no Spa2 condensates along the yeast actin cable. Remove this to the supplement or examine the effects of a wide range of Spa2 concentrations on actin biochemistry.

Response: We apologize if our initial explanation wasn't comprehensive enough or didn't effectively communicate with scientists from diverse backgrounds. To provide further clarity, we've expanded our discussion on the significance of this figure and have included these additions on pages 9 and 17 of the revised manuscript.

Biomolecular condensation is an interdisciplinary field, intertwining the principles of biophysics, biochemistry, and cell biology to unravel the functionalities of complex molecular assemblies. The biophysical characterization of these molecular condensations plays a crucial role in outlining the progressive stages of molecular assembly and their

corresponding biochemical functions. It's important to note that most nanoscale functional condensates often surpass the normal microscopic XY-resolution and exhibit connected patterns when they are densely packed. This is particularly noticeable when there is wetting-supported connectivity between various condensates^{5,15,30,31}.

In many studies on phase separation, a common experimental challenge is the inability to clearly delineate the pattern of multivalent interactions within the assemblies. Instead, these works often show micro- or meso-scale droplets. Our Figure 4 significantly contributes to this discourse by dissecting how a weakly interacting IDR, in combination with a folded dimeric domain as the biocatalytic core, can assemble biomolecules into molecular condensates in a dose-dependent manner. This is one of the most critical experiments in this work to understand how a progressive assembly and recruitment of Spa2 onto the surface of F-actin would eventually assemble into a wetting-like multivalent assembly for F-actin crosslinking.

Spa2 has indeed been shown to decorate F-actin *in vivo*. However, due to the limitations of our resolution of around 110-140nm with the spinning-disk-confocal coupled with a high-resolution system, we are unable to directly characterize the condensation *in vivo*. This is why we have performed *in vitro* AFM experiments (Fig. 5e), which provide a nanometer resolution in the Z-direction to investigate how Spa2 can condense and spread on the F-actin surface. A similar approach was also recently used by Petry lab in Princeton to study TPX2 complex's wetting on microtubules^{32,33}

Minor Points:

16) Although Abp140 was used for quantification of F-actin in live yeast cells, full-length Abp140 has an F-actin bundling activity *in vitro* (Oncogene 16:121) and hence, may bind F-actin cooperatively and in a non-linear fashion. Confirmation of the key findings in Figure 1 using fluorescent phalloidin is recommended.

Response: We appreciate the Reviewer's suggestion. We had the results from phalloidin staining in Extended Fig. 3.

17) Include a summary and a graphical explanation for the known domain functions and interacting molecules of Spa2 (in Figure 2a).

Response: We have included the suggested information in Extended Fig.3a now.

18) Describe in detail how actin solutions were diluted with TIRF buffer before imaging and what is the final constituents during imaging.

Response: We initiated our experiment by preparing a 2X TIRF buffer, which contained 20 mM imidazole, 100 mM DTT, 30 mM glucose, 100 mM KCl, 2 mM MgCl₂, 2 mM EGTA, 200 mg/ml glucose oxidase, 80 ug/ml catalase and 0.5% methylcellulose (4000 cP), 0.6 mM ADP or ATP, adjusted to pH 7.4. We rinsed the chamber with a 1X dilution of the TIRF buffer. Simultaneously, we prepared a mixture of G-actin and Spa2 proteins in G-buffer, this mixture was added to the chamber along with the 2X TIRF buffer, and carefully mixed thrice before proceeding with image capture.

19) Did TIRF buffer contain "50 mM" DTT? Doesn't it affect actin polymerization and disassembly?

Response: Our TIRF buffer includes DTT, which is based on the protocols established by the Pollard and Goode labs^{25,34}.

20) Clearly indicate in the method section which data was obtained by using "Total internal reflection fluorescence microscopy" and "In vitro actin polymerization and imaging".

Response: We have now clarified them in detail in the method.

21) "Whereas phalloidin stabilizes the D-loop in a closed conformation, jasplakinolide (JASP) inhibits phosphate release, ..." in p12: This sentence needs correction. Phalloidin also inhibits phosphate release (Biochimica et Biophysica Acta 1990, 1035:197-200)

Response: We appreciate the reminder. We've revised our wording to clarify, stating, "Both phalloidin and JASP inhibit phosphate release. However, while phalloidin stabilizes the D-loop in a closed conformation, JASP locks the D-loop in an open state." This correction has been updated on page 14.

As we previously explained, the preference of Spa2 for ADP-actin over ATP-actin is due to the closed D-loop conformation in ADP-actin under physiological conditions. When under drug-treated conditions such as with phalloidin, as long as the D-loop remains closed, the inhibited release of ADP:Pi doesn't interfere with Spa2 binding. This is distinct from the native ADP:Pi-actin under physiological condition, which possesses an open D-loop conformation.

Reviewer #4 (Remarks to the Author):

Review of Nature Communications 414160

In the manuscript "Spa2 remodels ADP-actin via molecular condensation under glucose starvation", Ma and colleagues identify the protein, Spa2, responsible for actin bundle formation that is observed shortly after glucose and energy starvation of yeast. Through the detailed experiments, the authors discovered that Spa2 can nucleate ADP-G-actin filaments and protect ADP-actin filaments from severing by cofilin. Spa2 modulation of ADP-actin monomers and filaments depends on the actin D-loop. Furthermore, the authors implicate condensation of Spa2 as a key mechanism by which it can modulate both filament nucleation, stabilization, and bundling. These results are exciting and provide insight into a new mechanism by which actin dynamics can be regulated in cells.

Importantly, this study is well suited for the wide readership of Nature Communications. The manuscript will be of interest for researchers in stress response, actin regulation, and condensate biology, as well as those interested in fungi biology, as Spa2 is well conserved. Below are comments for the authors to consider while revising their manuscript. If these can be addressed, this Reviewer will support publication of this study in Nature Communications.

Comments:

1) In the abstracts, the authors should highlight the new role for Spa2 that they have discovered. Perhaps they can contrast this new role with a few words describing what was previously understood about Spa2.

Response: We appreciate the Reviewer's excellent suggestion. We have now incorporated these highlights into our Abstract.

2) In the third line on page 3 of the current version of the manuscript, the authors write ADP:ATP ratio. For consistency with the rest of the manuscript, they should use ATP:ADP.

Response: We apologize for the confusion. We have now changed them to “decrease in ATP level”.

3) In Figure 1 and associated methods, the authors quantify various aspects of actin bundling. It would be helpful in either the results or methods to comment on the thresholds used for identifying quantifiable filaments. Was there an intensity or length of filament that served as a cut off, below which structures were not analyzed?

Response: We appreciate the Reviewer's suggestion, and we have now incorporated more detailed methodologies in our revised manuscript. We conducted an unbiased selection process for all the observed actin cables.

4) In Figure 1E, why does the WT image appear to have similar bundling as GS after 5 minutes?

Response: In Figure 1e, the top panel represents conditions with the presence of glucose, while the bottom panel shows conditions where glucose is absent. In the case of the wild type, an increase in the actin bundle was observed within 5 minutes of initiating glucose starvation.

5) For experiments shown in Figure 2, were the Spa2 variants expressed in a Spa2 delta background, or is wild-type Spa2 also present in these cells?

Response: All the Spa2(1-535)-GFP and Spa2(1-281)-GFP variants are expressed under native SPA2 promoter and were investigated in the background of SPA2 deletion. We have clarified these in the method.

6) In Figure 2, are Spa2 variants expressed equally? One may imagine that some of the observed effects may be due to expression differences. Because they are fluorescently tagged in initial experiments, fluorescence intensity per cell can be used as an expression calibration. However, in experiments using unlabeled Spa2 variants and ABP-GFP, the authors should include data from western blots showing what the expression is across

the population of cells expressing each variant. If the expression levels are different, the authors should include a comment about what impact this may have on cellular response to GS and ES, especially since phase separation is invoked later in the manuscript.

Response: We are grateful for the Reviewer's suggestion. A quantitative comparison of local concentrations would indeed provide a more accurate understanding of the functionality of each Spa2 variant, something not directly achievable through a cell lysate-based western blot approach. Accordingly, we have now incorporated fluorescence intensity quantification data in Extended Figure 3. When compared to the concentration of Spa2 at the bud tip, we found that the concentrations of Spa2(1-535) on the actin filament and Spa2(1-281) in the cytosol are similar, approximately 1 μ M. This concentration is sufficient for actin bundling in crowded conditions, as shown in the new Extended Figure 5. However, the concentration of Spa2(281-535) in puncta is significantly lower, at 0.3 μ M. We attribute this lower local concentration to the decreased oligomerization status of Spa2(281-535) compared to Spa2(1-535), stemming from the absence of multivalent IDR. We have now included this information and interpretation on page 6 of our revised manuscript

7) In Figure 4, since the N-terminal IDR contributes to condensation in the following assays, why is no assembly detected in these assays? It would be interesting to see if AUC at higher concentrations induced self-assembly of the 2-281 fragment. This data and the microscopy data in Figure 4 suggest that it is both N-terminal and dimerization of 281-535 that drive condensate formation. The title of this section makes it seem that the N-terminal alone drive phase separation, but there seems to be contribution from the entire protein. The title of this section should be corrected.

Response: thanks for pointing this out. We totally agree. We have now revised the title to "Molecular condensation of Spa2 depends on the valency derived from its N-terminal IDR"

8) The final sentence in the top paragraph of page 8 of the current version of the manuscript is a bit confusing. If there is no obvious self-association, how is there dimerization?

Response: Sorry for the confusion. This statement is in reference to the interaction between Spa2(281-535) dimers. We have now rephrased it for clarity.

9) In the first sentence of the first full paragraph on page 9, the authors refer to microtubule wetting. This has also been observed for LAT condensate proteins on actin filaments, PMID 31268421.

Response: We now have added this citation.

10) The authors refer to mechanical stabilization of actin filaments preventing cofilin severing. The concept of mechanical stabilization should be clarified a bit for the reader. Is the intended description that the filament is super stable and thus cannot be severed or that Spa2 coats the filament and prevents cofilin binding and severing. In the former, cofilin can still access the filament. In the latter, cofilin cannot access the filament. It's probably a combination of both, so a more detailed description is warranted.

Response: We appreciate this great point from the reviewer. Indeed, many crosslinking mechanisms previously explored typically employ folded-domain-based binding methods akin to those observed in fimbrin-like proteins. It is probable that the mechanisms for F-actin bundles generated by flexible intrinsically disordered regions (IDRs) differ when faced with cofilin-mediated depolymerization. This antagonistic effect could be attributed to either the stabilization of bundles, thereby reducing cofilin's accessibility, or the result of competition over actin-binding sites. Our study posits that the former is more likely, given that 500 nM Spa2 did not prevent 50 nM Cof1-mediated severing (Figure 3b), suggesting that the binding is not competition-based. We added such clarification on page 12 now.

11) In the 'In vivo actin cable imaging and image analysis, a sentence states that Yeast cells were cropped with background subtraction background was subtracted...' Remove one of the background subtractions.

Response: we have corrected it now.

12) If possible, please use magenta and green color schemes in their figures for microscopy images to improve accessibility for readers with red/green color blindness.

Response: we have updated now.

13) In Figure 4H, drpolets should be droplets.

Response: we have corrected it now.

Reference:

- 1 Carlier, M.-F. & Shekhar, S. Global treadmilling coordinates actin turnover and controls the size of actin networks. *Nat. Rev. Mol. Cell Bio.* **18**, 389-401 (2017).
- 2 Harker, A. J. *et al.* Ena/VASP processive elongation is modulated by avidity on actin filaments bundled by the filopodia cross-linker fascin. *Mol. Biol. cell* **30**, 851-862 (2019).
- 3 Chou, S. Z. & Pollard, T. D. Cryo-electron microscopy structures of pyrene-labeled ADP-Pi- and ADP-actin filaments. *Nat. Commun.* **11**, 5897 (2020).
- 4 Delarue, M. *et al.* mTORC1 controls phase separation and the biophysical properties of the cytoplasm by tuning crowding. *Cell* **174**, 338-349. e320 (2018).
- 5 Pappu, R. V., Cohen, S. R., Dar, F., Farag, M. & Kar, M. Phase transitions of associative biomacromolecules. *Chem. Rev.* (2023).
- 6 Woodruff, J. B. *et al.* The centrosome is a selective condensate that nucleates microtubules by concentrating tubulin. *Cell* **169**, 1066-1077. e1010 (2017).
- 7 Xie, Y. *et al.* Polarisome scaffold Spa2-mediated macromolecular condensation of Aip5 for actin polymerization. *Nat. Commun.* **10**, 1-18 (2019).
- 8 Xu, L. & Bretscher, A. Rapid glucose depletion immobilizes active myosin V on stabilized actin cables. *Curr. Biol.* **24**, 2471-2479 (2014).
- 9 Case, L. B., Zhang, X., Ditlev, J. A. & Rosen, M. K. Stoichiometry controls activity of phase-separated clusters of actin signaling proteins. *Science* **363**, 1093-1097 (2019).
- 10 Su, X. *et al.* Phase separation of signaling molecules promotes T cell receptor signal transduction. *Science* **352**, 595-599 (2016).
- 11 Banani, S. F., Lee, H. O., Hyman, A. A. & Rosen, M. K. Biomolecular condensates: organizers of cellular biochemistry. *Nat. Rev. Mol. Cell Bio.* **18**, 285-298 (2017).
- 12 King, M. R. & Petry, S. Phase separation of TPX2 enhances and spatially coordinates microtubule nucleation. *Nat. Commun.* **11**, 270, doi:10.1038/s41467-019-14087-0 (2020).
- 13 Gouveia, B. *et al.* Capillary forces generated by biomolecular condensates. *Nature* **609**, 255-264 (2022).
- 14 Setru, S. U. *et al.* A hydrodynamic instability drives protein droplet formation on microtubules to nucleate branches. *Nat. Phys.* **17**, 493-498, doi:10.1038/s41567-020-01141-8 (2021).
- 15 Case, L. B., Ditlev, J. A. & Rosen, M. K. Regulation of transmembrane signaling by phase separation. *Annu. Rev. Biophys.* **48**, 465-494 (2019).
- 16 Joyner, R. P. *et al.* A glucose-starvation response regulates the diffusion of macromolecules. *elife* **5**, e09376 (2016).

- 17 Xie, Y. *et al.* Orchestrated actin nucleation by the *Candida albicans* polarisome complex enables filamentous growth. *J. Biol. Chem.* **295**, 14840-14854 (2020).
- 18 Chou, S. Z., Chatterjee, M. & Pollard, T. D. Mechanism of actin filament branch formation by Arp2/3 complex revealed by a high-resolution cryo-EM structure of the branch junction. *Proc. Natl Acad. Sci. USA* **119**, e2206722119 (2022).
- 19 Jansen, S. *et al.* Single-molecule imaging of a three-component ordered actin disassembly mechanism. *Nat. Commun.* **6**, 7202 (2015).
- 20 Merino, F. *et al.* Structural transitions of F-actin upon ATP hydrolysis at near-atomic resolution revealed by cryo-EM. *Nat. Struct. Mol. Biol.* **25**, 528-537 (2018).
- 21 Pospich, S., Merino, F. & Raunser, S. Structural effects and functional implications of phalloidin and jasplakinolide binding to actin filaments. *Structure* **28**, 437-449. e435 (2020).
- 22 Xie, Y. *et al.* Polarisome scaffolder Spa2-mediated macromolecular condensation of Aip5 for actin polymerization. *Nat. Commun.* **10**, 5078 (2019).
- 23 Sun, H. *et al.* *Xanthomonas* effector XopR hijacks host actin cytoskeleton via complex coacervation. *Nat. Commun.* **12**, 4064 (2021).
- 24 Ma, Z. *et al.* Formin nanoclustering-mediated actin assembly during plant flagellin and DSF signaling. *Cell Rep.* **34** (2021).
- 25 Kovar, D. R., Harris, E. S., Mahaffy, R., Higgs, H. N. & Pollard, T. D. Control of the assembly of ATP- and ADP-actin by formins and profilin. *Cell* **124**, 423-435 (2006).
- 26 Brühmann, S. *et al.* Distinct VASP tetramers synergize in the processive elongation of individual actin filaments from clustered arrays. *Proc. Natl Acad. Sci. USA* **114**, E5815-E5824 (2017).
- 27 Yaginuma, H. *et al.* Diversity in ATP concentrations in a single bacterial cell population revealed by quantitative single-cell imaging. *Sci. Rep.* **4**, 6522 (2014).
- 28 Özalp, V. C., Pedersen, T. R., Nielsen, L. J. & Olsen, L. F. Time-resolved measurements of intracellular ATP in the yeast *Saccharomyces cerevisiae* using a new type of nanobiosensor. *J. Biol. Chem.* **285**, 37579-37588 (2010).
- 29 Zahm, J. A. *et al.* The bacterial effector VopL organizes actin into filament-like structures. *Cell* **155**, 423-434 (2013).
- 30 Miao, Y., Guo, X., Zhu, K. & Zhao, W. Biomolecular condensates tunes immune signaling at the Host-Pathogen interface. *Curr. Opin. Plant Biol.* **74**, 102374 (2023).
- 31 Brangwynne, C. P., Tompa, P. & Pappu, R. V. Polymer physics of intracellular phase transitions. *Nat. Phys.* **11**, 899-904 (2015).

- 32 King, M. R. & Petry, S. Phase separation of TPX2 enhances and spatially coordinates microtubule nucleation. *Nat. Commun.* **11**, 270 (2020).
- 33 Setru, S. U. *et al.* A hydrodynamic instability drives protein droplet formation on microtubules to nucleate branches. *Nat. Phys.* **17**, 493-498 (2021).
- 34 Johnston, A. B. *et al.* A novel mode of capping protein-regulation by twinfilin. *elife* **7**, e41313 (2018).

Reviewers' Comments:

Reviewer #1:

Remarks to the Author:

The authors have addressed my comments appropriately.

I found a few typos to still correct in the revised manuscript:

Line 67 : can you reformulate ? Not sure what « unfacilitating of splicing” means.

Line 90: please remove the “especially the yeast fimbrin Sac6”. How can a protein “especially” not do something? Do all the others actually do something?

Line 120-122: modify Spa2-535 to Spa2(1-535) and Spa2-281-535 to Spa2(281-535)

Line 246: typo in Supplementary

Line 290: typo in phosphate

Reviewer #2:

Remarks to the Author:

Ma et al. performed a number of additional experiments to address concerns raised by four referees, including myself, and included the new results into the modified version. Furthermore, they have made extensive re-wording for clarity. I am happy with the revisions regarding most of my comments, except for the following two.

1. First, the authors performed additional experiments to demonstrate that Spa2 actually nucleates ADP actin filaments. Although direct observation by TIRF was not successful, they provided another set of data that supports the hypothesis that Spa2 nucleates ADP actin polymerization (Extended Figure 4D), which is very good. Nonetheless, the situation has not been changed qualitatively, and I suggest the authors to state explicitly that the evidence for the nucleating activity is indirect, and/or direct or rigorous demonstration that Spa2 is a nucleator is a future goal. Regarding the newly added phrases regarding Extended Figure 4D, the authors describes Spa2 as an actin “polymerizer” (line 163). This is not an accepted term in the research field, and the data actually suggests that Spa2 is an actin nucleator. I suggest changing this to “nucleator”.

2. My second remaining concern is the nature of “ATP-F-actin” used in this study. The authors now state that “ATP-F-actin” is just standard actin filaments prepared by incubating ATP-G-actin in a polymerizing buffer for 30 min at room temperature. As Reviewer 3 also pointed out (latter half of his/her major point 1), most of the actin protomers after 30 min of incubation in polymerizing condition would contain only ADP even if they started with bound ATP, and this is similar to “ADP-F-actin”. There may be subtle differences between the two F-actin preparations, however. For example, unlike “ADP-F-actin”, filaments of “ATP-F-actin” would contain small “ATP-caps”. The possible contribution of ATP-caps to the observed differences between “ADP-F-actin” and “ATP-F-actin” can be examined by adding ATP to “ADP-F-actin” solution prior to the experiment. Alternatively, “ATP-F-actin” protomers may “remember” that they had bound ATP or ADP-Pi until a while ago. This is not entirely impossible (e.g., Collins et al., 2011; DOI: 10.4161/bioa.1.3.16714), although Reviewer 3 rejects such a possibility. The concentrations of monomeric actin in the assay solutions are also different between the two F-actin preparations since the critical concentrations differ, although I am not sure how the concentrations of monomeric actin could affect the assay. At any rate, the authors did not respond adequately to the nature of “ATP-F-actin” issue raised by Reviewer 3 and myself. They noted in the rebuttal “the Reviewer's point that “Spa2 binds and bundles F-actin regardless of the initial nucleotides” only applies to a drug-treated condition wherein the fixed D-loop conformation directly determines binding”. What Reviewer 3 pointed out, and what I also implied, is that regardless of the initial nucleotide state, the most of the actin protomers are in the ADP-bound state after 30 min of polymerization. This makes it unlikely that Spa2 can distinguish the nucleotide state per se. Something must be done regarding this issue before I can fully support the publication of this paper in Nature Communications.

Reviewer #3:

Remarks to the Author:

I am concerned with the author's response to review comments. Their response to some of the raised issues appears to lack scientific honesty. While the discovery that Spa2 plays an important role in energy starved yeast cells is intriguing, I don't recommend publication of this manuscript.

1) In my major points #1, I raised the enhanced fragmentation of F-actin induced by Spa in the second and third rows of Figure 3b. The response was "Moreover, we quantified the actin severing events as shown in Figure 3b. We found no significant difference in the severing efficiency over time after introducing Spa2, when compared with the Cof1 control." Their quantification is shown in Figure R3 of the Rebuttal (p22). I did reanalysis of the data in Figure 3b images and found that 1-535, 281-535 and 1-281 increased "breaks per um F-actin" by 6.4-, 1.6- and 2.4-fold at 40 s, respectively. Thus, there is a huge discrepancy between Figure 3b images and the Figure R3 graph. It is clear that Spa2 fragments indeed accelerate the cofilin-induced filament severing. I don't understand why the authors cover up their new, insightful findings.

2) I also raised an issue in point #1 that "difference (in actin bundling) with "ATP-F-actin" and "ADP-F-actin" in this paper can be attributed to the different concentrations (0.5 uM versus 3 uM) of F-actin but not the initial nucleotide states". As I stated in the same paragraph, this concern is only for the experiments employing preformed filaments (Figure 5a-c; Extended Fig. 5c and 5d). However, the authors failed to reproduce these key findings in the experiments that can compare ATP- and ADP-F-actin at the same concentration under various conditions. Only a few images with "3 uM" F-actin are now in Extended Fig. 3f. However, comparison was never made over time in the presence of various concentrations of Spa2 fragments (and PEG or Cof1). Moreover, images in Extended Fig. 3f expose a problem. The filament density is too low for the 3 uM F-actin tapped with the poly-lysine coated glass surface. I doubt that something was not accurately controlled in the experiments for Extended Fig. 3f. Other reviewers also raised a concern over the difference between ATP- and ADP-F-actin. This revision fails to sufficiently address this issue.

3) It is inappropriate to delete the previous data in Extended Fig. 4b which may be inconvenient for the authors' conclusion and replace it with a new but still low-quality data "which, the authors state (in Rebuttal, p28), consistently supports our original conclusion". The authors should not delete the original data in the revised manuscript nor deliberately present the results that best support the hypothesis.

REVIEWER COMMENTS

Reviewer #1 (Remarks to the Author):

The authors have addressed my comments appropriately.

I found a few typos to still correct in the revised manuscript:

Line 67 : can you reformulate ? Not sure what « unfacilitating of splicing” means.

Line 90: please remove the “especially the yeast fimbrin Sac6”. How can a protein “especially” not do something? Do all the others actually do something?

Line 120-122: modify Spa2-535 to Spa2(1-535) and Spa2-281-535 to Spa2(281-535)

Line 246: typo in Supplementary

Line 290: typo in phosphate

Response: we appreciate your positive feedback. In response, we have made the modifications according to the reviewer's suggestions.

- Line 83 : “unfacilitating of splicing” is changed to “in which a splicing of cytochrome B oxidase preRNA is disabled”
- “especially the yeast fimbrin Sac6” is removed
- All correction are made for Spa2 names within Line 137-139.
- Line 270: typo corrected
- Line 314: typo corrected

Reviewer #2 (Remarks to the Author):

Ma et al. performed a number of additional experiments to address concerns raised by four referees, including myself, and included the new results into the modified version. Furthermore, they have made extensive re-wording for clarity. I am happy with the revisions regarding most of my comments, except for the following two.

Response: we very much appreciate Reviewer #2’s very positive feedback of our last extensive revision. We also appreciate Reviewer #2 for providing very constructive comments on how to best phrase our words, and pointing out the following two more points to be clarified.

1. First, the authors performed additional experiments to demonstrate that Spa2 actually nucleates ADP actin filaments. Although direct observation by TIRF was not successful, they provided another set of data that supports the hypothesis that Spa2 nucleates ADP actin polymerization (Extended Figure 4D), which is very good. Nonetheless, the situation has not been changed qualitatively, and I suggest the authors to state explicitly that the evidence for the nucleating activity is indirect, and/or direct or rigorous demonstration that Spa2 is a nucleator is a future goal.

Response: We appreciate the recommendation from Reviewer #2. We acknowledge the need for caution in drawing conclusions about the ADP-nucleator, especially in the absence of direct

evidence for the barbed-end association by Spa2 due to technical limitations. We have now clearly mentioned this and emphasized the necessity for further confirmation in subsequent studies on line 350, page 17.

2. Regarding the newly added phrases regarding Extended Figure 4D, the authors describes Spa2 as an actin “polymerizer” (line 163). This is not an accepted term in the research field, and the data actually suggests that Spa2 is an actin nucleator. I suggest changing this to “nucleator”.

Response: We have substituted the term "polymerizer" with "nucleator" as it is a more appropriate term.

3. My second remaining concern is the nature of “ATP-F-actin” used in this study. The authors now state that “ATP-F-actin” is just standard actin filaments prepared by incubating ATP-G-actin in a polymerizing buffer for 30 min at room temperature. As Reviewer 3 also pointed out (latter half of his/her major point 1), most of the actin protomers after 30 min of incubation in polymerizing condition would contain only ADP even if they started with bound ATP, and this is similar to “ADP-F-actin”. There may be subtle differences between the two F-actin preparations, however. For example, unlike “ADP-F-actin”, filaments of “ATP-F-actin” would contain small “ATP-caps”. The possible contribution of ATP-caps to the observed differences between “ADP-F-actin” and “ATP-F-actin” can be examined by adding ATP to “ADP-F-actin” solution prior to the experiment. Alternatively, “ATP-F-actin” protomers may “remember” that they had bound ATP or ADP-Pi until a while ago. This is not entirely impossible (e.g., Collins et al., 2011; DOI: 10.4161/bioa.1.3.16714), although Reviewer 3 rejects such a possibility.

Response: We are grateful for the insightful comments from Reviewer #2 about the possible composition differences between “ATP-F-actin” and “ADP-F-actin” when polymerized over extended periods. Here, the critical point is to determine if the observed incompatibility between Spa2 and the 30-minute polymerized “ATP-F-actin” stems from it being purely ATP-F-actin or if other factors deter Spa2 from associating with this form of actin. We performed new experiments to exam the points in detail that aim to provide solid conclusion.

First, aim to validate this by producing “ATP-F-actin” within a much shorter timeframe, trying to keep as much as ATP-F-actin possible, prior to introducing Spa2, we performed experiments by polymerizing ATP-F-actin for 2 min before adding Spa2(1-535). We did not observe any actin bundling, both in the absence or presence of 5% PEG, when incubated with Spa2(1-535) at the indicated concentration. We've included this new result from the 2-minute observation in the Extended fig5c, d. Nevertheless, it's plausible that at the 2-minute mark, this ATP-F-actin could still be mixed with actin in other nucleotide states due to rapid hydrolysis. Base on previous report, the ratio of F-ATP: F-ADP-P_i: F-ADP might be around 2:5:3¹. While we can not accurately determine the percentage of ADP-F-actin in our sample, it remains unclear why we did not detect Spa2-mediated actin crosslinking in such a 2-minute preparation. Therefore, we proceeded to prepare AMPPNP-F-actin as a more reliable non-ADP-F-actin control.

Secondly, to emulate pure ATP-F-actin, we prepared 3 μM F-actin using the slowly hydrolyzed ATP analog, AMPPNP²⁻⁸. According to previous studies, AMPPNP is stable for at least 16 hours in solution at room temperature⁸. We polymerized AMPPNP-G-actin for 5 minutes to produce

AMPPNP-F-actin before incubating it with Spa2(1-535). During this time, we expect AMPPNP not to undergo hydrolysis. In our findings, Spa2(1-535) did not crosslink with AMPPNP-F-actin. We have incorporated this new observation into Extended Fig. 5c, d.

We contend that the 2-minute fast polymerization assay and the AMPPNP-F-actin directly address the reviewer's concerns. These issues originated from our earlier 30-minute ATP-F-actin, which might have contained mixed nucleotide species, casting doubts on its reliability as a control. While we deliberated the proposed experiment of adding ATP to ADP-F-actin to discern differences between "ADP-F-actin" and "ATP-F-actin", we believe our two aforementioned tests more effectively mitigate concerns about the 30-minute ATP-F-actin potentially having ADP-actin. Thus, there's no immediate need to examine the presence of an "ATP-Cap" in the 30-minute ATP-F-actin sample.

4. The concentrations of monomeric actin in the assay solutions are also different between the two F-actin preparations since the critical concentrations differ, although I am not sure how the concentrations of monomeric actin could affect the assay.

Response: In our monomer actin nucleation assay, we adjusted the concentrations based on differing critical concentrations. In our initial submission, we used various concentrations of G-actin, but in the last revision, we added an additional low-concentration nucleation assay for ADP-actin to address the concern of G-actin for nucleation assay.

Here, from this comment regarding F-actin, we've realized there's another unsolved confusion regarding the monomeric actin concentration used in the F-actin binding and bundling assay, stemming from our insufficient clarity on the concentration employed. We sincerely apologize for any misunderstandings and offer the following clarification, supported by additional experiments.

In our experiment, all the different F-actin populations compared for their interaction with Spa2 were uniformly diluted to a final same concentration, in equivalent to 0.05 μM G-actin concentration. We have now highlighted this detail in the methods section on line 708, page 33. Because we have always used same F-actin concentration for comparison, thereby we have mistaken the reviewer's point in the last revision about the "same polymerization concentrations". Here, we repeated experiments and used the same concentration of 5 μM ATP-G-actin and 5 μM ADP-G-actin to polymerize ATP-F-actin and ADP-F-actin respectively, ensuring a uniform F-actin concentration during polymerization and microscopic imaging. For the incubation with Spa2 truncating variants, both ATP-F-actin and ADP-F-actin at a concentration of 3 μM were used. Next, during the imaging, to avoid crowding, ADP-F-actin/Spa2 and ATP-F-actin/Spa2 were diluted to a concentration equivalent to 0.05 μM G-actin. Now, we have made a clear clarification on line 708-730 page 33-34. We believe now we have captured all the possible means the Reviewer #2 asks.

5. At any rate, the authors did not respond adequately to the nature of "ATP-F-actin" issue raised by Reviewer 3 and myself. They noted in the rebuttal "the Reviewer's point that "Spa2 binds and bundles F-actin regardless of the initial nucleotides" only applies to a drug-treated condition wherein the fixed D-loop conformation directly determines binding". What Reviewer 3 pointed out, and what I also implied, is that regardless of the initial nucleotide state, the most of the actin protomers are in the ADP-bound state after 30 min of polymerization. This makes it unlikely that Spa2 can distinguish the nucleotide state per se. Something must be

done regarding this issue before I can fully support the publication of this paper in Nature Communications.

Response: This concern aligns with point #3, which questioned the suitability of the 30-minute ATP-F-actin as a control due to its potentially mixed nucleotide population. Such a mix could make it less than ideal for distinguishing nucleotide states, especially for evaluating Spa2's specific function on ADP-F-actin. In response to this and similar to our reply to point #3 from reviewer 2, we have conducted additional experiments and generated new results to directly address this issue.

First, aim to validate this by producing "ATP-F-actin" within a much shorter timeframe, trying to keep as much as ATP-F-actin possible, prior to introducing Spa2, we performed experiments by polymerizing ATP-F-actin for 2 min before adding Spa2(1-535). We did not observe any actin bundling, both in the absence or presence of 5% PEG, when incubated with Spa2(1-535) at the indicated concentration. We've included this new result from the 2-minute observation in the Extended fig5c, d. Nevertheless, it's plausible that at the 2-minute mark, this ATP-F-actin could still be mixed with actin in other nucleotide states due to rapid hydrolysis. Base on previous report, the ratio of F-ATP: F-ADP-P_i: F-ADP might be around 2:5:3¹. While we can not accurately determine the percentage of ADP-F-actin in our sample, it remains unclear why we did not detect Spa2-mediated actin crosslinking in such a 2-minute preparation. Therefore, we proceeded to prepare AMPPNP-F-actin as a more reliable non-ADP-F-actin control.

Secondly, to emulate pure ATP-F-actin, we prepared 3 μM F-actin using the slowly hydrolyzed ATP analog, AMPPNP²⁻⁸. According to previous studies, AMPPNP is stable for at least 16 hours in solution at room temperature⁸. We polymerized AMPPNP-G-actin for 5 minutes to produce AMPPNP-F-actin before incubating it with Spa2(1-535). During this time, we expect AMPPNP not to undergo hydrolysis. In our findings, Spa2(1-535) did not crosslink with AMPPNP-F-actin. We have incorporated this new observation into Extended Fig. 5c, d.

Additionally, we want to emphasize further evidence supporting our findings: the G-actin assay. This assay distinguishes between polymerization initiated by ATP-G-actin and ADP-G-actin. By conducting it immediately after sample preparation, we mitigate concerns related to substantial nucleotide species conversion.

Reviewer #3 (Remarks to the Author):

I am concerned with the author's response to review comments. Their response to some of the raised issues appears to lack scientific honesty. While the discovery that Spa2 plays an important role in energy starved yeast cells is intriguing, I don't recommend publication of this manuscript.

Response: Thank you for sharing your feedback on our manuscript and response to our first rebuttal letter.

Firstly, I wish to express our genuine sadness upon reading the concerns raised, especially in relation to scientific honesty. If we have misunderstood any point the reviewer #3 has raised, we sincerely apologize for causing such misunderstanding. We wish reviewer to read our clarification and hear that it is disheartening to be perceived as lacking honesty, especially when our primary goal is to contribute positively and authentically to the scientific community.

Our team has approached this research with the utmost dedication, rigor, and integrity. We've always been committed to advancing our understanding of the topic at hand and have made every effort to carefully address the reviewers #3's comments thoroughly and transparently in the first response letter with full respect, which includes 15 major comments and 6 minor comments from the Reviewer #3.

If there are specific points of contention or instances where our responses may have seemed inadequate, we'd appreciate an opportunity to address them more directly or provide further clarification. We value the review process and believe that constructive dialogue is essential for the advancement of science.

We earnestly hope to alleviate any lingering concerns. To address the specific issue at hand:

First, if such misunderstanding stems from our substitution of the original Extended Fig. 4b with results from new experiments, we sincerely regret any confusion caused. Our primary intention behind presenting the new data, which includes three biological replicates, was to offer robust and clear results. These findings are not tailored to any hypothesis but reflect the real outcomes of our experiments. Every data point, from the initial three biological replicates to the three additional ones provided in the first revision, stems from genuine experiments. We have no intention of concealing any information. Should this have inadvertently caused significant concerns, we apologize. For clarity and thoroughness, we did another additional 3x biological replicates and now amalgamated the results from all nine biological replicates as the current final Extended Fig. 4b. We hope the reviewer finds that this is an acceptable way now.

Second, if such misunderstanding associated with the observation that Spa2 may enhance cofilin-induced filament severing: we genuinely appreciate your keen observation. However, we'd like to clarify that this was an aspect we hadn't initially aware of in such a light. In our previous revision, we focused on proving if Spa2 possesses the severing ability as Reviewer #3 hypothesized and asked. We appreciate Reviewer #3 for highlighting another new hypothesis now that we had yet to consider, especially as our results didn't explicitly inform us in this direction. We appreciate this comment about the potential influence in cofilin-mediated severing by Spa2. To explore this possibility, we diligently revisited and performed additional repeated experiments to exam the depolymerization experiments, combining Spa2 and Cof1. We've presented detailed statistical

information and provided all our data and images from a total of three biological replicates, encompassing a minimum of 90 filaments for each condition. Instead of using a one-time point of 40s to judge the severing, our measurement selects multiple appropriate time points by measuring time-dependent effects. It ensures fair comparisons among different combinations of proteins while avoiding potential errors from a one-time point date and a few seconds difference in handling each sample's incubation and image acquisition.

For clarity, we've marked all F-actin severing events we deemed reliably countable with arrows and shown below. We hope this comprehensive analysis demonstrates our commitment to presenting robust and transparent results. We regret that the previously chosen representative image in Figure 3b might not serve the role of being a representative, causing concerns for Reviewer #3. As a corrective measure, we've replaced it with a more representative image and are sharing all original results to ensure complete transparency and withstand scrutiny.

In the following session, we provide specific clarifications to address the concerns raised.

1) In my major points #1, I raised the enhanced fragmentation of F-actin induced by Spa in the second and third rows of Figure 3b. The response was "Moreover, we quantified the actin severing events as shown in Figure 3b. We found no significant difference in the severing efficiency over time after introducing Spa2, when compared with the Cof1 control." Their quantification is shown in Figure R3 of the Rebuttal (p22). I did reanalysis of the data in Figure 3b images and found that 1-535, 281-535 and 1-281 increased "breaks per um F-actin" by 6.4-, 1.6- and 2.4-fold at 40 s, respectively. Thus, there is a huge discrepancy between Figure 3b images and the Figure R3 graph. It is clear that Spa2 fragments indeed accelerate the cofilin-induced filament severing. I don't understand why the authors cover up their new, insightful findings.

Response: We thank Reviewer's new input. In our first revision, our primary objective was to investigate whether Spa2 has the severing capability, as hypothesized and raised by Reviewer #3. Given this current possibility, we conducted comprehensive depolymerization experiments with more experiment data and critically examined this hypothesis, especially when combining Spa2 and Cof1.

We bolstered our data by increasing the number of actin filaments used for cumulative severing event quantification: from the previous n=15 to the current counts of n=93 for Cof1, n=98 for Cof1+Spa2(1-535), n=91 for Cof1+Spa2(281-535), and n=90 for Cof1+Spa2(1-281). For a more equitable comparison of severing activities, we analyzed the data at multiple timepoints: 20s, 40s, 60s, and 80s. This approach, we believe, offers a more precise evaluation than a single time point, like the 40s mark. It also gives a comparative view of the severing trend over time, minimizing potential discrepancies due to timing or sample handling. Our analysis involved manually counting the number of severing per μm of actin filament using ImageJ. Now, we provide a detailed breakdown for each filament used.

Figure R1. Single actin filament severing information by Cof1 and Spa2.
 The red arrow labeled the new severing events at each time point from 0 to 80s.

As a result, our cumulative severing data does not conclude a significant enhancement of Cof1-mediated severing by Spa2 (Fig. R2). These new data have also been incorporated into Extended Fig4d and e. If there remains a possibility, as hypothesized by Reviewer #3, that Spa2 influences Cof1's activity in F-actin severing, our current findings can not identify it. Addressing this potential influence would require further studies, which we believe fall outside the scope of our current work.

Figure R2. Analysis of Enhanced Actin Filament Disassembly by Cof1 and Spa2.

(a) TIRF assays provided measurements for cumulative filament breaks per micron at various time points. The respective filament counts are 93 for Cof1, 98 for Cof1+Spa2(1-535), 91 for Cof1+Spa2(281-535), and 90 for Cof1+Spa2(1-281). (b) The average maximal severing rates for each condition were computed by evaluating the slope averages of curves during the 20 to 80-second time frame. The displayed error bars represent mean \pm standard deviation.

Furthermore, we apologize for our previous Fig. 3b, which may have inadvertently given the impression that Spa2 can expedite the Cof1's severing. Both the earlier and current images are presented below with statistics for clarity. This oversight in providing representative images was unintentional. We hope our updated presentation offers a clearer perspective, stands up to scrutiny, and provides a comprehensive overview of our research.

Figure R3. Representative time-lapse TIRFM images of actin filament disassembly by Cof1 and Spa2. (a) Displays data from the initial submission. (b) Represent data in this 2nd revision.

2) I also raised an issue in point #1 that “difference (in actin bundling) with “ATP-F-actin” and “ADP-F-actin” in this paper can be attributed to the different concentrations (0.5 μM versus 3 μM) of F-actin but not the initial nucleotide states”. As I stated in the same paragraph, this concern is only for the experiments employing preformed filaments (Figure 5a-c; Extended Fig. 5c and 5d). However, the authors failed to reproduce these key findings in the experiments that can compare ATP- and ADP-F-actin at the same concentration under various conditions. Only a few images with “3 μM ” F-actin are now in Extended Fig. 3f. However, comparison was never made over time in the presence of various concentrations of Spa2 fragments (and PEG or Cof1).

Response: We genuinely apologize for any confusion resulting from unclear descriptions regarding actin concentrations in various experiments.

In our work, all distinct F-actin populations examined for their interaction with Spa2 were consistently diluted to an equivalent final concentration, corresponding to 0.05 μM G-actin. We acknowledge the oversight in our previous descriptions and any resulting ambiguity. Since we consistently used the same F-actin concentration for comparison, we misunderstood the reviewer's comment in the prior revision concerning "same polymerization concentrations." We've now emphasized this aspect in the methods section (line 708, page 33).

To further clarify and preempt any confusion, it seems Reviewer 3 might be referencing the need for a uniform initial G-actin concentration during F-actin polymerization. This observation likely stems from our prior lack of clarity regarding the concentrations used. To address this, we conducted experiments and used the same concentration of 5 μM ATP-G-actin and 5 μM ADP-G-actin to polymerize ATP-F-actin and ADP-F-actin respectively, ensuring a uniform F-actin concentration during polymerization and microscopic imaging. For the incubation with Spa2 truncating variants, both ATP-F-actin and ADP-F-actin at a concentration of 3 μM were used. Next, during the imaging, to avoid crowding, ADP-F-actin/Spa2 and ATP-F-actin/Spa2 were diluted to a concentration equivalent to 0.05 μM G-actin. This clarification is now explicitly mentioned in the Method on line 708-730, page 33-34.

In response to other potential queries concerning actin concentration, ATP nucleotide status, and the presence of PEG, we undertook two further experimental sets. Our objective was to present a more comprehensive ATP-actin control for the F-actin binding assay.

1, aim to validate this by producing “ATP-F-actin” within a much shorter timeframe, trying to keep as much as ATP-F-actin possible, prior to introducing Spa2, we performed experiments by polymerizing ATP-F-actin for 2 min before adding Spa2(1-535). We did not observe any actin bundling, both in the absence or presence of 5% PEG, when incubated with Spa2(1-535) at the indicated concentration. We've included this new result from the 2-minute observation in the Extended fig5c, d. Nevertheless, it's plausible that at the 2-minute mark, this ATP-F-actin could still be mixed with actin in other nucleotide states due to rapid hydrolysis. Base on previous report, the ratio of F-ATP: F-ADP-P_i: F-ADP might be around 2:5:3¹. While we can not accurately determine the percentage of ADP-F-actin in our sample, it remains unclear why we did not detect Spa2-mediated actin crosslinking in such a 2-minute preparation. Therefore, we proceeded to prepare AMPPNP-F-actin as a more reliable non-ADP-F-actin control.

2, to emulate pure ATP-F-actin, we prepared 3 μM F-actin using the slowly hydrolyzed ATP analog, AMPPNP²⁻⁸. According to previous studies, AMPPNP is stable for at least 16 hours in solution at room temperature⁸. We polymerized AMPPNP-G-actin for 5 minutes to produce AMPPNP-F-actin before incubating it with Spa2(1-535). During this time, we expect AMPPNP not to undergo hydrolysis. In our findings, Spa2(1-535) did not crosslink with AMPPNP-F-actin. We have incorporated this new observation into Extended Fig. 5c, d.

Furthermore, to address reviewer #3's concern on insufficient titration of Spa2 in Extended Fig. 3f, we have now repeated experiment and addition a total of 6 concentrations in the absence of PEG and seven concentration in the presence of PEG.

Here, we have made every effort to address the potential concerns raised by Reviewer #3 and have provided the above clarifications and additional experiments in response.

3. Moreover, images in Extended Fig. 3f expose a problem. The filament density is too low for the 3 μM F-actin tapped with the poly-lysine coated glass surface. I doubt that something was not accurately controlled in the experiments for Extended Fig. 3f. Other reviewers also raised a concern over the difference between ATP- and ADP-F-actin. This revision fails to sufficiently address this issue.

Response: First, as we have clarified now in the method (line 708-730, Page 33-34), we have diluted all 3 μM F-actin to 0.05 μM immediately before imaging to demonstrate clear filaments signal and patterns. And detailed information can also be found now in the above reply to point #2 regarding the difference between ATP- and ADP-F-actin. We have provided new experiments to address all the concern now.

4) It is inappropriate to delete the previous data in Extended Fig. 4b which may be inconvenient for the authors' conclusion and replace it with a new but still low-quality data "which, the authors state (in Rebuttal, p28), consistently supports our original conclusion". The authors should not delete the original data in the revised manuscript nor deliberately present the results that best support the hypothesis.

Response: We aimed to replace the original set with an experimental dataset that had a similar number of biological replicates to the original assay. We agree with the reviewer's viewpoint on the figure substitution. In our revised manuscript, we've incorporated results from three experimental batches and nine biological replicates, which is clearly detailed on page 61, line 1171, and in the updated Extended Fig. 4b. If our new data presentation in last revision is taken as potential "scientific dishonesty," we stress that our actions were rooted in sincerity. We never intended to alter data to fit a pre-established narrative. We're grateful for the chance to address this issue and have now included all the original raw data from the three sets and nine biological replicates for verification.

Figure R4. Anisotropy Measurements of Spa2-Truncations Interacting with ATP-Actin Monomers.

a, Displays the raw data from the initial submission. b, Illustrates the raw data from the 1st revision. c, The results from the third repeated experiment in this 2nd revision. Spa2 truncations were tagged with Alexa 488 and introduced to escalating concentrations of ATP-actin. The mean values, accompanied by an error bar indicating \pm SD, were derived from three biological replicates and plotted using the Hill slope equation.

Reference:

- 1 Carrier, M. & Pantaloni, D. Direct evidence for ADP-inorganic phosphate-F-actin as the major intermediate in ATP-actin polymerization. Rate of dissociation of inorganic phosphate from actin filaments. *Biochemistry* **25**, 7789-7792 (1986).
- 2 Courtemanche, N. & Pollard, T. D. Interaction of profilin with the barbed end of actin filaments. *Biochemistry* **52**, 6456-6466 (2013).
- 3 Kanematsu, Y. *et al.* Structures and mechanisms of actin ATP hydrolysis. *Proceedings of the National Academy of Sciences* **119**, e2122641119 (2022).
- 4 Kudryashov, D. S. & Reisler, E. ATP and ADP actin states. *Biopolymers* **99**, 245-256 (2013).
- 5 Iwasa, M., Takeda, S., Narita, A., Maéda, Y. & Oda, T. Mutagenic analysis of actin reveals the mechanism of His161 flipping that triggers ATP hydrolysis. *Frontiers in Cell and Developmental Biology* **11**, 1105460 (2023).
- 6 Chen, X., Ni, F., Kondrashkina, E., Ma, J. & Wang, Q. Mechanisms of leiomodin 2-mediated regulation of actin filament in muscle cells. *Proceedings of the National Academy of Sciences* **112**, 12687-12692 (2015).
- 7 Chou, S. Z. & Pollard, T. D. Mechanism of actin polymerization revealed by cryo-EM structures of actin filaments with three different bound nucleotides. *Proceedings of the National Academy of Sciences* **116**, 4265-4274 (2019).
- 8 Yount, R. G., Babcock, D., Ballantyne, W. & Ojala, D. Adenylyl imidiodiphosphate, an adenosine triphosphate analog containing a PNP linkage. *Biochemistry* **10**, 2484-2489 (1971).

Reviewers' Comments:

Reviewer #2:

Remarks to the Author:

In this second revision, Ma et al. characterized interaction between Spa2 fragments and AMPPNP-actin filaments. They demonstrated that Spa2 does not bundle AMPPNP-actin filaments, which is consistent with one of the main conclusions of this study i.e., Spa2 distinguishes between ADP-actin filaments from ATP- or ADP-Pi-actin filaments, and bundles only ADP-actin filaments. This new data is an important addition to this manuscript, but the questionable situation regarding this conclusion was not resolved by this addition.

As mentioned in the authors' own rebuttal, at as early as 2 min after the onset of polymerization of ATP-G-actin, 30% of the protomers are in the ADP-bound state due to rapid ATP hydrolysis followed by Pi release. At 30 min, which is the so-called "ATP-F-actin" preparation in this study, the vast majority would be the ADP-bound protomers, which was the concern of Reviewer 3 and myself. This "ATP-F-actin" was not bundled by Spa2, meaning that, despite the majority protomers being in the ADP-bound state, Spa2 is unable to bundle those filaments. In other words, certain types of filaments are not bundled by Spa2 even though they consist mainly of ADP-actin. I thus disagree with the authors' rebuttal that "our two aforementioned tests (=the AMPPNP experiment and the 2 min-old ATP-F-actin experiment) more effectively mitigate concerns about the 30-minute ATP-F-actin potentially having ADP-actin."

To address this issue, it would be best if the authors perform some new experiment to unveil what additional condition is required for bundling of mostly ADP-filaments by Spa2. If this is difficult, then at the least, the authors should add an explicit reservation to the conclusion that some other, unidentified additional condition is necessary for the Spa2 interaction with mostly ADP-filaments. Related to the above, "ATP-F-actin" used frequently in this ms is misleading and I suggest changing this to something like "F-actin prepared from ATP-G-actin". Alternatively, please explain explicitly that "ATP-F-actin" is only an operational term and majority of the constituent protomers likely carry bound ADP, not ATP or ADP+Pi.

Also, please explain how ADP-Pi actin filaments were prepared. If you simply polymerize ADP-Pi G-actin in a buffer that does not contain a high concentration of Pi, you will probably end up with ADP-actin filament due to Pi release.

Reviewer #3:

Remarks to the Author:

Judging from the authors' rebuttal to Reviewer 2 and myself, the authors seem to have misunderstood our intention. I suggested based on our knowledge that ATP- and ADP-F-actin should become biochemically and structurally indistinguishable after phosphate release. It would be a big surprise to the entire actin research field if Spa2 could distinguish the two filaments. That is why I suspected the difference in the filament concentration inducing different bundling. Reviewer 2 also needed more solid evidence, kindly suggesting the possible involvement of ATP exchange at the barbed end. However, the new experiments were designed in the completely opposite direction. Read our previous comments carefully. We requested identical conditions for ATP- and ADP-actin for comparison, namely no remaining ADP-Pi-bound actin, the same filament concentration and removal of ATP in the assay buffer as ATP may exchange at the barbed end.

The revised method section states "For ATP-F-actin preparation, 2 μ M and 5 μ M ATP-G-actin were respectively used to generate 0.5 μ M ATP-F-actin (30 min incubation) and 3 μ M ATP F-actin (2 min incubation) by adding 10 \times KME before Spa2 incubation" (p33-34). Under this condition, only a small amount of F-actin is formed at the start of incubation. Actin should polymerize thereafter but some remains bound to ADP-Pi even at the end. The current manuscript thus shows comparison of the results between poorly controlled nucleotide-states. As it turned out that ADP-Pi-actin might exist in the negative control, the effect of inorganic phosphate must be carefully tested in all in vitro assays. Without clarification of this issue, substantial confusion may arise among the readers.

I have additional concerns. Especially the points #1 and #2 are serious. I want a sincere response from the authors.

1) I am perplexed with Figure R4a in the rebuttal letter, p11. Figure R4a cannot derive from the

same data set as in the initial Extended Fig. 4b. How can the authors replace the data again and again?

2) Although I strongly opposed the deletion and replacement of the data in the last review round, the authors again replaced another data in Fig. 3b after my 2nd criticism. In this situation, I would like to ask the authors to upload the original timelapse data on the journal server without any modifications (including time stamps) for both the previous and the current images shown in Fig. 3b. I cannot see severing events very well in the small images on the rebuttal letter p7 (Figure R1).

3) I apologize for not reading the method section carefully in the last review round. I overlooked the text, "before being diluted with F-Buffer (G-buffer plus 1×KME) and applied to polylysine (0.01%)-coated coverglass ..." in the previous version. However, dilution should be avoided to visualize actin bundles. Depending on the conditions, bundling is reversed very quickly after dilution (see Goverman et al. Biophys J 71:1485, 1996, for example). It should be easy to repeat the experiments and visualize actin bundles at low magnification.

4) Can you explain what individual dots on the graph in Extended Fig. 4e stand for? How many experiments were carried out? How were the "curves" obtained for individual dots?

5) I also request carefully confirming the methods throughout this manuscript.

REVIEWER COMMENTS

Reviewer #2 (Remarks to the Author):

In this second revision, Ma et al. characterized interaction between Spa2 fragments and AMPPNP-actin filaments. They demonstrated that Spa2 does not bundle AMPPNP-actin filaments, which is consistent with one of the main conclusions of this study i.e., Spa2 distinguishes between ADP-actin filaments from ATP- or ADP-Pi-actin filaments, and bundles only ADP-actin filaments. This new data is an important addition to this manuscript, but the questionable situation regarding this conclusion was not resolved by this addition.

As mentioned in the authors' own rebuttal, at as early as 2 min after the onset of polymerization of ATP-G-actin, 30% of the protomers are in the ADP-bound state due to rapid ATP hydrolysis followed by Pi release. At 30 min, which is the so-called "ATP-F-actin" preparation in this study, the vast majority would be the ADP-bound protomers, which was the concern of Reviewer 3 and myself. This "ATP-F-actin" was not bundled by Spa2, meaning that, despite the majority protomers being in the ADP-bound state, Spa2 is unable to bundle those filaments. In other words, certain types of filaments are not bundled by Spa2 even though they consist mainly of ADP-actin. I thus disagree with the authors' rebuttal that "our two aforementioned tests (=the AMPPNP experiment and the 2 min-old ATP-F-actin experiment) more effectively mitigate concerns about the 30-minute ATP-F-actin potentially having ADP-actin." To address this issue, it would be best if the authors perform some new experiment to unveil what additional condition is required for bundling of mostly ADP-filaments by Spa2. If this is difficult, then at the least, the authors should add an explicit reservation to the conclusion that some other, unidentified additional condition is necessary for the Spa2 interaction with mostly ADP-filaments.

Response: We are grateful for the reviewer's suggestions and concur with the perspective provided by Reviewer #2. We have now thoroughly addressed the diversity of ADP- and ATP-bound species in F-actin samples, which originates from the swift hydrolysis when prepared from ATP-actin monomers. In our latest revision, while we presented evidence showing Spa2's lack of crosslinking ability with AMPPNP-actin filaments, we have yet to address why Spa2 does not crosslink with actin filaments prepared from ATP-actin monomers, which contain all nucleotide species, including ADP.

Following the recommendation, we have exercised additional caution in our assertions, incorporating a discussion of the yet-to-be-elucidated mechanisms underlying Spa2's preferential interaction with ADP-rich filaments, both in the results section (page 12), the abstract, and the discussion (last paragraph).

In addition, to prevent any confusion regarding the nucleotide state of F-actin, we have revised our terminology. Wherever we previously referred to "ATP-F-actin," we now use the term "actin filament prepared from ATP-actin monomers" consistently across the revised manuscript. This change underscores the enigmatic reasons behind the requirement for a high proportion of ADP-bound actin in F-actin for proper function.

Furthermore, we have amended the wording of our conclusion in the abstract to clarify that Spa2 specifically binds to ADP-actin monomers and predominantly bundles F-actin that prepared from ADP-actin monomer.

Related to the above, "ATP-F-actin" used frequently in this ms is misleading and I suggest changing this to something like "F-actin prepared from ATP-G-actin". Alternatively, please explain explicitly that "ATP-F-actin" is only an operational term and majority of the constituent protomers likely carry bound ADP, not ATP or ADP+Pi.

Response: Thank you reviewer#2 suggestion and we now use the term "actin filament prepared from ATP-actin monomers" consistently across the revised manuscript.

Also, please explain how ADP-Pi actin filaments were prepared. If you simply polymerize ADP-Pi G-actin in a buffer that does not contain a high concentration of Pi, you will probably end up with ADP-actin filament due to Pi release.

Response: We have introduced a dedicated section in the Methods, now located on page 34, to provide a detailed description of our procedure. To prepare ADP-P_i filaments, we combined Mg-ADP-actin monomers with an ADP-P_i-F-buffer. This buffer has a final concentration of 60 mM potassium phosphate, a level that ensures saturation conditions for the association and dissociation of ADP-actin at both the barbed and pointed ends¹. Notably, this concentration exceeds those typically employed in protein crystallization and structural studies²⁻⁴.

Reviewer #3 (Remarks to the Author):

Judging from the authors' rebuttal to Reviewer 2 and myself, the authors seem to have misunderstood our intention. I suggested based on our knowledge that ATP- and ADP-F-actin should become biochemically and structurally indistinguishable after phosphate release. It would be a big surprise to the entire actin research field if Spa2 could distinguish the two filaments. That is why I suspected the difference in the filament concentration inducing different bundling. Reviewer 2 also needed more solid evidence, kindly suggesting the possible involvement of ATP exchange at the barbed end. However, the new experiments were designed in the completely opposite direction. Read our previous comments carefully. We requested identical conditions for ATP- and

ADP-actin for comparison, namely no remaining ADP-Pi-bound actin, the same filament concentration and removal of ATP in the assay buffer as ATP may exchange at the barbed end.

We are grateful to Reviewer 3 for highlighting this issue clearly, and we apologize for our initial misunderstanding. We now fully comprehend your comments. As requested, we have conducted additional experiments and have all concentrations and conditions consistent across the various actin nucleotide species tests. The results of these new experiments have been incorporated into Extended Figure 5. Our findings remain consistent with our original conclusions.

The reactivity of AMPPNP and ADP-actin filaments with Spa2 displays contrasting behaviors, which does not lead to ambiguity. The primary source of confusion was the F-actin formed from ATP-actin monomers, which contains a mixture of nucleotides, including ADP, yet still yielded negative results in the Spa2 crosslinking assay. Our data suggest a stringent requirement for ADP-rich F-actin for Spa2 binding, as Reviewer 2 indicated, hinting at as-yet-undetermined mechanisms. We have taken extra care to articulate our findings and to refine our language to preclude potential misunderstandings or confusion. These clarifications have been explicitly mentioned in the abstract, results, and discussion sections.

We have now thoroughly addressed the diversity of ADP- and ATP-bound species in F-actin samples, which originates from the swift hydrolysis when prepared from ATP-actin monomers. Following the recommendation, we have exercised additional caution in our assertions, incorporating a discussion of the yet-to-be-elucidated mechanisms underlying Spa2's preferential interaction with ADP-rich filaments, both in the results section (page 12), the abstract, and the discussion (last paragraph).

In addition, to prevent any confusion regarding the nucleotide state of F-actin, we have revised our terminology. Wherever we previously referred to "ATP-F-actin," we now use the term "actin filament prepared from ATP-actin monomers" consistently across the revised manuscript. This change underscores the enigmatic reasons behind the requirement for a high proportion of ADP-bound actin in F-actin for proper function.

Furthermore, we have amended the wording of our conclusion in the abstract to clarify that Spa2 specifically binds to ADP-actin monomers and predominantly bundles F-actin that contains mostly ADP-actin.

The revised method section states "For ATP-F-actin preparation, 2 μ M and 5 μ M ATP-G-actin were respectively used to generate 0.5 μ M ATP-F-actin (30 min incubation) and 3 μ M ATP F-actin (2 min incubation) by adding 10 \times KME before Spa2 incubation" (p33-34). Under this condition, only a small amount of F-actin is formed at the start of incubation. Actin should polymerize thereafter but some

remains bound to ADP-Pi even at the end. The current manuscript thus shows comparison of the results between poorly controlled nucleotide-states. As it turned out that ADP-Pi-actin might exist in the negative control, the effect of inorganic phosphate must be carefully tested in all in vitro assays. Without clarification of this issue, substantial confusion may arise among the readers.

Response: The "poorly defined nucleotide states" is originated from the nature of rapid hydrolysis of ATP in F-actin samples prepared from ATP-actin monomers. To address this issue, we included experiments with AMPPNP-F-actin in our last revision. This change, along with the use of uniform concentrations across experiments, has resolved the confusion mentioned previously and the concerns regarding variable concentrations. However, while we presented evidence showing Spa2's lack of crosslinking ability with AMPPNP-actin filaments, we have yet to address why Spa2 does not crosslink with actin filaments prepared from ATP-actin monomers, which contain all nucleotide species, including ADP. We appreciate the reviewer's feedback for highlighting this aspect again, which helps in enhancing the accuracy and clarity of our conclusions.

Additionally, here, we have conducted the requested experiments on ADP-Pi-F-actin. By testing the interaction of Spa2 with ADP-Pi-actin filaments, both in the absence and presence of PEG, we have obtained clear results. Spa2 does not exhibit crosslinking activity with ADP-Pi-actin filaments. These findings (Figure R1) have now been added to Extended Figure 5c and d.

Figure R1. Representative fluorescence microscopy images show 3 μM ADP-Pi-actin filaments after incubation with varying concentrations of Spa2(1-535), both with and without PEG. These images were captured using a 100× oil immersion objective lens.

I have additional concerns. Especially the points #1 and #2 are serious. I want a sincere response from the authors.

1) I am perplexed with Figure R4a in the rebuttal letter, p11. Figure R4a cannot derive from the same data set as in the initial Extended Fig. 4b. How can the authors replace the data again and again?

Response: We deeply appreciate reviewer keen discovery of inconsistency that the raw value and the normalized value of anisotropy experiment. Firstly we did not replace data, and these three batches data in last revision are provided as raw anisotropy value data base on experiments. Secondly, we apologies for the unintentional omission in last revision to clarify the miscalculation we made for the normalized anisotropy value in the very initial submission. For calculating the normalized value, the first step is to minus the minimum value of that set data, the minimum value were 0.077421 and 0.06129 for the first list data of Spa2(1-535) and Spa2(1-281), but we minus 0.0077421 and 0.061229 for each panel of data, so the normalized data become inconsistent with the raw value. In our 2nd version, we only provided the raw anisotropy value. Hence, we forgot to elaborate on the error concerning the normalized results in the initial version, which had since been corrected in the second revision and was no longer present.

The detailed history we have presented of the issues encountered serves to demonstrate that there was no intent of scientific dishonesty or arbitrary data substitution. We are indeed thankful for Reviewer 3's meticulous and conscientious examination of our manuscript. Their efforts have enabled us to detect and rectify these oversights, for which we are profoundly grateful.

2) Although I strongly opposed the deletion and replacement of the data in the last review round, the authors again replaced another data in Fig. 3b after my 2nd criticism. In this situation, I would like to ask the authors to upload the original timelapse data on the journal server without any modifications (including time stamps) for both the previous and the current images shown in Fig. 3b. I cannot see severing events very well in the small images on the rebuttal letter p7 (Figure R1).

Response: First, we revised the representative figures in the last submission to better reflect our quantitative data and prevent any misinterpretation. Second, our images documenting severing events in the last rebuttal letter were captured at high resolution; however, due to space constraints, they were presented in a compressed format. If the reviewer magnifies these images by dragging and expanding the figure, the details of each severing event should become clear. Nonetheless, to ensure complete transparency, we have now uploaded all the raw images of each filament to the Figshare data repository, and they can be accessed using the provided identification number (<https://figshare.com/s/f1d73eb1ff79bdd92615>).

3) I apologize for not reading the method section carefully in the last review round. I overlooked the text, "before being diluted with F-Buffer (G-buffer plus 1×KME) and applied to polylysine (0.01%)-coated coverglass ..." in the previous version. However, dilution should be avoided to visualize actin bundles. Depending on the conditions, bundling is reversed very quickly after dilution (see Gorman et al. Biophys J 71:1485, 1996, for example). It should be easy to repeat the experiments and visualize actin bundles at low magnification.

Response: We appreciate the suggestion from the reviewer. The study by Goverman et al. (Biophys J, 71:1485, 1996) demonstrated that bundle formation can be reversible when an F-buffer without additional cations is used. In our experiments, however, the F-buffer contained 50 mM KCl and 1 mM Mg²⁺.

As requested, we have conducted additional experiments to visualize actin bundles using a low magnification 40x objective lens without dilution. Compared with previous diluted samples, the current results (Figure R2) showed same reactivity with Spa2(1-535) for crosslinking actin filaments polymerized from ADP-G-actin, ATP-G-actin, ADP-Pi-actin, or AMPPNP-G-actin. Given the low resolution of these images and the fact that the results remain unchanged, we have chosen not to include them in our figures. Including these images could potentially detract from the figure's compactness and compromise the overall clarity of the presentation.

Figure R2. Representative fluorescence micrographs of ADP-actin, ATP-actin, ADP-Pi-actin, and AMPPNP-actin filaments at the indicated concentrations were incubated with various concentrations of Spa2(1-535), both with and without PEG. The images were captured using a 40× oil immersion objective lens.

4) Can you explain what individual dots on the graph in Extended Fig. 4e stand for? How many experiments were carried out? How were the “curves” obtained for individual dots?

Response:

1. We quantified the number of filaments as follows: we observed 93 for Cof1, 98 for Cof1+Spa2(1-535), 91 for Cof1+Spa2(281-535), and 90 for Cof1+Spa2(1-281) from the 0s to 80s timepoints. We manually counted the number of severing events every 20 seconds, accumulating the severing event totals at 20s, 40s, 60s, and 80s. Once we obtained the cumulative number of severing events for each filament from 20 to 80 seconds for each protein, we divided by the filament length to calculate the values for the Y-axis in Extended Fig. 4d. The individual dots in Extended Fig. 4e represent the slopes (rate of severing events) for the Y-axis versus X-axis from Extended Fig. 4d.

2. We conducted two independent sets of experiments.

3. We utilized the linear regression equation from GraphPad to calculate the slope, which represents the number of breaks per μm per second. We then multiplied the slope by 60 to convert this to breaks per μm per minute. This methodology has been detailed on page 25/26 of our manuscript.

5) I also request carefully confirming the methods throughout this manuscript.

Response: Thank you for the suggestion. Following your advice, we have thoroughly and carefully reviewed the methods section.

- 1 Fujiwara, I., Vavylonis, D. & Pollard, T. D. Polymerization kinetics of ADP-and ADP-Pi-actin determined by fluorescence microscopy. *Proceedings of the National Academy of Sciences* **104**, 8827-8832 (2007).
- 2 Chou, S. Z. & Pollard, T. D. Mechanism of actin polymerization revealed by cryo-EM structures of actin filaments with three different bound nucleotides. *Proc Natl Acad Sci U S A* **116**, 4265-4274 (2019). <https://doi.org:10.1073/pnas.1807028115>
- 3 Courtemanche, N. & Pollard, T. D. Interaction of profilin with the barbed end of actin filaments. *Biochemistry* **52**, 6456-6466 (2013).
- 4 Oosterheert, W., Klink, B. U., Belyy, A., Pospich, S. & Raunser, S. Structural basis of actin filament assembly and aging. *bioRxiv*, 2022.2003.2029.486216 (2022). <https://doi.org:10.1101/2022.03.29.486216>

Reviewers' Comments:

Reviewer #2:

Remarks to the Author:

In this latest version of the manuscript, the authors now explicitly discuss that there is a serious problem in their original simple conclusion that Spa2 bundles ADP-F-actin but not ATP-F-actin. It would be ideal if the additional factor that governs the binding specificity of Spa2 is determined, but the discovery of an ADP-G-actin specific nucleator and demonstrating its physiological significance is noteworthy. I now support the publication of this manuscript in a leading journal such as Nature Communications.

Reviewer #3:

Remarks to the Author:

Now the authors share the movie files uploaded on the web server. Although they did not upload "the original data files" as requested, the movies are conveniently aligned side by side. It is easy to compare the results between different conditions.

Let's look at the "Fig3b.tif". Between 35 and 45 seconds, the filament severing is much more promoted in "+Spa2(1-535)" than in the left control images, "+cof1". Especially fragmentation is prominent on the right side of "+Spa2(1-535)" movie. This supports my criticisms for the oversight of enhanced filament severing by Spa2.

This is not the only example indicative of enhanced filament severing. For instance, we can find a less prominent but clear example even with "ATP-G-actin". Carefully watch the movie "Extended Fig.4l.tif", which shows actin assembly using 0.5 μ M ATP-actin monomer with or without 10 nM Spa2(1-535), Spa2(281-535), or Spa2(1-281). Between 5 and 8 min, the length of F-actin was apparently shorter in Spa2(1-535) than in ATP-actin alone. The data suggest that Spa2 alone may enhance filament fragmentation, especially in solution but not of glass-anchored filaments.

I don't understand why the authors continue to misinterpret the data. The filament severing activity works to increase newly growing filaments. This effect is widely known for the cofilin/ADF family. To distinguish filament severing from nucleation, very carefully designed experiments are required. In all presented data, increases in the amount of F-actin may be explained by severing. I pointed out this possibility in the first review round. Unfortunately, the authors still do not appear to understand this point.

In addition, I found a case of the lack of honesty again in the rebuttal. They now include a clarification in the rebuttal for why the anisotropy data looks so different between initial submissions and the rebuttal Figure R4a in the 2nd rebuttal "414160_2_rebuttal_8074551_s14vbn" despite the same data. I am not impressed by the clarification. Such normalization would never change the graph trend between the two.

REVIEWER COMMENTS

Reviewer #2 (Remarks to the Author):

In this latest version of the manuscript, the authors now explicitly discuss that there is a serious problem in their original simple conclusion that Spa2 bundles ADP-F-actin but not ATP-F-actin. It would be ideal if the additional factor that governs the binding specificity of Spa2 is determined, but the discovery of an ADP-G-actin specific nucleator and demonstrating its physiological significance is noteworthy. I now support the publication of this manuscript in a leading journal such as Nature Communications.

Response: We are grateful for the reviewer's endorsement of our novel discovery.

Reviewer #3 (Remarks to the Author):

Now the authors share the movie files uploaded on the web server. Although they did not upload "the original data files" as requested, the movies are conveniently aligned side by side. It is easy to compare the results between different conditions.

1. Let's look at the "Fig3b.tif". Between 35 and 45 seconds, the filament severing is much more promoted in "+Spa2(1-535)" than in the left control images, "+cof1". Especially fragmentation is prominent on the right side of "+Spa2(1-535)" movie. This supports my criticisms for the oversight of enhanced filament severing by Spa2.

This is not the only example indicative of enhanced filament severing. For instance, we can find a less prominent but clear example even with "ATP-G-actin". Carefully watch the movie "Extended Fig.4I.tif", which shows actin assembly using 0.5 μ M ATP-actin monomer with or without 10 nM Spa2(1-535), Spa2(281-535), or Spa2(1-281). Between 5 and 8 min, the length of F-actin was apparently shorter in Spa2(1-535) than in ATP-actin alone. The data suggest that Spa2 alone may enhance filament fragmentation, especially in solution but not of glass-anchored filaments.

I don't understand why the authors continue to misinterpret the data. The filament severing activity works to increase newly growing filaments. This effect is widely known for the cofilin/ADF family. To distinguish filament severing from nucleation, very carefully designed experiments are required. In all presented data, increases in the amount of F-actin may be explained by severing. I pointed out this possibility in the first review round. Unfortunately, the authors still do not

appear to understand this point.

Response: First and foremost, we acknowledge Reviewer #3's observation and have identified the reason that prevented us from concluding this with criteria to follow statistical analysis that supports better overall evaluation. We included a corresponding discussion in the revised manuscript. We did not misinterpret our data and did not insist in doing so. We have taken great care to avoid overstating our findings, which in itself is a precautionary measure.

Regarding the proposal that Spa2 promotes F-actin severing, as suggested by the reviewer, it was based on two observations:

R1. Between 35 and 45 seconds in the representative movie in Figure 3b, filament severing is more pronounced in the "+Spa2(1-535)" sample compared to the control images, in the presence of Cof1.

R2. Between 5 and 8 minutes, the length of F-actin assembled using 0.5 μ M ATP-actin monomer was shorter in Spa2(1-535) as shown in Extended Figure 4l.

On the other side, we have drawn our conclusions from the data and experiments as follows:

A1. In 1st revision, we investigated whether Spa2 directly causes severing activity on F-actin. We used 500 nM Spa2 variants and did not observe noticeable severing events in Extended Figure 4c.

A2. Extended Figure 4l aimed to compare the elongation speed with 10 nM Spa2 proteins over a 10-minute time-lapse, accounting for the fact that these experiments, which start with G-actin, are performed sequentially one by one. Despite variations in slide preparation and sample addition, which could introduce minor time discrepancies of starting point with several seconds or even slightly more although we have tried to be as quick as possible for all. Considering this potential time variation, we think comparing the data using a fixed time window is a fair way to make a conclusion, such as a comparison of elongation speed. As a result, we did not observe filament-breaking, as shown in Figure 3b, to ascribe direct F-actin severing by 10 nM of Spa2 (1-535). This elaboration provides our response to the reviewer's point R2 above.

We cannot definitively determine whether a concentration of 10 nM Spa2 in Extended Figure 4l fragments the F-actin in the solution, as suggested by the reviewer, as direct observation and quantification are not as practical and accurate as TIRF-based assay. However, in a separate experiment aimed at assessing Spa2's impact on ATP-G-actin bundling activity, we employed a consistent concentration of 500 nM ATP-G-actin along with 5 μ M of Spa2 in Extended Figure 5f. In this instance, we still did not observe any noticeable Spa2-mediated direct depolymerization of F-actin. Taking all of this evidence into account, we are unable to draw a conclusion regarding Spa2's direct severing activity.

A3. We attribute the increased number of seeds observed with an additional 500nM of Spa2 (1-535) in Figure 3b to the mixed effect of facilitating Cof1 severing and de novo F-actin production. Without this, the F-actin strands would shorten over time and fail to settle on the coverslip, as observed with Cof1 alone in Figure 3b.

A4. In an ADP G-actin polymerization assay, 10 nM of Spa2 (1-535) or Spa2 (281-535) demonstrated strong nucleation activities, correlating with the capability of synthesizing short F-actin strands as seen in Figure 3b.

Together, our goal has consistently been to provide a thorough and fair interpretation that incorporates all relevant experimental findings, guided by rigorous statistical analysis. Our above-listed results demonstrated clear evidence of Spa2 nucleation on ADP-G-actin, observed no activity in directly severing F-actin derived from ATP-G-actin, and were unable to draw any definitive conclusions regarding Spa2's role in direct depolymerization-based on our statistical analysis in Figure 3b. We are confident that the phenotype exhibited in Figure 3b is largely due to Cof1-mediated severing and Spa2-mediated ADP-G actin nucleation activities.

Nevertheless, by carefully thinking over and over Reviewer #3's comments, we have pondered over all potential reasons for not catching this event in our statistical analysis, while the 35-45s phenotype in Figure 3b revealed such a message of more depolymerization with Spa2 (1-535) and Cof1, but not other Spa2 truncating variants. Eventually, we identified a systematic error that could have led to an oversight of a slight overall increase in severing in Figure 3b with additional Spa2 (1-535). Severing events for each filament were tracked manually across increasing time intervals. We are confident in our ability to accurately trace every severing event up to 80 seconds, and initially, severing efficiency was measured within a 20-80 second window, as shown in Figure R1a (was the previous Extended Figure 4e). However, a system error could be that the short filaments could drift out of the TIRF microscope's field of view late. Anticipating that such error could compound over longer observation periods, we revised our measurement window to 20-60 seconds and re-conducted our statistical evaluation using the same sets of severing counting events. The reanalyzed severing using 20-60 seconds turned out to reveal a statistically significant difference when comparing Cof1+Spa2(1-535) to Cof1 alone, with a p-value of 0.0036, although other Spa2 versions exhibit null in statistics. This finding of Cof1+Spa2(1-535), which we initially overlooked, has now been incorporated into the revised Extended Figure 4e, result on page 8-9, and discussion on page 20, taking into consideration that the 20-60 second timeframe should be more precise than that of the 20-80 second range, given the reduced potential for the aforementioned systematic errors that may occur over the longer observation period. Here, we thank reviewer's feedback, which has enabled us to rectify a calculation error that could have been easily overlooked. We believe we have understood the reviewer's point correctly.

Although Spa2 (235-535) did not demonstrate a noticeable enhancement of Cof1-mediated severing activity, Spa2 (1-353) exhibited such promotion. We hypothesize that the intrinsically disordered region (IDR) in Spa2 (1-535) may introduce some conformational stability or alter the positioning of Cof1, potentially influencing its association with F-actin and biochemical activities. Exploring this mechanistic aspect further is a promising avenue for future research, and it may benefit from leveraging the elegant Cryo-EM technique, such as previously employed by Kotaro Tanaka et al. in Nature

Communications 2018. We have updated the analysis using the 20-60s with the above-claimed reasons and also discussed in the discussion session.

Figure R1. Analysis comparison of Actin Filament Disassembly by Cof1 and Spa2 variants.

a. It presents data from the 3rd round submission, where the average severing rates for each condition were determined by calculating the average slopes of curves between the 20- to 80-second time interval. b. It presents data from the average severing rates for each condition, which were computed by evaluating the average slopes of curves during the 20- to 60-second time frame. The error bars shown represent the mean \pm standard deviation.

2. In addition, I found a case of the lack of honesty again in the rebuttal. They now include a clarification in the rebuttal for why the anisotropy data looks so different between initial submissions and the rebuttal Figure R4a in the 2nd rebuttal "414160_2_rebuttal_8074551_s14vbn" despite the same data. I am not impressed by the clarification. Such normalization would never change the graph trend between the two.

In the 3rd revision, we clarified that we did not repeatedly replace data. Here, we reiterate our clarification again with the same fact. We draw a timeline of all the critiques, corrections, and clarifications, which helps us all to recall the review process (Figure R2).

Extended fig4b

Timeline of Critique and Clarification

Figure R2. Timeline of Critique and Clarification for Extended Fig. 4b.

1. We used a normalized anisotropy value for the initial submission in February 2023 (Figure R3a).
2. During our 1st revision process (Feb-July 2023), in clarifying the presentation of the anisotropy assay and supporting our conclusion, we conducted a new set of experiments with three biological replicates, mirroring the original number of replicates from the initial submission. To maintain a consistent statistical sample size for the final presentation, we at that time decided to use the normalized data from the new experiments in lieu of the initially submitted figures. This decision was found to be inappropriate and disagreed by Reviewer #3. Next, we tried our best to address this misunderstanding and explained in the 2nd revision. We agree with Reviewer #3's point that integrating all experiments is a better way to show the experiment's results. To prove ourselves, we performed an additional third set of experiments with another three replicates. Subsequently, we amalgamated all nine biological replicates, updating Figure 4b in the Extended Data for the second revision (July-Sept 2023).

Following Reviewer #3's suggestion, we presented original data without normalization, which differed from the normalized format in the original submission. (Figure R3b)

- In the 3rd round of review, which took place from September to October 2023, reviewer #3 pointed out an issue with the initial normalized data (Figure R3a), stating that it couldn't have been generated from the original raw data (Figure R3b). In our 3rd round revision, from October to November 2023, we provided an explanation acknowledging our previous omission of mentioning the normalization calculating mistakes made in the 3rd revision. We found we actually used the wrong equation to generate the normalized data, and the wrong equation is **The normalized ratio = (anisotropy value-the minimum anisotropy value)/maximum value**. This is the main reason why the trend of the normalized data is different from the raw data. Unfortunately, in 3rd revision, we did not provide details for such a wrong equation explicitly. Since we stopped using the normalization data in the second revision, and also during the revision, we focused on addressing the issue and did not delve into very detailed explanations regarding this calculation mistake, believing that it was not necessary. Such judgment appears to be wrong. Nevertheless, Reviewer #3's characterization of our rebuttal as lacking honesty with an unimpressive explanation is disheartening. Here, we have endeavored to provide clarity through the descriptions above and the additional details below.
- In this 4th round of review (Nov to Dec 2023), we surmise that this critique might stem from our failure to display the comprehensive original data alongside the miscalculated and calculations detail in a collective format in the 3rd round of revision. We have now done so, which can be reviewed below.

Figure R3. Detail of the anisotropy assay for NP-ATP-G-actin with Spa2.

- The graph used in the initial submission Extended Fig. 4b with the normalized data.
- The corresponding source data of (a). The normalization data was using the following wrong equation: **the normalized ratio = (anisotropy value-the minimum anisotropy value) /maximum value**. The normalization process for generating (a) from (c) is the following: Step (1) anisotropy raw value minus the minimum anisotropy value from the same set of data. We suppose to
- The graph from raw anisotropy value used for clarification in the 2nd revision rebuttal letter Figure R2a.
- The corresponding source data of (c). The normalization data was using the following wrong equation: **the normalized ratio = (anisotropy value-the minimum anisotropy value) /maximum value**. The normalization process for generating (a) from (c) is the following: Step (1) anisotropy raw value minus the minimum anisotropy value from the same set of data. We suppose to

subtract 0.07742 in the Spa2 (1-535) 1st column and 0.06129 in 3rd column. However, we subtracted 0.007742 and 0.061229 (highlighted in red), respectively, instead. (2) Step 1/maximum value (3) Step 2 * 100.

Reviewers' Comments:

Reviewer #2:

Remarks to the Author:

After reviewing the authors' rebuttal and considering Reviewer 3's concerns regarding authors' scientific honesty, I concur with Reviewer 3's observation that there may be a presence of bias in the selection and presentation of data to bolster the desired conclusion in the initial revision. Furthermore, the absence of a comprehensive explanation for the normalization miscalculation in the third revision, leading to a qualitatively different graph appearance, further intensified Reviewer 3's skepticism.

Despite these issues, I am hesitant to categorize the authors' actions as unacceptable or extraordinary. I am not inclined to believe that there was a deliberate effort to fabricate data. Instead, I acknowledge the potential value of reporting a novel, ADP-actin specific nucleator in a prestigious journal like Nature Communications.

Reviewer #3:

Remarks to the Author:

The last rebuttal (p5) explains the reason for the apparent discrepancy between the same data in the initial submission and in the Figure R4a of the 2nd rebuttal ("414160_2_rebuttal_8074551_s14vbn") as "We found we actually used the wrong equation to generate the normalized data, and the wrong equation is The normalized ratio = (anisotropy value-the minimum anisotropy value)/maximum value." I would say that the equation itself is not the reason for their mistakes. The authors simply do not understand basics in data analysis. What they did for the normalization of anisotropy data was rescaling each value between the 0 (bottom) to 100 (top) range of EACH set of data in triplicates. And then, they averaged the three RESCALED values to make the graph showing the "normalized data". This is terrible. With this calculation, the outliers in each set have unwanted, strong impact on each 0 to 100 range and thereby the distribution of "normalized values" in each data set. Their comparison means nothing. The authors apparently lack the scientific literacy. I recommend the editors to re-evaluate the reviewers' comments and the rebuttals. There still remain unaddressed or mistreated issues. For example;

- 1) The URL of the file server for original data set (line 838) doesn't work.
- 2) The "original data" for some movie data (ex. Extended Fig. 4b) in the previous server stop much earlier than those presented in the figure images.
- 3) In addition, several "original" movies are shown only in the limited area and/or for limited examples.
- 4) Even after reanalysis of severing rates in Extended Fig. 4e, the authors never attempt to revise the graph in Extended Fig. 4d, which I criticized for its inaccuracy twice in the previous review rounds. I still have strong concerns over the accuracy of measurement in BOTH graphs. For instance, the severing events much faster than the Extended Fig. 4e can be found with a fraction of filaments if we observe them at the ORIGINAL 5 sec intervals. Please correct them.
- 5) The authors do not understand the very basics in actin biochemistry. Each severing event increases the filament number by one, which is often faster than nucleation. At low G-actin concentrations such as in Fig. 3b and Extended Fig, 4g and 4l, actin nucleation is very slow at early time points because nucleation speed is proportional to the square or the cube of the free G-actin concentration. On the other hand, filament severing increases the filament number effectively if there is contamination of filament seeds. We see such occasions in Figure 3d, Extended Fig. 4g and 4j, in which the authors tried to measure the nucleation activity. All the data show apparent contamination of filament seeds at the initial time points in many samples. Therefore, the current data do not necessarily support the authors' conclusion. In addition, last time I pointed out the formation of shorter filaments in some of Extended Fig. 4l. This phenomenon is commonly found in TIRF experiments as severed short actins diffuse fast (faster than longer seeds) being captured on the coverslips. Accelerated severing is quite likely. I suggest improving the data by (i) completely removing the filament seeds (by high speed centrifugation) in the above nucleation experiments or (ii) challenging a wide range of G-actin concentrations to conclude the nucleation activities by Spa2. At least, the discussion for the limitations in the current data along this argument should be included.

6) Please read my current and PREVIOUS criticisms again and address them carefully. I am afraid that the readers will find the same problems as in my review comments.

Reviewer #2

After reviewing the authors' rebuttal and considering Reviewer 3's concerns regarding authors' scientific honesty, I concur with Reviewer 3's observation that there may be a presence of bias in the selection and presentation of data to bolster the desired conclusion in the initial revision. Furthermore, the absence of a comprehensive explanation for the normalization miscalculation in the third revision, leading to a qualitatively different graph appearance, further intensified Reviewer 3's skepticism.

Despite these issues, I am hesitant to categorize the authors' actions as unacceptable or extraordinary. I am not inclined to believe that there was a deliberate effort to fabricate data. Instead, I acknowledge the potential value of reporting a novel, ADP-actin specific nucleator in a prestigious journal like Nature Communications.

Response: We appreciate the impartial assessment provided by Reviewer #2

Reviewer #3

The last rebuttal (p5) explains the reason for the apparent discrepancy between the same data in the initial submission and in the Figure R4a of the 2nd rebuttal ("414160_2_rebuttal_8074551_s14vbn") as "We found we actually used the wrong equation to generate the normalized data, and the wrong equation is The normalized ratio = (anisotropy value-the minimum anisotropy value)/maximum value." I would say that the equation itself is not the reason for their mistakes. The authors simply do not understand basics in data analysis. What they did for the normalization of anisotropy data was rescaling each value between the 0 (bottom) to 100 (top) range of EACH set of data in triplicates. And then, they averaged the three RESCALED values to make the graph showing the "normalized data". This is terrible. With this calculation, the outliers in each set have unwanted, strong impact on each 0 to 100 range and thereby the distribution of "normalized values" in each data set. Their comparison means nothing. The authors apparently lack the scientific literacy. I recommend the editors to re-evaluate the reviewers' comments and the rebuttals. There still remain unaddressed or mistreated issues. For example;

- 1) The URL of the file server for original data set (line 838) doesn't work.

Response: We have now updated the URL (<https://figshare.com/s/f1d73eb1ff79bdd92615>) on page 39.

- 2) The "original data" for some movie data (ex. Extended Fig. 4b) in the previous server stop much earlier than those presented in the figure images.

Response: Firstly, Extended Fig. 4b is an anisotropy assay, and there is no accompanying movie for it.

Secondly, we have thoroughly reviewed the representative images in the figures, along with their corresponding supplementary movies and the “original data” with a larger view (source file, URL <https://figshare.com/s/f1d73eb1ff79bdd92615>). We confirm that the representative images in the figures and their corresponding supplementary movies are all correct.

In the process of preparing the “original data”, we needed to label the movies with time information, including starting time and intervals, as this information was not automatically generated during time-lapse imaging system. With these thoroughly check, we identified and corrected three instances of mislabeling in the “original data”:

1. The interval of 5s was mislabeled as 10s in Extended Fig. 4l;
2. The starting time of movies Extended Fig. 5f and Extended Fig. 7j were mislabeled as 0 min, but should have been 3 min. Consequently, this early shift of 3 min in the original data results in an earlier stop than depicted in the figure images, which might be the point reviewer has mentioned. Although our reported figures and movies are accurate, we apologize for the mislabeling in the "original data".

- 3) In addition, several “original” movies are shown only in the limited area and/or for limited examples.

Response: There are two distinct cases explaining the different sizes observed in the original data.

Firstly, in the original data for Figures (Figures 3b, 3d, 5b, 5c, Extended Figures 4c, g, l, o, 5f, 5g, 7d, 7j), the choice of area was determined by adhering to imaging focus criteria for quality control in data quantification. Despite using iLasV2 Ring-TIRF, GATACA Systems, one of the leading products in the market, achieving uniform illumination and focus throughout the TIRF system for sCMOS camera remains challenging. Hence, for a few movies with non-uniform illuminations, we crop out the focused center portion for quantitative analysis. This focused area is presented as the meaningful source data, devoid of flawed data and potential confusion.

Secondly, the variation in area size in Extended Fig. 4l occurred due to an inadvertent definition of a smaller area for time-lapse imaging in the control sample (ATP-actin). Subsequently, we realized the possibility of making a wider selection, and for all subsequent imaging within the same set of experiments, we employed the full camera view. It's important to note that the initial smaller view for ATP-actin was not intentionally cropped from a larger view. Although the original imaging data for the ATP-actin assay is smaller compared to the other three assays with Spa2 variants, it still contains sufficient filaments for our analysis, contributing to the generation of Extended Fig. 4m, n.

- 4) Even after reanalysis of severing rates in Extended Fig. 4e, the authors never attempt to revise the graph in Extended Fig. 4d, which I criticized for its inaccuracy twice in the previous review rounds. I still have strong concerns over the accuracy of measurement in BOTH graphs. For instance, the severing events much faster than the Extended Fig. 4e can be found with a fraction of filaments if we observe them at the ORIGINAL 5 sec intervals. Please correct them.

Response: First, we are uncertain about the specific changes the reviewer is suggesting to enhance accuracy. In the second revision's rebuttal letter, we presented time-lapse images and movies for every single filament ($n > 90$). Additionally, we marked every severing event with arrows on each filament over time, facilitating manual counting based on visual observation. All these actin filaments with arrows used for Extended Fig. 4d, e have been previously uploaded in "original data". Such a manual counting filament by filament resulted in Extended Fig. 4d. This approach aligns with well established severing counting method¹. We believe we have reported our counting approach accurately and transparently.

Second, in the fourth revision, we conducted an improved analysis and updated Extended Figure 4e, revealing a slightly enhanced severing effect for the Cof1+Spa2(1-535) sample compared to Cof1 alone, Cof1+Spa2(281-535), and Cof1+Spa2(1-281). This conclusion was drawn from the manual counting of randomly selected individual filaments ($n > 90$ each), as mentioned earlier. Upon reviewing the 20-25s interval in Figure 3b/Extended Figure 4d,e, referred to as the "original 5 sec interval" by the reviewer, we attribute the perceived "faster" severing in the movie to variations in F-actin density on the coverslip before adding Cof1 and Spa2 proteins. This variation results from the flowing speed during the addition of diluted F-actin into the chambers, resulting in different landing efficiency of F-actin. This disparity in initial F-actin density does not affect the severing activity and our manual counting when using the same concentration of 50nM Cof1, considering the low F-actin concentration corresponding to 100nM G-actin. The slight enhancement in severing in Cof1+Spa2(1-535) was added and discussed in the last revision round. We reiterate the details of our counting methodology, providing all the original data and filaments for transparency last time. As emphasized in the last revision, we believe the overall evolution and statistics for quantitative analysis offer a fair assessment. Experimental variations are inherent, and we have diligently worked to control and minimize them in every assay. Both the control sample and Spa2(1-281) exhibit a range of distribution, reflective of the facts of variation and genuine results analysis.

- 5) The authors do not understand the very basics in actin biochemistry. Each severing event increases the filament number by one, which is often faster than nucleation. At low G-actin concentrations such as in Fig. 3b and Extended Fig, 4g

and 4l, actin nucleation is very slow at early time points because nucleation speed is proportional to the square or the cube of the free G-actin concentration.

Response: We value the clarification provided by the reviewer and would like to reiterate that this question was addressed during the initial review process. In our first rebuttal letter, we offered the following explanation, emphasizing the potential for a high local concentration conducive to effective nucleation by Spa2: “Spa2 could be utilizing F-actin severing to generate new actin seeds, especially considering the depolymerization occurring near the coverslip. This would result in a high local concentration of ADP-G-Actin. In the TIRF field of view, which penetrates 100-200nm in the Z direction, the ROIs of second and third rows of Figure 3b measure $22 \times 22 \mu\text{m}^2$ in XY directions, thereby resulting in a volume of $\sim 53.24 \mu\text{m}^3$ in the view of imaging. This volume would contain approximately 122 micrometers of F-actin, comprising roughly 45140 monomers, potentially reaching a maximum local concentration of 1.4 μM . This concentration theoretically is sufficient to drive local nucleation. “

Moreover, we provided experimental evidence in Figure 3d and Extended Figure 4g, demonstrating that Spa2 directly nucleates ADP-G-actin. In the TIRF nucleation assays, higher concentrations (3 μM for Figure 3d and 0.8 μM for Extended Figure 4g) were employed, supporting our conclusion that Spa2 functions as an ADP-specific actin nucleator. It's important to note that these assays did not include cofilin. We trust that these explanations and experimental findings contribute to a comprehensive understanding of Spa2's nucleation capabilities.

On the other hand, filament severing increases the filament number effectively if there is contamination of filament seeds. We see such occasions in Figure 3d, Extended Fig. 4g and 4j, in which the authors tried to measure the nucleation activity. All the data show apparent contamination of filament seeds at the initial time points in many samples. Therefore, the current data do not necessarily support the authors' conclusion. In addition, last time I pointed out the formation of shorter filaments in some of Extended Fig. 4l. This phenomenon is commonly found in TIRF experiments as severed short actins diffuse fast (faster than longer seeds) being captured on the coverslips. Accelerated severing is quite likely. I suggest improving the data by (i) completely removing the filament seeds (by high speed centrifugation) in the above nucleation experiments or (ii) challenging a wide range of G-actin concentrations to conclude the nucleation activities by Spa2. At least, the discussion for the limitations in the current data along this argument should be included.

Response: Firstly, our manuscript has distinctly conveyed that Spa2 cannot directly induce severing on both ADP-F-actin (see Extended Figure 5c/without PEG and Extended Figure 5d/with PEG) and ATP-F-actin (see Extended Figure 4c).

Secondly, upon another careful reexamination of Figure 3d and Extended Figure 4g, we were still unable to detect any evidence of Spa2-mediated severing. The mention of Extended Figure 4j appears to be a mistake. Extended Figure 4j does not include any image data. If there is referring to Extended Figure 4l, we want to reiterate that, even in this case, we did not observe any noticeable Spa2-mediated severing. In response to queries raised in the fourth round of review for Extended Figure 4l, we provided a detailed explanation for the variation in initial short filament length. We reproduce our previous responses here for reference. It is crucial to emphasize that our F-actin severing assay did not reveal any Spa2-mediated severing.

“A2. Extended Figure 4l aimed to compare the elongation speed with 10 nM Spa2 proteins over a 10-minute time-lapse, accounting for the fact that these experiments, which start with G-actin, are performed sequentially one by one. Despite variations in slide preparation and sample addition, which could introduce minor time discrepancies of starting point with several seconds or even slightly more although we have tried to be as quick as possible for all. Considering this potential time variation, we think comparing the data using a fixed time window is a fair way to make a conclusion, such as a comparison of elongation speed. As a result, we did not observe filament-breaking, as shown in Figure 3b, to ascribe direct F-actin severing by 10 nM of Spa2 (1-535). This elaboration provides our response to the reviewer’s point R2 above.

We cannot definitively determine whether a concentration of 10 nM Spa2 in Extended Figure 4l fragments the F-actin in the solution, as suggested by the reviewer, as direct observation and quantification are not as practical and accurate as TIRF-based assay. However, in a separate experiment aimed at assessing Spa2’s impact on ATP-G-actin bundling activity, we employed a consistent concentration of 500 nM ATP-G-actin along with 5 μ M of Spa2 in Extended Figure 5f. In this instance, we still did not observe any noticeable Spa2-mediated direct depolymerization of F-actin. Taking all of this evidence into account, we are unable to draw a conclusion regarding Spa2’s direct severing activity.”

Thirdly, in our most recent revision, we introduced a discussion on Page 20 addressing the slightly enhanced Cof1 severing by Spa2, including the potential involvement of an IDR.

Fourthly, it's crucial to note that there is no Cof1 present in Figure 3d, Extended Figure 4g, and Extended Figure 4l. In response to the concerns about Spa2-mediated F-actin severing, we have presented supporting evidence in Extended Figure 4c and Extended Figure 5c, d. We maintain a high level of confidence in this conclusion, making it less likely that the observed "fast severing and fast diffusion" in Extended Figure 4l contributed to missing the Spa2-mediated severing mechanism. Additionally, the absence of suspected "severed shorter filaments" by Spa2 in Figure 3d strengthens our argument against the impact of fast diffusion. While acknowledging the reviewer's

point that fast severing coupled with fast diffusion might lead to oversight, we contend that this is not the case in our study, as explained in the previous rebuttal and reiterated here.

As reviewer suggested, we discussed the limitations now on page 21. We discuss two challenges in quantitatively comparing severing using initial short filaments in the TIRF assay. 1) the difficulty in precisely controlling the starting time of the reaction, leading to variations in the initial length of very short filaments, as clarified in the fourth rebuttal letter, and 2) the efficient removal of shorter filaments by diffusion in the chamber, particularly those not securely anchored despite the biotin-streptavidin layer coating on the coverslip. With such limitations, a Spa2-mediated F-actin severing assay presented in Extended Figure 4c and Extended Figure 5c, d offers a more reliable evaluation.

6) Please read my current and PREVIOUS criticisms again and address them carefully. I am afraid that the readers will find the same problems as in my review comments.

Response: We thank reviewer's kind reminder. Indeed, we have diligently addressed each question raised by the reviewer, taking great care to respond to each comment individually in every round of the review process. No question has been overlooked or omitted in any round. In cases of misunderstanding, we have made every effort to address, clarify, and rectify the issues with gratitude.

1 Jansen, S. *et al.* Single-molecule imaging of a three-component ordered actin disassembly mechanism. *Nature communications* **6**, 7202 (2015).